# MeGA-MP: Metric Graph Advection Message Passing

## Solving Dynamical Processes on Metric Graphs with Graph Neural Networks

## Abstract

Many real-world systems are organized as networks, where spatio-temporal dynamics unfold not only at nodes, but also along the connections between them. Such networks are known as *metric graphs*. Examples include utility networks and the propagation of signals in physical or biological media. The methods that approach such problems are mostly PINN-based with limited generalizability to PDE parametrization and boundary conditions. A recent work addresses the limitations of PINNs by proposing a neural operator for drift-diffusion dynamics. However, in many real-world settings, hyperbolic dynamics like advection dominate the spatial evolution of a system, which has not been addressed so far. In this work, we propose a novel graph operator that solves linear advection on metric graphs via message-passing. We provide an error bound on the approximation of ground truth obtained through multiple MP-iterations without the necessity of training. Empirically, we show that it solves advection competitive to numerical and neural solvers. Combined with trainable components like MLPs, we demonstrate how it can be applied to realistic advection-reaction dynamics in water distribution systems, where we achieve superior performance compared to baselines.

## 1 Introduction

In the last years, physics-informed graph neural networks (GNNs) have led to impressive zero-shot generalization capabilities when addressing dynamical systems on conventional graphs (Thangamuthu et al., 2022). Yet the study of dynamical processes on *metric graphs* using machine learning (ML) has only emerged recently (Blechschmidt et al., 2025; Laczkó et al., 2025). Such processes describe "quasi-one-dimensional systems coupled at vertices" (Laczkó et al., 2025) that model complex systems such as the spatial and temporal evolution of substances or information in physical or biological systems (Böttcher & Porter, 2024), for example in electrical grids, compressed air or water distribution systems. More concretely, each edge in a metric graph is associated with a one-dimensional domain (i.e., an interval of edge-dependent length) governed by a difference equation, an ordinary differential equation (ODE) or mostly, a partial differential equation (PDE). Together with suitable boundary conditions at nodes, such as continuity or conservation of mass, this defines a globally coupled system of PDEs (Böttcher & Porter, 2024; Blechschmidt et al., 2025).

Existing solutions on the one hand rely on numerical approaches (Böttcher & Porter, 2024) which are disconnected from end-to-end ML architectures, require computationally expensive sampling of function evaluations along each edge and have generally well-known limitations such as stability issues and a high demand for compute and memory. On the other hand, existing deep learning (DL) approaches (Blechschmidt et al., 2022; 2025; Laczkó et al., 2025) in this area either rely on PINN-based (Raissi et al., 2019) or on neural operator formulations (Lu et al., 2021). PINN-based methods have shown flexibility but typically require retraining when PDE parametrization such as flow fields, boundary conditions or initial values change (Blechschmidt et al., 2022; Laczkó et al., 2025). Operator learning approaches decompose the problem into edge-wise DeepONets and build a *lego*-like graph-agnostic strategy, but existing evaluations do not benchmark performance on advection-dominated problems with long-range spatial interactions that span multiple hops on the metric graph (Blechschmidt et al., 2025).

In this work, we aim to make a first step towards DL solutions for dynamic processes on metric graphs that circumvent these issues. We identify *advection*, i.e., the transport of a substance through

the presence of a flow field, as the main driver for common edge dynamics, ranging from utility networks, such as water distribution systems (Rossman et al., 2020), as well as traffic networks (Angulo & Burbano, 2025), oxygen or blood flow through vascular systems (Beard & Bassingthwaighte, 2001) and the propagation of signals in physical or biological media (Kajorndejnukul et al., 2015). We avoid computa-

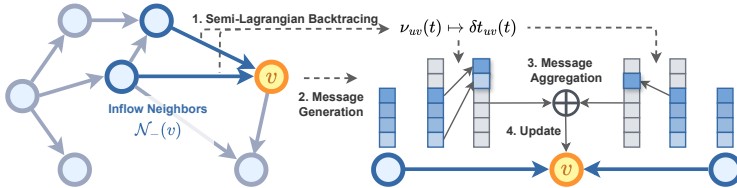

Figure 1: Initial iteration of MeGA-MP (Eq. (12)). Messages are passed from inflow neighbors $u \in \mathcal{N}_-(v)$ to the target node $v$ (for all $v \in V$): First, flow velocities $\nu_{uv}$ are converted to the transport time $\delta t_{uv}$ (Eq. (8)). Second, messages are generated via time-warping (Eq. (9)). Third, results are aggregated and assigned to the target node $v$ (Eq. (11)). The graph is directed according to the flow for visualization purposes. Grey entries correspond to padded entries and masked features according to Equation (12).

tionally expensive sampling of function evaluations along each edge by solving the advection PDE using the Method of Characteristics (MoC) and integrating the results in an iterative message passing neural network (MPNN) architecture. This enables an efficient and exact realization of the advection dynamics while maintaining end-to-end adaptability if observational data is available.

**Contributions** We propose *metric graph advection message passing (MeGA-MP)*, a MPNN framework that provably models linear advection on a metric graph iteratively without the necessity of learning (Section 3). We provide a rigorous theory of how MeGA-MP models the underlying physics exactly, given suitable boundary conditions of the edge-wise PDEs and smoothness assumptions (Section 4). We demonstrate the overall power of MeGA-MP by extending it with learning components to solve the spatio-temporal node forecasting task on a large-scale and realistic metric graph with advection-reaction edge dynamics (Section 5.1) and show how MeGA-MP can be used as a classical numerical solver to the advection equation on a one-dimensional domain (Section 5.2).

**Related Work** MeGA-MP can serve as a standalone solver for linear advection, but can also be integrated into a trainable MPNN to solve more complex, advection-dominated problems. Thus, our work lies at the intersection of physics-informed ML, numerical and neural PDE solvers.

Raissi et al. (2019) introduced physics-informed neural networks (PINNs), a framework for a neural network (NN) that models the unknown solution of a given PDE on a continuous time and space domain. Since PDE solutions depend on configurations like initial conditions, boundary conditions, and control signals, PINNs typically learn the solution to one of such configurations. Neural operators (Kovachki et al., 2023) address this limitation by learning a map from functions that configure a PDE to its solution. Prominent examples are DeepONet by Lu et al. (2019; 2021) and a variation of neural operators by Li et al. (2020a;b;c; 2024). While these methods approach learning solutions of PDEs on a Euclidean domain, they do not transfer to the graph domain. This issue was recently approached by Blechschmidt et al. (2022; 2025) and Laczkó et al. (2025), who applied a version of PINN or DeepONet edge-wise to solve the drift-diffusion and the time-independent Schrödinger equation on a metric graph, respectively, but with limitations as elaborated above.

An orthognal line of work has shown that the discretization of PDEs in time and space yield powerful physics-motivated MPNNs suitable for conventional tasks on graphs, such as node classification (Chamberlain et al., 2021; Eliasof et al., 2021; Rusch et al., 2022; Choi et al., 2023; Eliasof et al., 2024), but which can usually not be used to model dynamical processes on metric graphs. We provide a more detailed discussion on why advection-dominated dynamics on metric graphs come with challenges that cannot be addressed with these GNN architectures in Appendix A. Here, we also provide a visualization that emphasizes the difference of advection on metric graphs versus on non-metric graphs (Figure 5).

## 2 BACKGROUND AND TASK DEFINITION

The domain we are considering is a metric graph, which can be modeled as a finite, connected and symmetric directed graph $\mathcal{G} = (V, E)$ with nodes $V = \{v_1, \ldots, v_{n_\mathrm{n}}\}$ and edges $E = \{e_1, \ldots, e_{n_\mathrm{e}}\}$,

where each edge $e_{vu} \in E$ connects a node $v \in V$ to one of its neighbors $u \in \mathcal{N}(v)$ and is additionally equipped with a length $l_{e_{vu}} > 0$ and associated with edge dynamics as detailed below.

**Message Passing** has become a common principle in GNNs (Gilmer et al., 2017). It describes how local information at the nodes of a graph is passed to neighboring nodes. Message passing as defined in Gilmer et al. (2017) consists of three consecutive steps: Message generation, message aggregation, and node update. Details can be found in Appendix A.

**Advection** on a spatial domain $\Omega \subset \mathbb{R}^d$ describes how a substance is transported by the presence of a flow field $\nu : T \times \Omega \to \mathbb{R}$ over time $T = \mathbb{R}$ and through space $\Omega$. For a substance that is characterized by its concentration $c : T \times \Omega \to \mathbb{R}_{\geq 0}$, one-dimensional linear advection is defined for each $t \in T$ and $z \in \Omega$ by the PDE

$$\partial_t c(t, z) = -\nu(t, z)\, \partial_z c(t, z). \tag{1}$$

**Advection on Metric Graphs** If the overall domain corresponds to a graph $\mathcal{G} = (V, E)$, advection – such as any other dynamics – can be approached from two distinct paradigms. In the first, graphs discretize an underlying continuous space, with nodes representing spatial locations and edges encoding adjacency based on geometric proximity. This setup preserves the structure of the continuum, e.g., Euclidean spaces, and is common in numerical schemes for PDEs or physics-motivated MPNN architectures such as Eliasof et al. (2024). In contrast, a dynamical process such as advection on a *metric graph* is independent of an underlying continuous domain. Here, edges carry explicit lengths $l_{uv} > 0$ and, potentially, capacities $\alpha_{uv} > 0$, and are associated with a one-dimensional domain governed by a PDE. This enables the modeling of transport phenomena across heterogeneous or non-spatial networks, such as electrical grids, compressed air or water distribution systems. We highlight the difference on these paradigms in Figure 5 in Appendix A.

On a metric graph $g$, the general goal is to learn the propagation of a signal $\mathbf{c} : V \to F(\mathbb{R}, \mathbb{R}_{\geq 0})$, which encodes the temporal evolution $\mathbf{c}(v) = c_v : \mathbb{R} \to \mathbb{R}_{\geq 0}, t \mapsto c_v(t)$ of the signals per node $v \in V$ and over time $\mathbb{R}$, subject to the following two properties:

1. The *information transfer* in-between two nodes $v \in V$ and $u \in \mathcal{N}(v)$ can be described by the metric graph's *edge dynamics* governed by a PDE along the edge $e_{uv} \in E$, that is,

   $$\partial_t c_{uv}(t, z) = f(t, z, c_{uv}, \nu_{uv}) \quad \text{for all } t \in \mathbb{R} \text{ and } z \in (0, l_{uv}), \tag{2}$$

   where $c_{uv} : \mathbb{R} \times (0, l_{uv}) \to \mathbb{R}$ corresponds to the signal of interest along the one-dimensional sub-domain $\Omega_{uv} = (0, l_{uv})$, i.e., the edge, and $\nu_{uv} : \mathbb{R} \times (0, l_{uv}) \to \mathbb{R}$ is a known function that parametrizes the edge-wise PDE through a suitable function $f$.

2. The *information aggregation* can be described by

   $$c_v(t) = \sum_{u \in \mathcal{N}_-(v,t)} w_{uv}(t)\, \phi(t, e_{uv}, \mathbf{c}, \boldsymbol{\nu}), \tag{3}$$

   where $w_{uv} : T \to \mathbb{R}_{\geq 0}$ denotes normalized weights over information-sending *inflow neighbors* $u \in \mathcal{N}_-(v,t)$ at time $t \in \mathbb{R}$, $\boldsymbol{\nu} : E \mapsto F(\mathbb{R} \times (0, l_{uv}), \mathbb{R})$ encodes – similar to the signal $\mathbf{c}$, but with a slight abuse of notation – the temporal evolutions $\nu_e : \mathbb{R} \times (0, l_e) \mapsto \nu_e(t, z)$ per time $t \in \mathbb{R}$ and edge $e \in E$ and $\phi$ describes the information transfer from the inflow neighbors $u \in \mathcal{N}_-(v,t)$ of $v$ determined by Equation (2).

If the edge dynamics are advection-dominated, $f$ is defined by Equation (1) (or an extension of such) for each concentration $c_{uv}$ carried by the flow field $\nu_{uv}$ along the edge $e_{uv} \in E$. We use this PDE to specify Equation (3) for advection dynamics in Section 3 and rigorously derive it under physical considerations in Section 4.

For incompressible flow $\boldsymbol{\nu}$, which we focus on here, the following identities hold:

$$\nu_{uv}(t, \cdot) = const. \text{ and } \nu_{vu}(t, \cdot) = -\nu_{uv}(t, \cdot). \tag{4}$$

Therefore, we omit the space-dependency of the flow field $\nu_{uv}$ and only consider it as a function $\nu_{uv} : T \to \mathbb{R}$ over time $T = \mathbb{R}$. We moreover use the convention that the sign of a flow velocity $\nu_{uv}(t)$ is positive (negative) if the direction of the corresponding edge $e_{uv} \in E$ aligns (does not align) with the direction of the physical flow, i.e., if the physical flow is from $u \in \mathcal{N}(v)$ to $v \in V$ (from $v \in V$ to $u \in \mathcal{N}(v)$). For details, we refer to Appendix A.

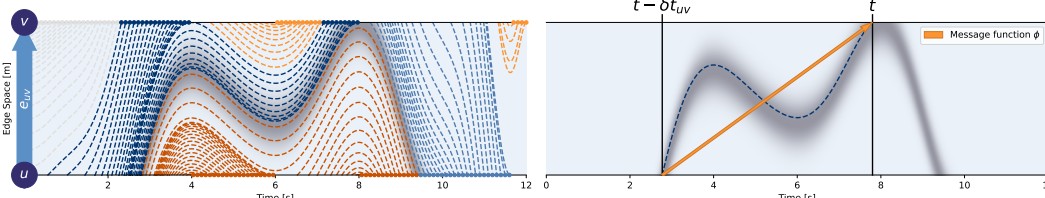

Figure 2: Characteristic curves along the edge $e_{uv} \in E$: If a curve originates at node $u$ at time $t - \delta t_{uv}$ and terminates at node $v$ at time $t$, we call it a *pass-through* (dark blue). If the direction is reversed, we call it *inverse pass-through* (light blue). If a curve starts and ends at the same node, we call it a *self-loop* (dark orange) if that node is $u$ and *inverse self-loop* (light orange) if that node is $v$ (start or terminal node of an edge). The corresponding transport time $\delta t_{uv} > 0$ (right) denotes the duration between the curve's start and end. The shaded area (gray) in the background shows a high-concentration pulse entering the domain at node $u$, that is constant along characteristic curves.

**Problem Definition**   Practically, dynamic processes on metric graphs translate to the task of *spatio-temporal node forecasting with boundary conditions*: We access the signal $\mathbf{c}$ at discrete times $t_i \in \mathbb{R}$ for suitable indices $i \in I$ from an index set $I$. Fixing a discrete time $t_i \in \mathbb{R}$, we denote $T_{\mathrm{hi}}(t_i) = \{t_{i-k_{\mathrm{hi}}+1}, ..., t_i\}$ as the *history set* consisting of the $k_{\mathrm{hi}} \in \mathbb{N}$ discrete times at which we have knowledge of $\mathbf{c}$ and $T_{\mathrm{pr}}(t_i) = \{t_{i+1}, ..., t_{i+k_{\mathrm{pr}}}\}$ as the *prediction set* consisting of $k_{\mathrm{pr}} \in \mathbb{N}$ discrete times at which we do not have knowledge of $\mathbf{c}$ yet, but which we want to learn based on the knowledge that we have. For simplicity, we omit the dependency of $T_{\mathrm{hi}}$ and $T_{\mathrm{pr}}$ on $t_i$ if $t_i$ is clear from the context. If we have *boundary conditions* of the dynamical system on the metric graph, this knowledge extends to values of $\mathbf{c}$ at *all* discrete times at a subset of nodes $V_{\mathrm{b}} \subsetneq V$. Given that knowledge, for each discrete time $t_i \in \mathbb{R}$, we want to to learn the mapping $\left( (c_v(t))_{v \in V \setminus V_{\mathrm{b}}, t \in T_{\mathrm{hi}}}, (c_v(t))_{v \in V_{\mathrm{b}}, t \in T_{\mathrm{hi}} \sqcup T_{\mathrm{pr}}} \right) \longmapsto (c_v(t))_{v \in V \setminus V_{\mathrm{b}}, t \in T_{\mathrm{pr}}}$.

In this work, we identify Equation (3) as a special case of message-passing (Eq. (13)). We also identify *advection* as the main driver of more complex dynamical processes on graphs, and therefore hypothesize that modeling the advection in a message passing framework without learning components will help to solve more complex tasks. We discuss why this is a challenging task that can not be solved with conventional GNN architectures in Appendix A.

Therefore, in the Section 3, we illustrate how to integrate the physical priors of information transfer (Eq. (2)) and information aggregation (Eq. (3)) presented in this section into a novel MPNN that only requires knowledge of the system states at nodes and the flow field along edges to solve the spatio-temporal node forecasting task on metric graph with advection dynamics.

## 3   METHOD: METRIC GRAPH ADVECTION MESSAGE PASSING

In this section, we describe our message-passing method to solve advection problems on metric graphs based on the general concepts introduced in Section 2. In this context, the signal $\mathbf{c}$ usually refers to a **c**oncentration. The mathematical structure of Equation (3) is already very close to message passing (Eq. (13)). Building upon this insight, in Section 3.1, we relate the *information transfer* to *message generation* by deriving the advection operator $\phi$ as the message generation function in message passing (Eq. (13)) from the general advection formula (Eq. (1)). Consecutively, we relate *information aggregation* to *message aggregation and update* in Section 3.2 and present the overall MeGA-MP algorithm in Section 3.3.

### 3.1   MESSAGE GENERATION: ADVECTIVE TRANSPORT ALONG EDGES

**Characteristic curves**   We model advection of the concentration $c_{uv} : \mathbb{R} \times (0, l_{uv}) \to \mathbb{R}_{\geq 0}$ along an edge $e_{uv} \in E$ in between two nodes $v \in V$ and $u \in \mathcal{N}(v)$ as the transport process governed by Equation (1) with incompressible flow field $\nu_{uv} : \mathbb{R} \to \mathbb{R}$. In this case, by the MoC, we can derive *characteristic curves* along which the function $c_{uv}$ remains constant over time $\mathbb{R}$ and space $(0, l_{uv})$:

**Lemma 3.1** (Constant concentration along characteristic curves). *Let $v \in V$ and $u \in \mathcal{N}(v)$ hold. If the function $c_{uv}$ obeys Equation (1), for each $t_0 \in \mathbb{R}$ and $z_0 \in \mathbb{R}$, the curve*

$$\gamma_{t_0, z_0} : T_{t_0, z_0} \longrightarrow \mathbb{R}_{\geq 0}, \ \ t \longmapsto c_{uv}\left(t, z_0 + \int_{t_0}^t \nu_{uv}(s) \, ds\right) \text{ is constant.}$$

An intuition on Lemma 3.1 can be found in Appendix C.1. Figure 2 (left) visualizes several characteristic curves for an exemplary flow field $\nu_{uv} : \mathbb{R} \to \mathbb{R}$ along an edge $e_{uv} \in E$ of length $l_{uv} = 1[m]$. The heatmap in the background shows a high-concentration pulse that travels from the one boundary at $z = 0$ to the other at $z = l_{uv}$ in $\delta t_{uv} \approx 5[s]$. Following Lemma 3.1, the concentration $c_{uv}$ is the same at all positions along such a curve. Consecutively, for the example given in Figure 2 (right), knowing the concentration $c_u(t - \delta t_{uv})$ at the node $u \in \mathcal{N}(v)$ ($z = 0$) is sufficient to determine the concentration $c_v(t)$ at node $v \in V$ ($z = l_{uv}$).

**Transport times** To formulate this relationship as a message function, we need a way to compute the start and end coordinates of characteristic curves that correspond to the sender and receiver node, respectively, as well as the time shift $\delta t_{uv} > 0$, which we generally speaking denote as *transport time*. Among such, we use specific terminology to differentiate the following situations, which are also described visually in Figure 2 (left). To get a deeper understanding of the different kinds of transport times, details can also be found in Appendix C.1 (Definition C.6 up to Theorem C.12).

**Definition 3.2** (Transport times). *Let $v \in V$, $u \in \mathcal{N}(v)$ and $t \in \mathbb{R}$ hold. If there exists a $\delta t_{uv} := \delta t_{uv}(t) \in \mathbb{R}$, such that*

$$\int_{t - \delta t_{uv}}^t \nu_{uv}(s) \, ds = l_{uv} \qquad\qquad (-l_{uv}, 0, 0) \text{ and} \tag{5}$$

$$\int_{t - \delta t_{uv}}^{t'} \nu_{uv}(s) \, ds \in (0, l_{uv}) \qquad\qquad ((-l_{uv}, 0), (0, l_{vu}), (-l_{uv}, 0)) \tag{6}$$

*holds for all $t' \in (\min\{t - \delta t_{uv}, t\}, \max\{t - \delta t_{uv}, t\})$, we call $\delta t_{uv}$ the pass-through (inverse pass-through, self-loop, inverse self-loop) time with respect to (w.r.t.) the edge $e_{uv} \in E$ and time $t \in \mathbb{R}$. Usually, $e_{uv} \in E$ and $t \in \mathbb{R}$ are clear from the context and we just say pass-through (inverse pass-through, self-loop, inverse self-loop) time, just as we omit the dependency of $\delta t_{uv}$ on $t$ for better readability. In summary, we call such times transport times.*

**Message function** If the flow $\nu_{uv}$ is constant over time, the transport time $\delta t_{uv}$ is also constant, and concentrations are shifted between the nodes $u$ and $v$. If the flow $\nu_{uv}$ varies, so does the transport time $\delta t_{uv}$, and instead of a clear shift, we observe more complex *time warping*. As time warping generalizes shift, we use the prior as our message function. In order to identify the correct transport time for given time $t \in \mathbb{R}$, we need to measure the distance traveled in an arbitrary time interval $[t - \delta t, t]$. For this purpose, we define the function

$$\mathrm{z} : \mathbb{R} \longrightarrow \mathbb{R}, \ \delta t \longmapsto \mathrm{z}(\delta t) := \int_{t - \delta t}^t \nu_{uv}(s) \, ds, \tag{7}$$

where we omit the dependency on $v \in V$, $u \in \mathcal{N}(v)$, $t \in \mathbb{R}$ and the overall flow field $\boldsymbol{\nu}$ for better readability. Using $\mathrm{z}$, we can compute the transport time as the solution to the following optimization problem (OP):

$$\delta t_{uv} := \min\left\{ \delta t \in (0, \infty) \mid \mathrm{z}(\delta t) \in \{-l_{uv}, 0, l_{uv}\} \right\}. \tag{8}$$

In Appendix C.1, we prove that if there exists a solution, this solution uniquely determines one of the transport times from Definition 3.2 (cf. Theorem C.13). This in turn allows to define our advection-message-generation functions as

$$\phi : (t, e_{uv}, \mathbf{c}, \boldsymbol{\nu}) \longmapsto \phi(t, e_{uv}, \mathbf{c}, \boldsymbol{\nu}) := \begin{cases} c_u(t - \delta t_{uv}) & \text{if } \mathrm{z}(\delta t_{uv}) \in \{-l_{uv}, l_{uv}\}, \\ c_v(t - \delta t_{uv}) & \text{if } \mathrm{z}(\delta t_{uv}) = 0. \end{cases} \tag{9}$$

Especially to mention, using $\mathrm{z}$ again, $\phi$ automatically identifies whether information is carried along a(n) (inverse) pass-through or a(n) (inverse) self-loop and outputs the information from the node $v$ itself or its neighbor $u \in \mathcal{N}(v)$ accordingly. A rigorous derivation of Equation (9) under consideration of physical priors and the intuition behind this can be found in Appendix C.1.

## 3.2 MESSAGE AGGREGATION AND UPDATE: MIXING AT NODES

We model the mixing of concentrations at nodes that determines the concentration $c_v(t)$ at a node $v \in V$ and time $t \in \mathbb{R}$ as a realization of Equation (3) with normalized flow rates as weights

$$w_{uv}(t) = \left( \sum_{u \in \mathcal{N}_-(v,t)} q_{uv}(t) \right)^{-1} q_{uv}(t), \tag{10}$$

each flow rate given by $q_{uv}(t) = \alpha_{uv}\nu_{uv}(t)$, and the advection-message-generation function $\phi(t, e_{uv}, \mathbf{c}, \boldsymbol{\nu})$ as defined in Equation (9). This results in our final message-passing scheme

$$c_v(t) = \frac{\sum_{u \in \mathcal{N}_-(v,t)} q_{uv}(t) \cdot \phi(t, e_{uv}, \mathbf{c}, \boldsymbol{\nu})}{\sum_{u \in \mathcal{N}_-(v,t)} q_{uv}(t)} = \frac{\sum_{u \in \mathcal{N}(v)} \mathrm{ReLU}(q_{uv}(t)) \cdot \phi(t, e_{uv}, \mathbf{c}, \boldsymbol{\nu})}{\sum_{u \in \mathcal{N}(v)} \mathrm{ReLU}(q_{uv}(t))}. \tag{11}$$

## 3.3 FINAL ALGORITHM: MEGA-MP

So far, we considered the continuous time domain $\mathbb{R}$. In practice, we model the temporal concentration $c_v : \mathbb{R} \to \mathbb{R}_{\geq 0}$ at a node $v \in V$ as a discrete time series $(c_v(t_i))_{i \in I}$ over equidistant discrete times $t_i \in \mathbb{R}$ with indices $i \in I$. For each $t_i \in \mathbb{R}$ (but omitting the dependency on $t_i$ again for simplicity), we want to learn $\mathbf{y}_v := (c_v(t))_{t \in T_{\mathrm{pr}}}$ simultaneously for all $v \in V \setminus V_{\mathrm{b}}$ (cf. Problem Definition in Section 2). We do so by first initializing $\hat{\mathbf{c}}_v^{(0)} := (\mathbf{x}_v^T, \mathbf{0}_{k_{\mathrm{pr}}}^T) \in \mathbb{R}^{k_{\mathrm{hi}}+k_{\mathrm{pr}}}$, i.e., we pad the known history $\mathbf{x}_v := (c_v(t))_{t \in T_{\mathrm{hi}}}$ with zeros, which are placeholders for the predictions $\hat{\mathbf{y}}_v := (\hat{c}_v(t))_{t \in T_{\mathrm{pr}}} \in \mathbb{R}^{k_{\mathrm{pr}}}$ of $\mathbf{y}_v$ (cf. Figure 1 and first row of Eq. (12) below). Consecutively, we generate the predictions $\hat{c}_v(t)$ by applying our message-passing scheme (11) simultaneously for all $v \in V$ and $t \in T_{\mathrm{pr}}$, which requires access to the known histories $\mathbf{x}_v$, with details as follows:

**Interpolation** As a consequence of the discretized time series but continuous edge lengths $l_{uv}$ and flows $\nu_{uv}$, the required concentration at time shifts $t - \delta t_{uv}$ appearing in the message function $\phi$ (cf. Eq. (9)) is usually not part of the known history $\mathbf{x}_u$. Therefore, time warping is implemented as interpolation in-between the two values at times $t_j$ and $t_{j+1}$ for which $t - \delta t_{uv} \in [t_j, t_{j+1}]$ holds. Details can be found in Appendix B.1. We denote the approximated version of $\phi$ by $\varphi$.

**Model Iterations** For a node $v \in V \setminus V_{\mathrm{b}}$, assigning $\hat{c}_v(t)$ directly to the outcome of our message-passing scheme (11) only works for times $t \in T_{\mathrm{pr}}$ for which the required concentrations at time shifts $t - \delta t_{uv}$ lies in the known histories $\mathbf{x}_u$, i.e., for which $t - \delta t_{uv} < t_i$ holds for all $u \in \mathcal{N}(v)$. This puts a constraint on the overall flow field $\boldsymbol{\nu}$. However, generalizing this to arbitrary flow fields can be achieved by iterating Equation (11) via

$$\hat{c}_v^{(0)}(t) := \begin{cases} c_v(t) & \text{if } t \in T_{\mathrm{hi}} = \{t_{i-k_{\mathrm{hi}}+1}, ..., t_i\} \\ c_v(t) & \text{if } t \in T_{\mathrm{pr}} = \{t_{i+1}, ..., t_{i+k_{\mathrm{pr}}}\} \text{ and } v \in V_{\mathrm{b}} \\ 0 & \text{if } t \in T_{\mathrm{pr}} = \{t_{i+1}, ..., t_{i+k_{\mathrm{pr}}}\} \text{ and } v \in V \setminus V_{\mathrm{b}} \end{cases},$$

$$\hat{c}_v^{(k)}(t) := \begin{cases} 0 & \text{if } t \in T_{\mathrm{hi}} \\ 0 & \text{if } t \in T_{\mathrm{pr}} \text{ and } v \in V_{\mathrm{b}} \\ \sum_{u \in \mathcal{N}_-(v,t)} w_{uv}(t) \cdot \varphi(t, e_{uv}, \hat{\mathbf{c}}^{(k-1)}, \boldsymbol{\nu}) & \text{if } t \in T_{\mathrm{pr}} \text{ and } v \in V \setminus V_{\mathrm{b}} \end{cases} \quad \forall\, k \geq 1, \tag{12}$$

$$\hat{c}_v(t) := \sum_{k=1}^{\infty} \hat{c}_v^{(k)}(t) \text{ for } t \in T_{\mathrm{pr}},$$

where $\hat{\mathbf{c}}^{(k-1)}$ corresponds to the discretized version of the function $\mathbf{c} = (c_v)_{v \in V}$ (cf. Eq. (9)) which returns the values $\hat{c}_v^{(k-1)}(t)$ for all $v \in V$ and $t \in T_{\mathrm{hi}} \cup T_{\mathrm{pr}}$ and zero else. To build an intuition for how the iterations relax the constraint from $\boldsymbol{\nu}$, note that each iteration moves information forward in time by a discrete time step corresponding to the transport time $\delta t_{uv} > 0$ and along the flow direction from neighboring nodes (excluding self-loops). This can be thought of as a path along which information travels (cf. Figure 2). With each iteration, the path extends further, tracing out a spatio-temporal tree-like trajectory. As a result, a node receives information from increasingly distant nodes at progressively earlier times, eventually from either the known history $T_{\mathrm{hi}}$ or boundary nodes $V_{\mathrm{b}}$. A formalization of this intuition can be found in Appendix B.2. The initial iteration ($k = 0 \mapsto k = 1$) is visualized in Figure 1. Subsequent iterations follow the same schema, but with node feature vectors already containing information at future time steps $T_{\mathrm{pr}}$.

**Theorem 3.3** (Convergence of model iterations). *There exists a $\kappa \in \mathbb{N}$, such that $\hat{c}_v^{(k)}(t)$ as defined in Equation (12) is equal to zero for all $k > \kappa$, $v \in V$ and $t \in T_{hi} \cup T_{pr}$.*

Theorem 3.3 assures that the prediction $\hat{\mathbf{y}}_v = (\hat{c}_v(t))_{t \in T_{\text{pr}}}$ defined by the last row of Equation (12) is computable in finite time for all $v \in V \setminus V_b$. Finally, we show that MeGA-MP (Eq. (12)) reproduces the ground-truth (Eq. (9)) if the interpolation error is zero. An extended version of Theorem 3.4 with an error bound obtained through the interpolation can be found in Appendix B.2.

**Theorem 3.4** (Interpolation error). *Let $v \in V$ and $t \in \mathbb{R}$ hold. Let Assumption B.5 and B.6 hold. If the* temporal *interpolation error is zero, i.e., if $\phi = \varphi$ holds, we obtain $\hat{c}_v(t) = c_v(t)$.*

*Remark* 3.5 (Initial value problems and efficiency). For the sake of efficiency in both computational and memory complexity, we represent the solution of advection on a metric graph as time series at nodes only (cf. Problem Definition in Section 2 and Eq. (12)). The actual values along the 1-dimensional edge spaces are solved implicitly. This enables our model to scale linearly with the number of nodes and, consequently, to handle even large graphs. For this advantage, our method cannot directly receive classical initial value problems (IVPs), where dense values $c_{uv}(t, z)$ along the edge spaces $\mathbb{R} \times (0, l_{uv})$ are known at $t = 0$, except when they are constant. However, initial values along edges can be transformed into time series at nodes and vice versa. This is visualized by the gray characteristic curves in Figure 2. We will demonstrate this in Appendix D.2 as an extension of Experiment 5.2.

**Runtime Complexity** We now discuss the runtime complexity of MeGA-MP. Corresponding algorithms can be found in Appendix B.3. The runtime of MeGA-MP depends on five quantities: Edge lengths $l_{uv}$, the velocity field $\nu_{uv}$, the history and predictions steps $k_{\text{hi}}$ and $k_{\text{pr}}$, respectively, and the number of edges $|E|$ in the graph.

On the one hand, the edge lengths, velocities and the prediction horizon together determine the spatial extent of dependencies, i.e., the number of edges that information can traverse within the prediction horizon. This, in turn, corresponds to the number of message passing iterations required to solve the dynamics, also see Appendix A. An upper bound on the number of message passing iterations can be defined as $N = \frac{t_{i+k_{\text{pr}}} - t_i}{\omega_1}$, where $\omega_1 = \min_{e_{uv} \in E, t \in T_{pr}} \frac{l_{uv}}{\nu_{uv}(t)}$. In line with other message-passing algorithms, MeGA-MP also applies a message function per edge that processes vectors of length $T = k_{\text{hi}} + k_{\text{pr}}$, resulting in a worst-case runtime of $\mathcal{O}(N|E|T)$.

On the other hand, computing the transport times $\delta t_{uv}$ from the flow field $\nu_{uv}(t)$ (Algorithm 1) also contributes to the runtime. Again, the runtime depends on edge lengths and velocities. Specifically, the worst case number of iterations of the inner loop is $\omega_2 = \min(T, \max_{e_{uv} \in E, t \in T_{pr}} \frac{l_{uv}}{\nu_{uv}(t)})$, and since the outer loop iterates $T$ times, the resulting complexity is $\mathcal{O}(\omega_2 T)$ with an upper bound of $\mathcal{O}(T^2)$. Combining this with the message passing complexity yields an overall runtime of $\mathcal{O}(T^2 + N|E|T)$. Note that Algorithm 1 is a naive implementation and using more advanced data structures and imposing smoothness assumptions on the flow field could reduce the runtime complexity drastically.

## 4 PHYSICAL DERIVATION OF ADVECTION MESSAGE PASSING

The formulation of our advection-message-generation function (9) and the choice of weights that yield our final message-passing scheme (11) are not chosen arbitrarily. Indeed, MeGA-MP naturally inherits the physics of an advection metric graph, which satisfies the two physical principles (information transfer and information aggregation) introduced in Section 2 together with suitable boundary conditions and smoothness assumptions that model real-world systems and connect the two principles. Due to space constraints, in Appendix C, we present a rigorous derivation of MeGA-MP under this realm. Here, we only highlight the main physical priors and results of that section.

The information transfer and the information aggregation (cf. Sec. 2) can be linked via inflow and outflow boundary conditions of the edge-wise PDE (Eq. (2)). Intuitively, if there is a (physical) flow leaving a node and entering an edge, the concentration at the beginning of that edge is equal to the concentration at that node (*inflow boundary condition*). In contrast, if there is a (physical) flow entering a node and leaving an edge, the concentration at the end of that edge contributes to the weighted aggregation at that node (*outflow boundary condition*) (cf. Definition C.1). The latter

is defined by the *mixing at nodes*, which corresponds to a realization of the information aggregation (Eq. (3)) with weights as defined in Equation (10) and $\phi(t, e_{uv}, \mathbf{c}, \boldsymbol{\nu}) = c_{uv}(t, l_{uv})$. That is, while the information transfer is defined on the open sub-domain $\mathbb{R} \times (0, l_{uv})$ only, the information aggregation relies on the values $c_{uv}(\cdot, l_{uv})$ on the boundary $\partial(0, l_{uv}) = \{0, l_{uv}\}$ of $(0, l_{uv})$. Therefore, suitable smoothness assumptions (Assumption C.2) on $c_{uv}$ and $\nu_{uv}$ are required in order to shift information in-between nodes and edges.

Under these assumptions, we extend the findings from Lemma 3.1 and investigate the different phenomena as introduced in Definition 3.2 more thoroughly. Based on that, we provide a rigorous theory of why our message-passing scheme (Eq. (11)) models the described physics, i.e., the information transfer defined by the advective transport along edges, information aggregation defined by the mixing at nodes as outflow boundary conditions and the inflow boundary conditions, *exactly*:

**Theorem** (Theorem C.16). *Let $v \in V$, $u \in \mathcal{N}(v)$ and $t \in \mathbb{R}$ hold. Let $\phi$ be as defined in Equation (9). We assume that for each $u \in \mathcal{N}(v)$, the OP (8) has a solution. If the function $c_v$ obeys the mixing at nodes and if for each $u \in \mathcal{N}(v)$, the function $c_{uv}$ obeys Equation (2) with inflow and outflow boundary conditions, we obtain that Equation (11) holds.*

## 5 EXPERIMENTS

We empirically evaluate our method in two ways: First, we demonstrate how our message-passing method can be used in combination with other machine learning modules. As an example, we set up an advection-reaction dynamical system on a real-world metric graph, a water distribution system (WDS), and learn the reaction dynamics with a small multi layer perceptron (MLP) (Section 5.1). Second, we set up an advection dynamical system and solve boundary value problems (BVPs) with our method. We then compare our results to those of existing numerical solvers (Section 5.2).

### 5.1 EXPERIMENT 1: LEARNING ADVECTION AND REACTION ON A REAL-WORLD METRIC GRAPH

In this Section, we demonstrate how MeGA-MP can be extended by learnable components to solve more complex dynamics than advection on metric graphs.

**Domain** We consider WDSs, in which nodes, such as reservoirs or consumer junctions, are connected through pipes, which can be modeled as the edges of a metric graph. An example of the WDS L-Town (Vrachimis et al., 2020) is displayed in

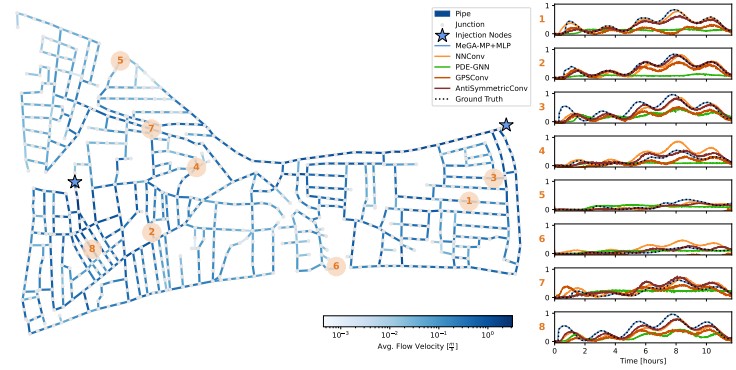

Figure 3: Predictions of baselines and MeGA-MP for the advection-reaction dynamical system at a subset of nodes of the WDS L-Town.

Figure 3. Another example of the WDS Hanoi (Vrachimis et al., 2018) can be found in Figure 7 in Appendix D.2. Both display well-known benchmark networks from the water community.

A common task in WDSs is to predict the distribution of a substance throughout the system. The two main drivers for such processes are advection and *reactions* (Rossman et al., 2020). Reaction processes describe purely temporal changes in state variables, such as chemical and physical transformations, interactions with the environment and decay. Typically, such dynamics are expressed as an ODE. When combined with advection, they extend Equation (2) as

$$\partial_t c_{uv}(t, z) = f(c_{uv}(t, z), \nu_{uv}(t, z)) = -\nu_{uv}(t)\, \partial_z c_{uv}(t, z) + f_r(t, z, e_{uv}),$$

where $f_r$ is a substance-specific reaction term. For this experiment, we simulate the injection of chlorine into the WDS. Chlorine decays over time due to chemical and physical interactions. Details on the specific formalization can be found in Appendix D.1.

**Dataset** We generate training and test data with and without reaction using EPyt-Flow (Artelt et al., 2024) based on EPANET-MSX (Shang et al., 2023), a hydraulic and water quality simulator for WDSs that supports advection and user-defined reactions such as Equation (33) in Appendix D.1. Details can also be found in Appendix D.1.

**Model** We extend MeGA-MP defined in Section 3 by a learnable component in order to be able to learn the reaction part of the advection-reaction dynamics. Specifically, we extend the message function $\phi$ of Equation (11) by an MLP. We design $\phi_r$ to be suitable for first-order kinetics, where the solution will be an exponential function. The resulting message function $\phi_r$ is given by

$$\phi_r(t, e_{uv}, \mathbf{c}, \boldsymbol{\nu}) = \phi(t, e_{uv}, \mathbf{c}, \boldsymbol{\nu}) \cdot \exp(|\delta t_{uv}| \cdot \mathrm{MLP}(\rho_{uv}^{-1})),$$

where $\rho_{uv} \in [0, 1]$ denotes normalized pipe diameters. Details on the architecture and the training of the MLP can also be found in Appendix D.1. Especially to mention, in order to show the ability of our model for transfer learning, we train it on the WDS Hanoi (cf. Figure 7) and test it on both Hanoi and the unseen WDS L-Town (cf. Figure 3 and 8). Moreover, as an ablation study to investigate on the contribution of both the non-learnable advection operator $\phi$ and the learnable MLP, we also conduct the advection-reaction experiment without learning components as well as the advection experiment with and without learning components.

**Baselines** We compare ourselves to the GNN by Gilmer et al. (2017) ("NNConv"), the transformer GNN by Rampášek et al. (2022) ("GPSConv"), the dynamic-system based GNN by Gravina et al. (2023) ("A-DGN") and the GNN by Hermes et al. (2025), which is tailored to WDSs ("PDE-GNN"). For details regarding these baselines, we also refer to Appendix D.1. There, we also provide details on why the adaptation of the more general methods from Blechschmidt et al. (2022; 2025) and Laczkó et al. (2025) were not applicable to our task.

**Results** In Figure 3, we showcase the ability of MeGA-MP to generalize over different networks, i.e., WDSs, and compare it to the result of the simulator and to baselines. More specifically, the figure shows the advection-reaction dynamical system on the metric graph L-Town, which was not used for training. MeGA-MP models all of the visualized nodes accurately, while NNConv and A-DGN can not accurately account for different reaction shapes over different nodes and GPSConv as well as PDE-GNN show severe smoothing behavior. In Section D.1, we also provide the results for advection-reaction dynamics on the training metric graph Hanoi and for the experiment without reaction. Moreover, the results of the ablation studies show that indeed, MeGA-MP without learning components performs poorly on the advection-reaction dynamics while MeGA-MP without learning components performs equally well on the advection

Table 1: MAE between simulator ground truth (EPANET) and solver solution of the advection-reaction dynamical system over time and all nodes of the training WDS Hanoi and the test WDS L-Town for different methods. Moreover, the computation time for computing the solution on both networks and for each method (batch size = 1) as well as for the simulator (EPANET).

| | WDS | Hanoi | L-Town |
|---|---|---|---|
| | | $|V| = 32, |E| = 34$ | $|V| = 785, |E| = 909$ |
| MAE | NNConv | $0.0474_{\pm 0.0650}$ | $0.0997_{\pm 0.0973}$ |
| | PDE-GNN | $0.0836_{\pm 0.1070}$ | $0.1968_{\pm 0.1822}$ |
| | GPSConv | $0.3756_{\pm 2.1406}$ | $0.3132_{\pm 1.8816}$ |
| | A-DGN | $0.0283_{\pm 0.0560}$ | $0.0865_{\pm 0.0915}$ |
| | A-DGN$_{Dia}$ | $0.0597_{\pm 0.0661}$ | $0.1572_{\pm 0.1278}$ |
| | MeGA-MP | $0.1540_{\pm 0.1527}$ | $0.1729_{\pm 0.1560}$ |
| | MeGA-MP+MLP | $\mathbf{0.0017}_{\pm 0.0023}$ | $\mathbf{0.0015}_{\pm 0.0061}$ |
| Time [s] | NNConv | $0.002_{\pm 0.000}$ | $0.002_{\pm 0.000}$ |
| | PDE-GNN | $2.370_{\pm 0.079}$ | $2.387_{\pm 0.090}$ |
| | GPSConv | $0.007_{\pm 0.000}$ | $0.009_{\pm 0.000}$ |
| | A-DGN | $0.007_{\pm 0.014}$ | $0.006_{\pm 0.000}$ |
| | A-DGN$_{Dia}$ | $0.009_{\pm 0.015}$ | $0.042_{\pm 0.000}$ |
| | MeGA-MP | $0.050_{\pm 0.002}$ | $0.480_{\pm 0.018}$ |
| | MeGA-MP+MLP | $0.059_{\pm 0.007}$ | $0.500_{\pm 0.025}$ |
| | EPANET | $0.683_{\pm 0.069}$ | $30.512_{\pm 0.493}$ |

dynamics as MeGA-MP+MLP - the one with learning components. This shows that indeed, MeGA-MP does not require any learning components to model advection alone accurately while the learning components are necessary to model more complex, advection-dominated dynamics.

Table 1 reports the mean absolute error over all nodes and time steps showing superior performance as compared to the baselines. Additionally, we report the runtime of each method. Since we use the transport times as input to each model, we report the execution time of the model without the time it takes to convert the flow field to transport times. Each of the model is faster than the EPANET simulator that we use as the ground truth on the large L-Town network. The 2-layer NNConv model is the fastest, since it is a simple MPNN and hence scales linearly with the number of edges in the graph. The performance of MeGA-MP, on the other hand, additionally depends on the flow field and edge lengths, as detailed in Section 3.3.

## 5.2 EXPERIMENT 2: ADVECTION ON A 1D EUCLIDEAN DOMAIN

**Domain and Dataset** We now compare the performance of our method to that of other numerical solvers on a one-dimensional domain $\Omega = [0, l]$ of length $l = 100$. We aim to predict the solution $c(t, z)$ of a synthetic BVP at equidistant grid points $t$ and $z$ of the time and space interval $[0, 100]$ and $[0, 100]$ for different temporal and spatial discretizations $dz$ and $dt$ from $\{ \frac{100}{1000}, \frac{100}{316}, \frac{100}{100}, \frac{100}{32}, \frac{100}{10} \}$, respectively. The problem is described in detail in Appendix D.2. It can be considered as a problem where we apply an injection at the boundary $z = 0$ over time $t \in [0, 100]$.

In light of the discussion on IVPs and efficiency (remark 3.5), we discuss a synthetic IVP with boundary values in Appendix D.2.

**Baselines and Results** In Figure 4, the difference between the analytical solution of the problem, which serves as ground truth, and the MeGA-MP solution are visualized at different discretizations and on the whole resulting grid. A visualization of both individual solutions can be found in Appendix D.2. Also, in Table 2, we compare our method to other numerical solvers (Runge-Kutta-4 and a semi-Lagragian scheme) at different discretizations. For details regarding the baselines, we also refer to Appendix D.2. We find that MeGA-MP approximates the ground truth well and provides competitive results as compared to other solvers, as also reflected in Figure 20 in Appendix D.2.

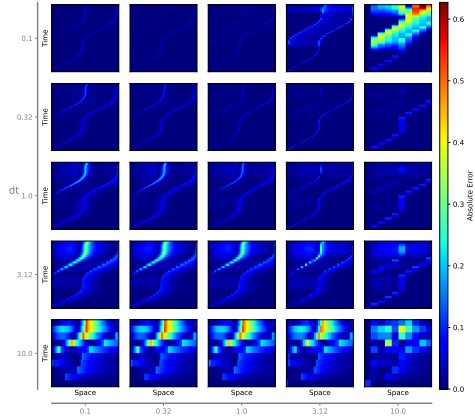

Figure 4: Difference between the analytical and the MeGA-MP solution of the BVP (34) for different combinations of spatial (columns) and temporal resolution (rows).

## 6 DISCUSSION

The comparison to other PDE solvers (Experiment 2) shows the saliency of our method, as our solution is comparable to that of other numerical solvers. Accuracy correlates with temporal resolution, since our method interpolates the solution temporally, which is more accurate for denser-sampled functions. This is in contrast to the baseline semi-Lagrangian solver, which interpolates spatially and thus benefits from higher spatial resolution. RK4 outperforms our method for the smallest temporal step-size but becomes unstable for certain combinations of spatial and temporal resolutions.

Table 2: MAE in-between analytical ground truth and solver solution over all temporal and spatial grid points. Find the full version of this table and the runtime analysis in Appendix D.2.

| $dt$ | $dx$ | semi- Lagr. | RK4 | MeGA-MP |
|------|------|-------------|-----|---------|
| 0.1 | 0.10 | **0.0013**$_{\pm 0.0050}$ | – | 0.0025$_{\pm 0.0080}$ |
| | 1.00 | 0.0106$_{\pm 0.0226}$ | 0.0032$_{\pm 0.0073}$ | **0.0027**$_{\pm 0.0082}$ |
| | 10.00 | 0.0461$_{\pm 0.0802}$ | **0.0311**$_{\pm 0.0425}$ | 0.1099$_{\pm 0.1435}$ |
| 1.0 | 0.10 | 0.0117$_{\pm 0.0175}$ | – | 0.0161$_{\pm 0.0290}$ |
| | 1.00 | 0.0116$_{\pm 0.0159}$ | **0.0106**$_{\pm 0.0137}$ | 0.0120$_{\pm 0.0199}$ |
| | 10.00 | 0.0433$_{\pm 0.0767}$ | 0.0276$_{\pm 0.0397}$ | **0.0106**$_{\pm 0.0198}$ |
| 10.0 | 0.10 | 0.1106$_{\pm 0.1265}$ | – | **0.0876**$_{\pm 0.1026}$ |
| | 1.00 | 0.1080$_{\pm 0.1223}$ | – | **0.0856**$_{\pm 0.1005}$ |
| | 10.00 | 0.0894$_{\pm 0.1091}$ | 0.1332$_{\pm 0.1345}$ | **0.0656**$_{\pm 0.0788}$ |

The application to a more complex dynamical system on a metric graph (Experiment 1) shows the utility of our method in such applications. Our method clearly outperforms the baselines and our result matches that of the numerical simulator (EPANET) well. Moreover, applying a model trained on the Hanoi WDS to the WDS L-Town shows that our method generalizes to novel graph topologies with the same governing equations underlying the dynamics. This is possible because the learnable component does not integrate information spatially, but is a point-wise operator.

**Limitations & Future Work** Our method is specifically designed for linear advection, it is naturally limited to PDEs where linear advection is the only dominant spatial dynamic. Future work can build upon the foundation laid out in this work by: 1) Extending our method to more complex spatial dynamics such as semi- and non-linear advection, or advection-diffusion via additional learnable operators. 2) Explore inverse problems of finding parameters of PDEs from given data (such as flow fields or reaction coefficients). 3) Experiment with more complex reaction terms. The latter would make this method a true surrogate model for the simulator (EPANET) for WDSs.

ETHICS STATEMENT

This work focuses on developing a machine learning method for physical modeling. It does not involve human subjects, personal data, or sensitive information. The data used in this work are self-generated, ensuring no privacy or ethical concerns. The proposed methods are intended to improve modeling in scientific and engineering contexts. We acknowledge the potential broader impacts of physics-informed artificial intelligence (AI) including misuse and emphasize that responsible development and validation are essential.

REPRODUCIBILITY STATEMENT

We are committed to transparency and reproducibility and will make the code publicly available. With it, all datasets can be generated and the experimental results can be reproduced. Model architectures and hyperparameters, as well as experimental setup are documented in this paper and also included in the released code.

LLM USAGE STATEMENT

We used ChatGPT for retrieval and discovery, such as finding related work. However, all related work cited in this paper was read and verified by the authors. ChatGPT was also used to assist with editing and proofreading sections of the manuscript with a focus on improving clarity and readability. Additionally, ChatGPT and ClaudeAI were used to generate boilerplate code that has been checked for correctness by the authors. The scientific content, equations, experimental design, results, and code implementing the methods presented in this work were not generated by large language models (LLMs).

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

ORGANIZATION OF THE SUPPLEMENTARY MATERIAL

## A   DETAILS ON THE BACKGROUND

**Message Passing**   Message generation describes how a feature vector at one node is transformed as it is sent to a neighbor. Message aggregation and node update describe how multiple messages arriving at a node are accumulated into a single feature vector and update the features at the target node. A general formulation of this method is given by

$$\mathbf{x}_v(t+1) = \theta \left( \sum_{u \in \mathcal{N}(v)} \phi\big(\mathbf{x}_u(t), \mathbf{x}_v(t), \mathbf{q}_{uv}(t)\big), \ \mathbf{x}_v(t) \right), \tag{13}$$

where $\phi$ is the message generation function, $\sum$ is a permutation-invariant aggregation function (e.g. sum or mean), $\theta$ is a function that updates the features $\mathbf{x}_v$ of node $v \in V$, and $\mathbf{q}_{uv}$ are optional edge features of the edge $e_{uv} \in E$ from node $u \in \mathcal{N}(v)$ to node $v \in V$. Moreover, $t$ denotes the message-passing iteration step.

**Advection on Metric Graphs**   Equation (4) holds because for each time $t \in \mathbb{R}$, the incompressible flow $\nu_{uv}(t, \cdot) : \Omega_{uv} \to \mathbb{R}$ defines a one-dimensional vector field along the interval $\Omega_{uv} = (0, l_{uv})$. By definition of incompressibility, $\mathrm{div}(\nu_{uv}(t, \cdot)) = \frac{\partial \nu_{uv}(t, \cdot)}{\partial z} = 0$ holds, or equivalently, the vector field $\nu_{uv}(t, \cdot)$ is constant on $\Omega_{uv} = (0, l_{uv})$, modeling the edge $e_{uv} \in E$.

Analogously, the vector field $\nu_{vu}(t, \cdot)$ is constant on $\Omega_{vu} = (0, l_{vu}) = (0, l_{uv})$, modeling the contrasting edge $e_{vu} \in E$. Since both constant vector fields $\nu_{uv}(t, \cdot)$ and $\nu_{vu}(t, \cdot)$ model the same physical phenomenon, namely the constant, actually observable *physical* flow field between the nodes $v \in V$ and $u \in \mathcal{N}(v)$, but along contrasting directions (the beginning of the edge $e_{uv}$ is the end of the edge $e_{vu}$ and vice versa), the orientation of $\nu_{vu}(t, \cdot)$ must be the opposite of the one of $\nu_{uv}(t, \cdot)$. Since they both correspond to the same, constant physical flow field, $\nu_{vu}(t, \cdot) = -\nu_{uv}(t, \cdot)$ must hold.

Together with the convention of signs (cf. Section 2), this allows to formally define the inflow neighbors from $v \in V$ at tie $t \in \mathbb{R}$, i.e., neighbors where there is a flow from $u \in \mathcal{N}(v)$ *in*to $v \in V$ at time $t \in \mathbb{R}$, as

$$\mathcal{N}_-(v, t) := \{u \in \mathcal{N}(v) \mid \mathrm{sgn}(\nu_{vu}(t) = -1\} = \{u \in \mathcal{N}(v) \mid \mathrm{sgn}(\nu_{uv}(t) = 1\}. \tag{14}$$

While incompressibility of a single flow field $\nu_{uv}(t, \cdot)$ implies the independence of the spatial variable $z \in (0, l_{uv})$, the incompressibility of the overall flow field $\boldsymbol{\nu}$ translates to the fact that the sum of flow rates $q_{uv}(t) = a_{uv}\nu_{uv}(t)$ into a node $v \in V$ is equal to the sum of the flow rates $q_{vu}(t) = a_{vu}\nu_{vu}(t)$ out of $v$:

$$\sum_{u \in \mathcal{N}_-(v, t)} q_{uv}(t) = \sum_{u \in \mathcal{N}_+(v, t)} q_{vu}(t).$$

**The Challenges of Advection-dominated Dynamics on Metric Graphs**   Advection-dominated or, more generally, hyperbolic dynamical systems on metric graphs, entail the challenge of a time-varying PDE-parameter and time-discretization dependency as well as an anti-symmetric and spatio-temporal dependency structure. We want visualize this using the simple example of an advection metric path graph, that is, a sequence of nodes $(v_0, v_1, ..., v_n)$ with edges $e_{j(j+1)}$ in-between two consecutive nodes $v_j$ and $v_{j+1}$. We assume that the corresponding edge spaces $\Omega_{j(j+1)}$ are of equal size $l_{j(j+1)} = 1m$ and have a constant flow field of $\nu_{j(j+1)} = 2m/s$ for all $e_{j(j+1)} \in E$. If we access information at each of the nodes $v_j$ at discrete time points $t_0, ..., t_i, t_{i+1}, ...$ with temporal discretization $dt = t_{i+1} - t_i = 60s$, information travels $dt \cdot \nu_{j(j+1)} = 60s \cdot 2m/s = 120m$ per time step $dt$, which corresponds to $\frac{dt \cdot \nu_{j(j+1)}}{l_{j(j+1)}} = 120$ hops in the path graph. By this, the information $c_v(t_{i+1})$ of a node $v$ at time $t_{i+1}$ depends on the information $c_u(t_i)$ at time $t_{i+1}$ and a node $u$ which lies 120 hops upstream of $v$. If such path does not exist, the path will be shorter and will originate at boundary nodes. The above example provides a sketch how the spatial dependency structure depends on the lengths, the flow field and the temporal discretization.

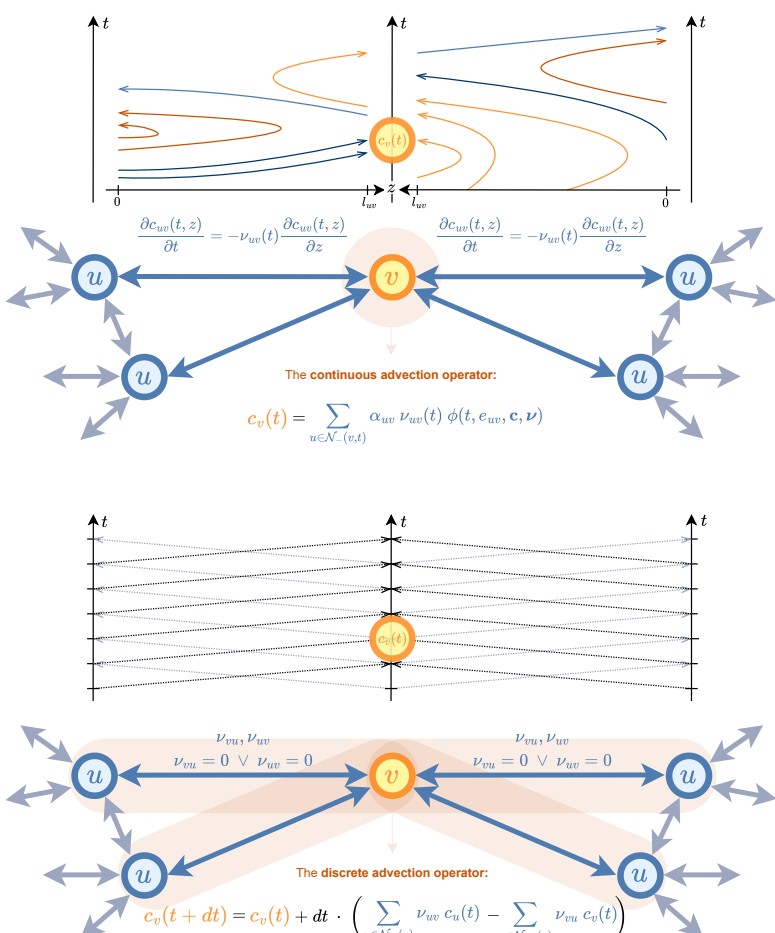

Figure 5: Advection on a metric graph (top) vs. on a non-metric graph (bottom): A metric graph is a system of one-dimensional subspaces $\Omega_{uv} = (0, l_{uv})$, corresponding to edges $e_{uv} \in E$, coupled at nodes $v, u \in V$. Each edge is associated with a one-dimensional PDE that defines the dynamics of the magnitude $c_{uv}(t, z)$ over time $t \in \mathbb{R}$ and along the subspace $\Omega_{uv} = (0, l_{uv})$, parametrized by a known function $\nu_{uv}(\cdot)$. In case of advection, the known functions $\boldsymbol{\nu} = (\nu_{uv})_{e_{uv} \in E}$ correspond to the time-dependent flow field of the metric graph. The coupling conditions $c_v(t)$ at nodes can be expressed as a weighted aggregation of features $c_{uv}(t, l_{uv}) = \phi(t, e_{uv}, \mathbf{c}, \boldsymbol{\nu})$ at the boundary $z = l_{uv}$ of incoming edges $e_{uv}$. Their value can be traced back to either features $c_u(t - \delta t_{uv})$ of neighboring nodes $u \in \mathcal{N}_-(v, t)$ (dark blue lines) or features $c_v(t - \delta t_{uv})$ of the node itself (light orange lines) and at earlier, over the neighboring nodes varying times $t - \delta t_{uv}$.

In contrast, a (conventional, non-metric) graph can be considered as a discretization of the spatial Euclidean space $\Omega = \mathbb{R}^n$. Here, edges $e_{uv} \in E$ preserve proximity in between nodes $v, u \in V$, corresponding to coordinates in the Euclidean domain. The overall space $\Omega = \mathbb{R}^n$ is associated with a multi-dimensional PDE that defines the dynamics of the magnitude $c(t, v) := c_v(t)$ over time $t \in \mathbb{R}$ and space $v \in V \subset \Omega$, parametrized by a function $\boldsymbol{\nu}(\cdot)$. In case of advection, the discretized advection-operator of the underlying multi-dimensional and continuous advection-PDE $\frac{\partial c_v(t)}{\partial t} = -\boldsymbol{\nu} \cdot \nabla c_v(t)$ defines a coupling condition $c_v(t + dt)$ that aggregates the net inflow from neighboring nodes $u \in \mathcal{N}_{\text{in}}(v)$ minus the outflow to neighboring nodes $u \in \mathcal{N}_{\text{out}}(v)$ and at a fixed time $t \in \mathbb{R}$. In practice, the discretized version $\boldsymbol{\nu} = (\nu_{uv})_{e_{uv} \in E}$ of the continuous flow field is a learned parameter of a GNN.

For more complex structured graphs and more complex configurations of PDE-parameters, the spatial dependency structure will be a tree that extends from the target (root) node $v$ at time $t$ to upstream nodes (leafs) (node $u$ in the example above) (also see Appendix B.2). The dependency tree will ex-

tend further in directions with high flow rate and less in direction where the flow rate is lower. This results in two major challenges for standard MPNNs: Firstly, the extent of the receptive field can vary drastically across the space of the metric graph and over time due to varying PDE-parameters, making it impossible to find a single suitable number of message passing layers. Secondly, the receptive field can be highly asymmetric. These two properties require highly selective and anisotropic message passing that depends on the structure of the metric graph, flow field and the temporal discretization.

While there already exist work that discusses graphs and GNNs specifically taylored to advection problems (Chapman, 2015; Eliasof et al., 2024), the underlying graph structure corresponds to a discretization of the Euclidean space; a paradigm different from the one on metric graphs (cf. Section 2). We highlight this difference in Figure 5.

# B  DETAILS ON THE METHOD

## B.1  FINAL ALGORITHM: MEGA-MP

**Interpolation**  Different interpolation schemes can be applied for time warping, depending on the use case. A version of linear interpolation of a value $c_u(t - \delta t_{uv})$ can be modeled as

$$c_u(t - \delta t_{uv}) \approx a(t_j) \cdot c_u(t_j) + a(t_{j+1}) \cdot c_u(t_{j+1}) \tag{15}$$

with $a(t_{j+1}) := a_j(t - \delta t_{uv}), a(t_j) := 1 - a_j(t - \delta t_{uv})$ and $a_j(t - \delta t_{uv}) = \frac{t - \delta t_{uv} - t_j}{t_{j+1} - t_j}$ for the $j \in I$ such that $t - \delta t_{uv} \in [t_j, t_{j+1})$ holds.

**Model Iterations**  Based on Equation (15), we formally define $\varphi(t, e_{uv}, \hat{\mathbf{c}}^{(k)}, \boldsymbol{\nu})$ from Equation (12) as

$$\varphi(t, e_{uv}, \hat{\mathbf{c}}^{(k)}, \boldsymbol{\nu}) := \begin{cases} a(t_j) \cdot \hat{c}_u^{(k)}(t_j) + a(t_{j+1}) \cdot \hat{c}_u^{(k)}(t_{j+1}) & \text{if } z(\delta t_{uv}) \in \{-l_{uv}, l_{uv}\} \\ a(t_j) \cdot \hat{c}_v^{(k)}(t_j) + a(t_{j+1}) \cdot \hat{c}_v^{(k)}(t_{j+1}) & \text{if } z(\delta t_{uv}) = 0 \end{cases} \tag{16}$$

for the $j \in I$ for which $t - \delta t_{uv} \in [t_j, t_{j+1})$ holds. In practice, we precompute the transport times $\delta t_{uv}$ and implement $\varphi$ as an interpolation operator $\mathcal{I}[c_w^{(k)}](t - \delta t_{uv})$ at the correct node $w \in \{u, v\}$ that does not depend on the flow field $\boldsymbol{\nu}$ anymore (cf. Algorithm 2).

## B.2  THEORETICAL GUARANTEES ON MEGA-MP

Not only to be able formalize the full version of Theorem 3.4, but also to be able to prove Theorem 3.3 and 3.4, we need to introduce some further notation. Before we do so, we provide some theory on a special graph structure, trees, which motivates the notation and builds the backbone of our theory. The following sections can also help to get a deeper understanding of our iterative algorithm, Equation (12), whose intuition was already describe in Section (3.3).

### B.2.1  PRELIMINARIES: TREES WITH CHILDREN-AGGREGATION SCHEME

In this section, we first investigate on the intuition behind the message passing scheme (11) (and (12)), which can also be written as

$$c_v(t) = \sum_{u \in \mathcal{N}_-(v,t)} w_{uv}(t) \cdot \phi(t, e_{uv}, \mathbf{c}, \boldsymbol{\nu}).$$

For simplicity, let us assume that the message function $\phi$ outputs $\phi(t, e_{uv}, \mathbf{c}, \boldsymbol{\nu}) = c_u(t - \delta t_{uv})$ (cf. Eq. (9)), that is, that at time $t$, we observe a pass-through along the edge $e_{uv}$ and not a an inverse self-loop (cf. Lemma C.15 below). In this case, Equation (11) can – with (1) a change in notation and (2) using the more formal notation from Equation (31) – be written as

$$c_v(t) \stackrel{(1)}{=} \sum_{v_1 \in \mathcal{N}_-(v,t)} w_{v_1 v}(t) \cdot c_{v_1}(t - \delta t_{v_1 v}) \stackrel{(2)}{=} \sum_{v_1 \in \mathcal{N}_-(v,t)} w_{v_1 v}(t) \cdot c_{v_1}(\underbrace{t - s(t, e_{v_1 v}, \boldsymbol{\nu})}_{=:t_1}).$$

For each inflow neighbor $v_1 \in \mathcal{N}_-(v, t)$ of $v$ at time $t$, we can again apply Equation (11), yielding

$$c_v(t) = \sum_{v_1 \in \mathcal{N}_-(v,t)} \sum_{v_2 \in \mathcal{N}_-(v_1,t_1)} w_{v_1 v}(t) \cdot w_{v_2 v_1}(t_1) \cdot c_{v_2}(\underbrace{t_1 - \underline{s}(t_1, e_{v_2 v_1}, \boldsymbol{\nu})}_{=:t_2})).$$

Intuitively, we can repeat this process as often as we want until a neighborhood $\mathcal{N}_-(v_l, t_l)$ is empty, which by definition is the case if $v_l \in V_{\mathfrak{b}}$ is a source node. What we create along the way are paths $p = ((v_0, t_0) := (v, t), (v_1, t_1), (v_2, t_2), ...)$ of both locations $v_l \in V$ and times $t_l \in \mathbb{R}$ for indices $l = 1, 2, ...$ along which we trace back information that contributes to the information $c_v(t)$.[1] We call such paths *inflow paths*. While the graph $\mathcal{G}$ is a symmetric directed graph (cf. Section 2), the set of all such inflow paths $\mathcal{P}(v, t)$ defines a tree structure[2], which we construct formally and iteratively from the root node $(v_0, t_0) = (v, t)$ to leaf nodes in the next Section B.2.2. Before, we provide a general result on trees with children-aggregation aggregation schemes:

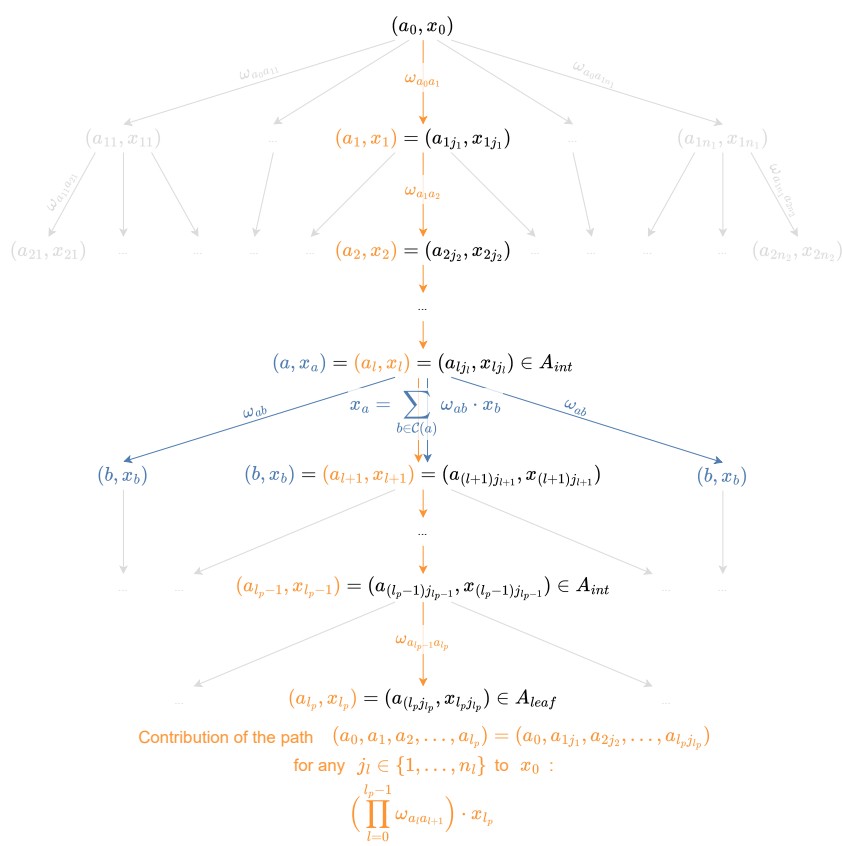

Figure 6: A visualization of the concepts from Theorem B.1.

**Theorem B.1** (Trees with children-aggregation scheme). *Let $\mathcal{T} = (A, E)$ be a finite tree with root node $a_0 \in A$, **int**ernal nodes $A_{int}$ and **leaf** nodes $A_{leaf}$ (that is, $A = \{a_0\} \sqcup A_{int} \sqcup A_{leaf}$ holds). If each node $a \in A$ is associated with a magnitude $x_a \in \mathbb{R}$ that satisfies*

$$x_a = \sum_{b \in \mathcal{C}(a)} \omega_{ab} \cdot x_b \text{ for all } a \in \{a_0\} \sqcup A_{int},$$

*where the set $\mathcal{C}(a) \subset A$ consists of the children of $a \in A$ and $w_{ab}$ is a weight of the edge $e_{ab} \in E$ that depends on both the parent $a \in A$ and its child $b \in \mathcal{C}(a)$, then the magnitude $x_0 := x_{a_0}$ of the*

---

[1]Note that the indices $l = 1, 2, ...$ do *not* necessarily correspond to the discrete times indices $I$ in Section (2).

[2]Note that by definition, the transport time is positive (Theorem C.13). Therefore, $t > t_1 > t_2 > ...$ holds.

*root node $a_0 \in A$ is determined by the magnitudes $x_{l_p} := x_{a_{l_p}}$ of the leaf nodes $a_{l_p} \in A_{leaf}$ by*

$$x_0 = \sum_{p=(a_0,...,a_{l_p}) \in \mathcal{P}(a_0, A_{leaf})} \omega(a_0, ..., a_{l_p}) \cdot x_{l_p},$$

*where $\mathcal{P}(a_0, A_{leaf})$ is the set of paths connecting the root node $a_0$ to one of its leaf nodes $A_{leaf}$ and*

$$\omega(a_0, ..., a_{l_p}) := \prod_{l=0}^{l_p-1} \omega_{a_l a_{l+1}}$$

*is the product of weights along a(n) (arbitrary) path $p = (a_0, ..., a_{l_p}) \in \mathcal{P}$ with length $l_p \in \mathbb{N}$.*

A visualization of such a tree can be found in Figure 6.

### B.2.2 CONSTRUCTION OF INFLOW TREES

In this section, we first provide some intermediate results – Theorem B.3 and Theorem B.4 – that help proving Theorem 3.3 and 3.4. In order to prove the former, we construct suitable trees in the sense of Theorem B.1:

**Definition B.2** (Inflow tree and interpolation inflow tree). Let $\mathcal{G} = (V, E)$ be a metric graph as defined in Section 2. Let $v \in V$ and $t \in \mathbb{R}$ hold.

We define the inflow tree $\mathcal{T}(v, t)$ of $v$ at time $t$ using the notation from Theorem B.1 by iteratively defining the nodes and children per layer $l \in \mathbb{N}_0$ of the tree until a stopping criteria is fulfilled:

- $l = 0$-th layer: We choose the root node $a_0 = (v_0, t_0) := (v, t)$.

- $l+1$-th layer: Given a parent node $a_l = (u_l, v_l, t_l)$[3] in the $l$-th layer, we choose its children $a_{l+1} = (u_{l+1}, v_{l+1}, t_{l+1}) \in \mathcal{C}(a_l)$ as each combination of

  - an inflow neighbor $u_{l+1} \in \mathcal{N}_-(v_l, t_l)$ of $v_l$ at time $t_l$ and
  - the corresponding shifted time $t_{l+1} := t_l - \mathfrak{s}(t_l, e_{u_{l+1}v_l}, \boldsymbol{\nu})$ obtained through the time shift applied to $t_l$ using the (positive) transport time $\mathfrak{s}(t_l, e_{u_{l+1}v_l}, \boldsymbol{\nu}) > 0$ (cf. Eq. (31)) along the edge $e_{u_{l+1}v_l} \in E$ and
  - the corresponding source node[4]

$$v_{l+1} := \begin{cases} u_{l+1} & \text{if } \mathfrak{z}\big(\mathfrak{s}(t_l, e_{u_{l+1}v_l}, \boldsymbol{\nu})\big) \neq 0, \\ v_l & \text{if } \mathfrak{z}\big(\mathfrak{s}(t_l, e_{u_{l+1}v_l}, \boldsymbol{\nu})\big) = 0. \end{cases} \tag{17}$$

- We stop the construction after an $a_{l+1} = (u_{l+1}, v_{l+1}, t_{l+1})$ if a stopping criteria is fulfilled, which for now can be any stopping criteria, or until naturally, the neighborhood $\mathcal{N}_-(v_l, t_l)$ is empty, which is by definition the case if $v_l \in V_\mathrm{b}$ is a source node.

- We denote the set of paths connecting the root node $a_0 = (v_0, t_0) = (v, v, t)$ to one of its leaf nodes $A_\mathrm{leaf}$ by $\mathcal{P}(v, t)$, which is an abbreviation for $\mathcal{P}(a_0, A_\mathrm{leaf})$.

We similarly define the interpolation inflow tree $\hat{\mathcal{T}}(v, t)$ of $v$ at time $t$ using the notation from Theorem B.1 by iteratively defining the nodes and children per layer $l \in \mathbb{N}_0$ of the tree until a stopping criteria is fulfilled:

- $l = 0$-th layer: We choose the root node $a_0 = (v_0, t_0) := (v, t)$.

- $l+1$-th layer: Given a parent node $a_l = (u_l, v_l, t_l)$ in the $l$-th layer, we choose its children $a_{l+1} = (u_{l+1}, v_{l+1}, t_{l+1}) \in \mathcal{C}(a_l)$ as each combination of

---

[3]Note that the definition of $a_{l+1}$ does not depend on $u_l$, which is why this step is well-defined for $l = 0$ and $l + 1 = 1$, too.

[4]In the former case, we observe a pass-through along the edge $e_{u_{l+1}v_l}$, that is, information flows from $v_{l+1} = u_{l+1}$ to $v_l$. In the latter case, we observe an inverse self-loop along the edge $e_{u_{l+1}v_l}$, that is, information flows from $v_{l+1} = v_l$ to $v_l$. For more details, see Section C.

- an inflow neighbor $u_{l+1} \in \mathcal{N}_-(v_l, t_l)$ of $v_l$ at time $t_l$ and
- an corresponding interpolation time $t_{l+1} \in \{t_j, t_{j+1}\}$ for the $j \in I$ for which $t_l - \mathfrak{s}(t_l, e_{u_{l+1}v_l}, \boldsymbol{\nu}) \in [t_j, t_{j+1})$ holds (cf. Eq. (16))[5] and
- the corresponding source node $v_{l+1}$ analogously to Equation (17).[6]

- We stop the construction after an $a_{l+1} = (u_{l+1}, v_{l+1}, t_{l+1})$ for which $t_{l+1} \in T_{\text{hi}} = \{t_{i-k_{\text{hi}}+1}, ..., t_i\}$ or $t_{l+1} \in T_{\text{pr}} = \{t_{i+1}, ..., t_{i+k_{\text{pr}}}\}$ and $v_{l+1} \in V_{\text{b}}$ holds.[7]

- We denote the set of paths connecting the root node $a_0 = (v_0, t_0) = (v, t)$ to one of its leaf nodes $A_{\text{leaf}}$ by $\hat{\mathcal{P}}(v, t)$, which is an abbreviation for $\hat{\mathcal{P}}(a_0, A_{\text{leaf}})$.

We now leverage these trees together with Theorem B.1 to prove the following two intermediate results, which allow to reconstruct the magnitudes $c_v(t)$ and $\hat{c}_v^{(k)}(t)$ of interest along inflow paths:

**Theorem B.3** (Closed form of advection message passing). *Let $v \in V$ and $t \in \mathbb{R}$ hold. Let $\mathcal{T}(v, t)$ be the inflow tree of $v$ at time $t$ as defined in Definition B.2. If $c_v(t)$ is defined by Equation (11), then*

$$c_v(t) = \sum_{p=((v_0,t_0)=(v,t),(u_1,v_1,t_1)...,(u_{l_p},v_{l_p},t_{l_p}))\in\mathcal{P}(v,t)} \left(\prod_{l=0}^{l_p-1} w_{u_{l+1}v_l}(t_l)\right) \cdot c_{v_{l_p}}(t_{l_p}) \qquad (18)$$

*with weights $w_{u_{l+1}v_l}(t_l)$ according to Equation (10) holds.*

**Theorem B.4** (Closed form of advection message passing iterations). *Let $v \in V$ and $t \in T_{hi} \cup T_{pr}$ hold. Let $\hat{\mathcal{T}}(v, t)$ be the interpolation inflow tree of $v$ at time $t$ as defined in Definition B.2. Let $k \geq \max_{p \in \hat{\mathcal{P}}(v,t)} l_p$ hold. If $\hat{c}_v^{(k)}(t)$ is defined by Equation (12), then*

$$\hat{c}_v^{(k)}(t) = \sum_{p=((v_0,t_0)=(v,t),(u_1,v_1,t_1)...,(u_{l_p},v_{l_p},t_{l_p}))\in\hat{\mathcal{P}}(v,t)} \left(\prod_{l=0}^{l_p-1} w_{u_{l+1}v_l}(t_l) \cdot a(t_{l+1})\right) \cdot \hat{c}_{v_{l_p}}^{(k-l_p)}(t_{l_p})$$

$$(19)$$

*with weights $w_{u_{l+1}v_l}(t_l)$ and $a(t_{l+1})$ according to Equation (10) and (15), respectively, holds.*

Theorem B.4 already allows a proof of Theorem 3.3, which is given together with the other proofs in Appendix E.

For the proof of Theorem 3.4, we need to make some further assumptions: While Equation (18) and (19) are already of similar structure, the number of paths $\hat{p} \in \hat{\mathcal{P}}(v, t)$ grows exponentially by a factor of $2^l$, where $l$ is the depth of the tree $T(v, t)$ and $\hat{\mathcal{T}}(v, t)$, as compared to the number of paths $p \in P(v, t)$. This is due the fact that per time that we observe in the inflow tree $T(v, t)$, we observe two interpolation times in the interpolation inflow tree $\hat{\mathcal{T}}(v, t)$ (cf. Definition B.2). To make each summand in Equation (19) comparable to a summand in Equation (18), we need to make the following assumptions:

**Assumption B.5** (Identifiable paths). Let $v \in V$ and $t \in \mathbb{R}$ hold. Let $T(v, t)$ and $\hat{\mathcal{T}}(v, t)$ be the inflow tree and the interpolation inflow tree from Definition B.2, respectively. We assume that each path $\hat{p} = ((\hat{v}_0, \hat{t}_0) = (v, t), (\hat{u}_1, \hat{v}_1, \hat{t}_1)..., (\hat{u}_{l_p}, \hat{v}_{l_p}, \hat{t}_{l_p})) \in \hat{\mathcal{P}}(v, t)$ is uniquely identifiable with a path $p = ((v_0, t_0) = (v, t), (u_1, v_1, t_1)..., (u_{l_p}, v_{l_p}, t_{l_p})) \in \mathcal{P}(v, t)$ in the sense that $\hat{u}_l = u_l$ and $\hat{v}_l = v_l$ holds for all $l = 1, ..., l_p$.

---

[5]Similar to the construction to the inflow tree $\mathcal{T}(v, t)$, $t_l - \mathfrak{s}(t_l, e_{u_{l+1}v_l}, \boldsymbol{\nu})$ is the – possibly not in the observable, discrete subdomain of $\mathbb{R}$ lying – shifted time obtained through the time shift applied to $t_l$ using the (positive) transport time $\mathfrak{s}(t_l, e_{u_{l+1}v_l}, \boldsymbol{\nu}) > 0$ (cf. Eq. (31)) along the edge $e_{u_{l+1}v_l} \in E$. Note that for $l \in \mathbb{N}$, $t_l$ already corresponds to one of the discrete time points that are known, that is, there exists an $j \in I$ such that $t_l = t_j$ holds.

[6]Note that the definition of $v_{l+1}$ does not depend on $t_{l+1}$, which is why the corresponding source node $v_{l+1}$ of both $t_{l+1} = t_j$ and $t_{l+1} = t_{j+1}$ is the same.

[7]Note that similar to the construction to the inflow tree $\mathcal{T}(v, t)$, we can also chose any other suitable stopping criteria. We chose the one that we will need later. Also note that with this stopping criterium, if $t \in T_{\text{hi}}$ or $t \in T_{\text{pr}}$ and $v \in V_{\text{b}}$ holds, the creation of the tree $\hat{\mathcal{T}}(v, t)$ stops immediately and does only consist of the "path" $p = ((v_0, t_0) = (v, t))$ of length zero.

**Assumption B.6** (Equal time for jumping into history). Let $v \in V$ and $t \in \mathbb{R}$ hold. Let $\hat{\mathcal{T}}(v,t)$ be the interpolation inflow tree from Definition B.2. In the definition of each $t_{l+1} \in \{t_j, t_{j+1}\}$, we assume that $t_j, t_{j+1} \in T_{\mathrm{pr}}$ or $t_j, t_{j+1} \in T_{\mathrm{hi}}$ holds.

Assumption B.5 is satisfied as long as the inflow neighborhoods $\mathcal{N}_-(v_l, t_l)$, from which we choose an inflow neighbor $u_{l+1}$ when creating a path in $\mathcal{T}(v,t)$, are equal to the corresponding neighborhoods $\mathcal{N}_-(\hat{v}_l, \hat{t}_l)$, from which we choose an inflow neighbor $\hat{u}_{l+1}$ when creating a path in $\hat{\mathcal{T}}(v,t)$. This is guaranteed if the underlying flow field $\boldsymbol{\nu}$ that defines these neighborhoods does not change its orientation during the time intervals $[t_j, t_{j+1}]$ for which $t_l \in [t_j, t_{j+1})$ holds. If the assumption holds, it allows the following definition.

**Definition B.7** (Joint paths). Let $v \in V$ and $t \in \mathbb{R}$ hold. Let $T(v,t)$ and $\hat{\mathcal{T}}(v,t)$ be the inflow tree and the interpolation inflow tree from Definition B.2, respectively. Let assumption B.5 hold. If the path $\hat{p} = ((\hat{v}_0, \hat{t}_0) = (v,t), (\hat{u}_1, \hat{v}_1, \hat{t}_1)..., (\hat{u}_{l_p}, \hat{v}_{l_p}, \hat{t}_{l_p})) \in \hat{\mathcal{P}}(v,t)$ is uniquely identifiable with the path $p = ((v_0, t_0) = (v,t), (u_1, v_1, t_1)..., (u_{l_p}, v_{l_p}, t_{l_p})) \in \mathcal{P}(v,t)$, we define the joint path of $\hat{p}$ by

$$\tilde{p} = ((v_0, t_0) = (v,t), (u_1, v_1, t_1, \hat{t}_1), ..., (u_{l_p}, v_{l_p}, t_{l_p}, \hat{t}_{l_p})). \tag{20}$$

We can still consider $\tilde{p}$ as an element of $\hat{\mathcal{T}}(v,t)$, since we only added information that are unused, in, e.g., Theorem B.4.

Moreover, Assumption B.6 guarantees that there exist two such paths that are exactly equal up to the last $\hat{t}_{l_p} \in \{t_j, t_{j+1}\}$ for a $j \in I$ in the sense of Definition B.2. Finally, using the property that per such two paths ending in such two left and right interpolation bounds, the factors $a(t_j)$ and $a(t_{j+1})$ add up to one, we can canonically extend the sum in Equation (18) to the sum in Equation (19) using the joint paths. This allows to prove Theorem 3.4, which in its full version is phrased as follows:

**Theorem B.8** (Interpolation error (full version)). *Let $v \in V$ and $t \in T_{hi} \cup T_{pr}$ hold. Let $\mathcal{T}(v,t)$ and $\hat{\mathcal{T}}(v,t)$ be the inflow tree and the interpolation inflow tree of $v$ at time $t$ as defined in Definition B.2, respectively. Let Assumption B.5 and B.6 hold. Then the prediction error in-between $c_v(t)$ and $\hat{c}_v(t)$ is given by*

$$|c_v(t) - \hat{c}_v(t)|$$

$$\leq \sum_{\tilde{p} \in \hat{\mathcal{P}}(v,t)} a(\hat{t}_1, ..., \hat{t}_{l_p}) \cdot \left( \underbrace{|c_{v_{l_p}}(t_{l_p}) - c_{v_{l_p}}(\hat{t}_{l_p})|}_{\text{Temporal interpolation error}} + |c_{v_{l_p}}(t_{l_p})| \cdot \underbrace{|w(t_0, ..., t_{l_p-1}) - w(\hat{t}_0, ..., \hat{t}_{l_p-1})|}_{\text{Accumulated weighting error}} \right)$$

*with joint paths $\tilde{p} = ((v_0, t_0) = (v,t), (u_1, v_1, t_1, \hat{t}_1), ..., (u_{l_p}, v_{l_p}, t_{l_p}, \hat{t}_{l_p})) \in \hat{\mathcal{P}}(v,t)$ according to Definition B.7,*

$$a(\hat{t}_1, ..., \hat{t}_{l_p}) := \prod_{l=0}^{l_p-1} a(\hat{t}_{l+1}) \quad and \quad w(\hat{t}_0, ..., \hat{t}_{l_p-1}) := \prod_{l=0}^{l_p-1} w_{u_{l+1}v_l}(\hat{t}_l)$$

*with weights $w_{u_{l+1}v_l}(t_l)$ and $a(t_{l+1})$ according to Equation (10) and (15), respectively. If the temporal interpolation error is zero, we obtain $|c_v(t) - \hat{c}_v(t)| = 0$.*

Theorem B.8 states that the error between the ground truth $c_v(t)$ and the prediction $\hat{c}_v(t)$ solely stems from the time interpolation: One the one hand, the temporal interpolation error is the error from observing a value $c_{v_{l_p}}(\hat{t}_{l_p})$ at a node $v_{l_p} \in V$ and at a discrete time $t_{l_p} \in T_{\mathrm{hi}}$, when actually, we needed to observe the value $c_{v_{l_p}}(t_{l_p})$ at an eventually non-dicrete time $t_{l_p} \in \mathbb{R}$. On the other hand, for exactly the same reason, the weights $w_{u_{l+1}v_l}(\hat{t}_l)$ that we observe at discrete times $t_l \in T_{\mathrm{hi}} \sqcup T_{\mathrm{pr}}$ might differ from the weights $w_{u_{l+1}v_l}(t_l)$ that we needed to observe at non-discrete times $t_l \in \mathbb{R}$ for all $l = 1, ..., l_p - 1$, yielding the accumulated weighted error. Both errors decrease with decreasing error in-between the times $t_l$ and $\hat{t}_l$, as well as slowly varying concentrations $\mathbf{c}$ and flow fields $\boldsymbol{\nu}$.

## B.3 RUNTIME COMPLEXITY OF MEGA-MP

---

**Algorithm 1** Preprocessing: Converting flows $\nu_{uv}(t)$ to transport times $\delta t_{uv}(t)$ (Definition 3.2). This algorithm is implemented in a parallelized/vectorized form for efficiency.

---

1: **Inputs:** Flow velocities $\nu_{uv} \in \mathbb{R}^T$ for edge $e_{uv} \in E$ for $T$ time steps ($k_{\text{hi}} + k_{\text{hi}}$); Edge length $L_{uv} \in \mathbb{R}$ for edge $e_{uv} \in E$.
2: $C \leftarrow \text{cumsum}(\nu_{uv}) \in \mathbb{R}^T$         ▷ cumulative distance traveled over time
3: $\Delta T^{pt} \in \mathbb{R}^T$ init. empty       ▷ array to store (inverse) pass-through transport times
4: $\Delta T^{sl} \in \mathbb{R}^T$ init. empty            ▷ array to store self-loop transport times
5:                           ▷ Compute (inverse) pass-through transport times
6: **for** $t = 0$ **to** $T - 1$ **do**          ▷ Trace characteristic curve starting at time $t$
7:     **for** $x = t$ **down to** $0$ **do**             ▷ Trace backward along time steps
8:         $d_{x,t} \leftarrow C_t - C_x$          ▷ distance traveled between $x$ and $t$
9:         **if** $d_{x,t} \geq L_{uv}$ **then**                 ▷ true for pass-throughs
10:            interpolate $\hat{x}$ between $C_x$ and $C_{x+1}$ if needed
11:            $\Delta T_t^{pt} \leftarrow t - \hat{x}$ **; break**
12:         **else if** $d_{x,t} \leq -L_{uv}$ **then**          ▷ true for inverse pass-throughs
13:            interpolate $\hat{x}$ between $C_x$ and $C_{x+1}$ if needed
14:            $\Delta T_t^{pt} \leftarrow \hat{x} - t$ **; break**
15:         **end if**
16:     **end for**
17: **end for**
18:                           ▷ Compute self-loop transport times
19: **for** $t = 1$ **to** $T - 1$ **do**          ▷ Trace characteristic curve starting at time $t$
20:     **for** $x = t$ **down to** $1$ **do**
21:         $d_1 \leftarrow C_{x-1} - C_t$; $d_2 \leftarrow C_x - C_t$     ▷ distance differences between consecutive steps
22:         **if** $\text{sign}(d_1) \neq \text{sign}(d_2)$ **then**         ▷ zero-crossing indicates self-loop
23:            interpolate zero crossing: $\hat{x} \leftarrow x - d_2/(d_2 - d_1)$
24:            $\Delta T_t^{sl} \leftarrow \text{sign}(d_2) \cdot (t - \hat{x})$ **; break**
25:         **end if**
26:     **end for**
27: **end for**
28: **Output:** $\Delta T^{pt} \in \mathbb{R}^T$; $\Delta T^{sl} \in \mathbb{R}^T$

---

**Algorithm 2** MeGA-MP: Metric Graph Advection Message Passing

---

1: **Inputs:** Graph structure $\mathcal{G} = (V, E)$; Initial node states / known history $x \in \mathbb{R}^{|V| \times k_{hi}}$; Transport times $\delta t_{uv}(t)$ for each edge $e_{uv} \in E$ and for each time step $t_i \in T$, where $T = k_{\text{hi}} + k_{\text{pr}}$.
2: $x^{(0)} \leftarrow \text{pad}(x, k_{\text{pr}})$            ▷ pad initial condition by prediction length $k_{\text{pr}}$
3: Apply Dirichlet boundary conditions to $x^{(0)}$ at boundary nodes
4: $x_{\text{out}} \leftarrow x^{(0)}$                   ▷ initialize output accumulator
5: **for** $i = 1$ **to** max_iter **do**
6:     **for** each node $v \in V$ and time $t$ **do**               ▷ Message-Passing
7:         $x_{v,t}^{(i)} \leftarrow \sum_{u \in \mathcal{N}(v)} w_{uv}(t) \mathcal{I}[x_u^{(i-1)}](t - \delta t_{uv}(t))$     ▷ cf. Eq. 10 and Sec. 3.3 and B.1
8:     **end for**
9:     Apply boundary conditions to $x^{(i)}$ at boundary indices
10:     $x_{\text{out}} \leftarrow x_{\text{out}} + x^{(i)}$
11: **end for**
12: **if** boundary values exist **then**
13:     Set $x_{\text{out}}$ at boundary nodes
14: **end if**
15: **Output:** $x_{\text{out}}$            ▷ final node states after advection message passing

---

## C   PHYSICAL DERIVATION OF ADVECTION MESSAGE PASSING

In this section, we present a full derivation of our message-passing algorithm MeGA-MP as introduced in Section 3, based on its underlying physical priors. The physical priors decompose into two main components, which we each relate to a main concept of message passing: In Section C.1, we derive the message-generation function $\phi$ as defined in Equation (9) based on the *information transfer* through *advective transport along edges*. For each node $v \in V$ and its neighbors $u \in \mathcal{N}(v)$, this refers to the change of the concentration $c_{uv} : T \times (0, l_{uv}) \to \mathbb{R}_{\geq 0}$ *along* or *within* the edge $e_{uv} \in E$ determined by the incompressable *flow field* $\nu_{uv} : T \to \mathbb{R}$ over time $T = \mathbb{R}$. Formally, this is defined by the PDE

$$\frac{\partial c_{uv}(t,z)}{\partial t} = -\nu_{uv}(t)\,\frac{\partial c_{uv}(t,z)}{\partial z} \tag{21}$$

as already introduced in Section 2, with boundary conditions that define the information transfer between nodes and edges: On the one hand, if there is a (physical) flow leaving a node and entering an edge, the concentration at the beginning of that edge is equal to the concentration at that node (*inflow boundary condition*). On the other hand, if there is a (physical) flow entering a node and leaving an edge, the concentration at the end of that edge contributes to the weighted aggregation at that node (*outflow boundary condition*). This is formally defined as follows:

**Definition C.1** (Inflow and outflow boundary conditions). Let $v \in V$, $u \in \mathcal{N}(v)$ and $t \in \mathbb{R}$ hold.

1. If $u \in \mathcal{N}_-(v,t)$, i.e., whenever $\mathrm{sgn}(\nu_{uv}(t)) = -\mathrm{sgn}(\nu_{vu}(t)) = +1$ holds[8], we set $c_{uv}(t,0) := c_u(t)$.

   That is, if there is a flow from $u \in \mathcal{N}_-(v,t)$ to $v$ at time $t$, the concentration $c_{uv}(t,0)$ at the beginning $z = 0$ of the edge $e_{uv} \in E$ is equal to the concentration $c_u(t)$ at the node $u$ out of which the flow flows *into that edge*, also called *inflow boundary condition*.

2. $c_v(t)$ is defined by the mixing at nodes as defined in Equation (22) below.

   That is, if there is a flow from $u \in \mathcal{N}_-(v,t)$ to $v$ at time $t$, the concentration $c_{uv}(t, l_{uv})$ at the end $z = l_{uv}$ of the edge $e_{uv} \in E$ is the contribution of $u$ to the concentration $c_v(t)$ in the mixing at the node $v$ into which the flow flows *out of that edge*, also called *outflow boundary conditions*.

3. We set $c_{vu}(t,z) := c_{uv}(t, l_{uv} - z)$ for all $z \in [0, l_{vu}]$.

   That is, given that we consider a symmetric directed graph $\mathcal{G} = (V, E)$ and the contrasting edges $e_{vu}, e_{uv} \in E$ of length $l_{vu} = l_{uv}$, we expect that the value $c_{vu}(t,z)$ along the edge $e_{vu}$ at time $t$ and position $z \in [0, l_{vu}]$ is equal to the value along the edge $e_{uv}$ at time $t$ and position $l_{vu} - z \in [0, l_{vu}]$.

Indeed, in practice, only the concentrations $c_{uv}(\cdot, 0)$ and $c_{uv}(\cdot, l_{uv})$ at the boundaries $\partial(0, l_{uv}) = \{0, l_{uv}\}$, i.e., *at the end* of such edges $e_{uv} \in E$ of length $l_{uv} > 0$ and capacity $\alpha_{uv} > 0$ are relevant for message passing (also cf. Eq. (22) below). Our message-generation function $\phi$ builds upon the MoC to avoid the explicit computation of concentrations $c_{uv}(\cdot, z)$ in between the boundaries, i.e., for all $z \in (0, l_{uv})$, as detailed in Section C.1.

Following Definition C.1.2, in Section C.2, we derive the message aggregation and node update scheme as defined in Equation (11) based the *information aggregation* through the *mixing at nodes*. For each node $v \in V$, the latter refers to the concentration $c_v : T \to \mathbb{R}_{\geq 0}$ determined by a weighted aggregation of the concentrations $c_{uv}(\cdot, l_{uv})$ carried by the *flow rates* $q_{uv} = \alpha_{uv}\nu_{uv} : T \to \mathbb{R}$ from (temporal) inflow neighbors $u \in \mathcal{N}_-(v, \cdot)$ over time $T = \mathbb{R}$. Formally, this is defined as

$$c_v(t) = \frac{\sum_{u \in \mathcal{N}_-(v,t)} q_{uv}(t) \cdot c_{uv}(t, l_{uv})}{\sum_{u \in \mathcal{N}_-(v,t)} q_{uv}(t)}, \tag{22}$$

which is a realization of Equation (3) with

$$w_{uv}(t) = \frac{q_{uv}(t)}{\sum_{u \in \mathcal{N}_-(v,t)} q_{uv}(t)} \text{ and } \phi(t, e_{uv}, \mathbf{c}, \boldsymbol{\nu}) = c_{uv}(t, l_{uv}).$$

---

[8]See Equation (14).

While Equation (21) is defined on the open sub-domain $\mathbb{R} \times (0, l_{uv})$ only, Equation (22) relies on the values $c_{uv}(\cdot, l_{uv})$ on the boundary $\partial(0, l_{uv}) = \{0, l_{uv}\}$ of $(0, l_{uv})$. Therefore, to be able to transfer information from these boundaries to the open sub-domain, but also for appearing expressions such as Equation (21) to be well-defined, we make the following assumptions throughout the rest of the section.

**Assumption C.2** (Smoothness assumptions). For each $v \in V$ and $u \in \mathcal{N}(v)$, let

- $c_{uv} : \mathbb{R} \times (0, l_{uv}) \to \mathbb{R}_{\geq 0}$ be differentiable,

- $c_{uv} : \mathbb{R} \times [0, l_{uv}] \to \mathbb{R}_{\geq 0}$ be continuous and

- $\nu_{uv} : \mathbb{R} \to \mathbb{R}$ be continuous.

### C.1 MESSAGE GENERATION: ADVECTIVE TRANSPORT ALONG EDGES

In this subsection, we assume that for each $v \in V$ and $u \in \mathcal{N}(v)$, Equation (21) with boundary conditions as defined in Definition C.1 holds. Hereby, as a function over time $\mathbb{R}$, the flow field $\nu_{uv} : \mathbb{R} \to \mathbb{R}$ can change its sign. Nevertheless, by the MoC, we can derive *characteristic curves* along which the function $c_{uv}$ remains constant over time $\mathbb{R}$ and space $(0, l_{uv})$.

**Lemma** (Lemma 3.1). *Let $v \in V$ and $u \in \mathcal{N}(v)$ hold. If the function $c_{uv}$ obeys Equation (1), for each $t_0 \in \mathbb{R}$ and $z_0 \in \mathbb{R}$, the curve*

$$\gamma_{t_0, z_0} : T_{t_0, z_0} \longrightarrow \mathbb{R}_{\geq 0}$$
$$t \longmapsto c_{uv}\left(t, z_0 + \int_{t_0}^{t} \nu_{uv}(s) \, ds\right)$$

*is constant on*

$$T_{t_0, z_0} = \left\{ t \in \mathbb{R} \mid 0 < z_0 + \int_{t_0}^{t'} \nu_{uv}(s) \, ds < l_{uv} \ \forall t' \in \left[\min\{t_0, t\}, \max\{t_0, t\}\right] \right\}.$$

Intuitively, in Lemma 3.1, we fix an arbitrary time $t_0 \in \mathbb{R}$ and a position $z_0 \in \mathbb{R}$ and follow any particle that is at this position $z_0$ at this time $t_0$ along the path $z(t) := z_0 + \int_{t_0}^{t} \nu_{uv}(s) \, ds$ it travels during the time $t - t_0$ (or $t_0 - t$) for some time $t \in T_{t_0, z_0}$ with possibly varying flow velocities $\nu_{uv}(s)$ for all $s \in [\min\{t_0, t\}, \max\{t_0, t\}]$. If we assign a concentration $c_{uv}(t_0, z_0)$ to this particle, the concentration should stay constant along the path this particle travels. The choice of $t \in T_{t_0, z_0}$ guarantees that the before-mentioned path remains within the edge $e_{uv}$, or in other words, in the open sub-domain $\mathbb{R} \times (0, l_{uv})$ of the function $c_{uv}$ on which Equation (21) is defined on, as we will investigate in detail in the next remark C.3.

*Remark* C.3 (On $T_{t_0, z_0}$). Since for $v \in V$ and $u \in \mathcal{N}(v)$, the velocity $\nu_{uv} : \mathbb{R} \to \mathbb{R}$ can change its sign, for each $t_0 \in \mathbb{R}$ and $z_0 \in \mathbb{R}$, the function

$$z : \mathbb{R} \longrightarrow \mathbb{R}, \ t \longmapsto z(t) := z_0 + \int_{t_0}^{t} \nu_{uv}(s) \, ds$$

is differentiable (and thus, continuous), but *not* monotone, meaning that even if $z(t_0) = z_0 \in (0, l_{uv})$ and $z(t) \in (0, l_{uv})$ holds for a $t \in \mathbb{R}$, this does *not* automatically mean that $z(t') \in (0, l_{uv})$ holds for all $t' \in [\min\{t_0, t\}, \max\{t_0, t\}]$, too. However, since Equation (21) only holds on the open sub-domain $\mathbb{R} \times (0, l_{uv})$ of the function $c_{uv} : \mathbb{R} \times [0, l_{uv}] \to \mathbb{R}_{\geq 0}$, any path from $z(t_0) = z_0$ to $z(t) = z_0 + \int_{t_0}^{t} \nu_{uv}(s) \, ds$ parametrized by time needs to be fully included in the space interval $(0, l_{uv})$, or in other words, $t \in T_{t_0, z_0}$ needs to hold.

Since the set $T_{t_0, z_0}$ can also be empty, we are interested in when this is not the case and what properties it satisfies in this case. The following Lemma C.4 answers these questions.

**Lemma C.4** (On $T_{t_0, z_0}$). *Let $v \in V$, $u \in \mathcal{N}(v)$, $t_0 \in \mathbb{R}$, $z_0 \in \mathbb{R}$ hold and let $T_{t_0, z_0}$ be defined as in Lemma 3.1.*

1. *If $z_0 \notin (0, l_{uv})$ holds, then $T_{t_0, z_0} = \emptyset$ holds.*

2. *If $z_0 \in (0, l_{uv})$ holds, then $T_{t_0,z_0} \neq \emptyset$ holds. Even more, $t_0 \in T_{t_0,z_0}$ holds.*

3. *If $z_0 \in (0, l_{uv})$ holds, there exist $t_l \in [-\infty, t_0)$ and $t_r \in (t_0, \infty]$ such that*

    (a) *if $t_l \neq -\infty$ holds, $z_0 + \int_{t_0}^{t_l} \nu_{uv}(s)\,ds \in \{0, l_{uv}\}$ holds,*

    (b) *if $t_r \neq +\infty$ holds, $z_0 + \int_{t_0}^{t_r} \nu_{uv}(s)\,ds \in \{0, l_{uv}\}$ holds and*

    (c) *$T_{t_0,z_0} = (t_l, t_r) = (t_0 - (t_0 - t_l), t_0 + (t_r - t_0))$ holds.*

    *Equivalently, there exist $\delta t_l, \delta t_r \in (0, \infty]$ such that $T_{t_0,z_0} = (t_0 - \delta t_l, t_0 + \delta t_l)$ holds.*

Since according to the previous Lemmas 3.1 and C.4, for any $t_0 \in \mathbb{R}$ and $z_0 \in (0, l_{uv})$, the concentration $c_{uv}$ is constant along characteristic curves $\gamma_{t_0,z_0}$ on an non-empty time interval $T_{t_0,z_0} = (t_1, t_r)$, it suffices to know the value of $c_{uv}$ at one position in this time interval to have knowledge of the concentration over all times in the interval:

**Theorem C.5** (Advective transport along edges). *Let $v \in V$ and $u \in \mathcal{N}(v)$ hold. If the function $c_{uv}$ obeys Equation (21), for each $t_0 \in \mathbb{R}$ and $z_0 \in (0, l_{uv})$,*

$$c_{uv}\left(t, z_0 + \int_{t_0}^{t} \nu_{uv}(s)\,ds\right) = c_{uv}\left(t, z_0 + \bar{\nu}_{uv}(t_0, t)\,(t - t_0)\right) = c_{uv}(t_0, z_0)$$

*holds for all $t \in T_{t_0,z_0}$ with $T_{t_0,z_0}$ as defined in Lemma 3.1 and for the mean velocity*

$$\bar{\nu}_{uv}(t_0, t) := \begin{cases} \fint_{t_0}^{t} \nu_{uv}(s)\,ds & \text{if } t \neq t_0 \\ 0 & \text{if } t = t_0 \end{cases} = \begin{cases} \frac{1}{t - t_0} \int_{t_0}^{t} \nu_{uv}(s)\,ds & \text{if } t \neq t_0 \\ 0 & \text{if } t = t_0 \end{cases}.$$

According to Lemma C.4.3, for any $t_0 \in \mathbb{R}$ and $z_0 \in (0, l_{uv})$, the set $T_{t_0,z_0}$ corresponds to a time interval $T_{t_0,z_0} = (t_1, t_r)$, such that on its boundary $\partial(t_1, t_r) = \{t_1, t_r\}$, the information reaches exactly one of the ends of the edge $e_{uv}$, i.e., the node $v \in V$ or the node $u \in \mathcal{N}(v)$. In view of the mixing at nodes (Eq. (22)), this motivates to leverage the advective transport along edges, i.e., Theorem C.5, and combine it with the boundary conditions, i.e., Definition C.1, in order to derive a representation of Equation (22) that does not depend on the concentrations $c_{uv}(t, l_{uv})$ at the end $z = l_{uv}$ of the edges $e_{uv}$ for each $u \in \mathcal{N}_-(v, t)$, but on the concentrations $c_u(t - \delta t_{uv})$ and $c_v(t - \delta t_{uv})$ at suitable times $t - \delta t_{uv} \in \mathbb{R}$. This, in turn, will serve as the basis for our message-passing algorithm (cf. Eq. (9)).

Intuitively, the time shift $\delta t_{uv} \in \mathbb{R}$ will correspond to the time that any information requires to travel along the edge $e_{uv}$ with the flow $\nu_{uv} : \mathbb{R} \to \mathbb{R}$ from one of the nodes $v \in V$ or $u \in \mathcal{N}(v)$ until it reaches the node $v \in V$ again. From the sender perspective, two different scenarios can occur: Either, the information leaves the node $u \in \mathcal{N}(v)$ at time $t - \delta t_{uv} \in \mathbb{R}$ and enters the node $v \in V$ at time $t \in \mathbb{R}$. We will call this scenario a *pass-through*. Or, the information leaves the node $v \in V$ at time $t - \delta t_{uv} \in \mathbb{R}$ and starts traveling to the node $u \in \mathcal{N}(v)$, but before it reaches that node, the flow $\nu_{vu} = -\nu_{uv} : \mathbb{R} \to \mathbb{R}$ changes its sign and the information enters the node $v \in V$ at time $t \in \mathbb{R}$ again. We will call this scenario a *self-loop*. Of course, the sign of the flow can also change multiple times before the information enters the node $v \in V$. As a first step, we now formalize the aforementioned pass-through and self-loop times formally and collect some helpful properties to get a deeper understanding of such. In practice, also corresponding inverse phenomena will appear. In summary, we call these times *transport times*, a name which we will motivate later.

**Definition C.6** (Transport times). *Let $v \in V$, $u \in \mathcal{N}(v)$ and $t \in \mathbb{R}$ hold. If there exists a $\delta t_{uv} \in \mathbb{R}$, such that*

$$\int_{t - \delta t_{uv}}^{t} \nu_{uv}(s)\,ds = l_{uv} \text{ and} \tag{23}$$

$$\int_{t - \delta t_{uv}}^{t'} \nu_{uv}(s)\,ds \in (0, l_{uv}) \tag{24}$$

*hold for all $t' \in (\min\{t - \delta t_{uv}, t\}, \max\{t - \delta t_{uv}, t\})$, we call $\delta t_{uv}$ the pass-through time w.r.t. the edge $e_{uv} \in E$ and time $t \in \mathbb{R}$. If instead, there exists a $\delta t_{uv} \in \mathbb{R}$, such that*

$$\int_{t - \delta t_{uv}}^{t} \nu_{uv}(s)\,ds = -l_{uv} \text{ and} \tag{25}$$

$$\int_{t-\delta t_{uv}}^{t^{'}} \nu_{uv}(s)\, ds \in (-l_{uv}, 0) \tag{26}$$

hold for all $t^{'} \in (\min\{t - \delta t_{uv}, t\}, \max\{t - \delta t_{uv}, t\})$, we call $\delta t_{uv}$ the *inverse pass-through time w.r.t. the edge $e_{uv} \in E$ and time $t \in \mathbb{R}$*. If instead, there exists a $\delta t_{uv} \in \mathbb{R}_{\neq 0}$, such that

$$\int_{t-\delta t_{uv}}^{t} \nu_{uv}(s)\, ds = 0 \text{ and} \tag{27}$$

$$\int_{t-\delta t_{uv}}^{t^{'}} \nu_{uv}(s)\, ds \in (0, l_{vu}), \tag{28}$$

hold for all $t^{'} \in (\min\{t - \delta t_{uv}, t\}, \max\{t - \delta t_{uv}, t\})$, we call $\delta t_{uv}$ the *self-loop time w.r.t. the edge $e_{uv} \in E$ and time $t \in \mathbb{R}$*. If instead, there exists a $\delta t_{uv} \in \mathbb{R}_{\neq 0}$, such that

$$\int_{t-\delta t_{uv}}^{t} \nu_{uv}(s)\, ds = 0 \text{ and} \tag{29}$$

$$\int_{t-\delta t_{uv}}^{t^{'}} \nu_{uv}(s)\, ds \in (-l_{uv}, 0), \tag{30}$$

hold for all $t^{'} \in (\min\{t - \delta t_{uv}, t\}, \max\{t - \delta t_{uv}, t\})$, we call $\delta t_{uv}$ the *inverse self-loop time w.r.t. the edge $e_{uv} \in E$ and time $t \in \mathbb{R}$*.

Usually, $e_{uv} \in E$ and $t \in \mathbb{R}$ are clear from the context and we just say *pass-through time*, *inverse pass-through time*, *self-loop time* or *inverse self-loop time*.

The following lemmas help to get a deeper understanding of the different kinds of transport times.

**Lemma C.7** (Pass-through time). *Let $v \in V$, $u \in \mathcal{N}(v)$ and $t \in \mathbb{R}$ hold. If there exists a pass-through time $\delta t_{uv} \in \mathbb{R}$ w.r.t. $e_{uv} \in E$ and $t \in \mathbb{R}$, the following properties hold:*

1. *$\delta t_{uv} \in \mathbb{R}_{\neq 0}$ and thus, $t - \delta t_{uv} \neq t$ holds.*

2. *There exists no other pass-through time of the same sign (i.e., there exists either exactly one or no positive pass-through time, and there exists exactly one or no negative pass-through time).*

3. *The mean velocity*

$$\overline{\nu}_{uv}(t) := \overline{\nu}_{uv}(t - \delta t_{uv}, t) = \fint_{t-\delta t_{uv}}^{t} \nu_{uv}(s)\, ds = \frac{1}{\delta t_{uv}} \int_{t-\delta t_{uv}}^{t} \nu_{uv}(s)\, ds$$

   *during the time interval $[\min\{t - \delta t_{uv}, t\}, \max\{t - \delta t_{uv}, t\}]$ as defined in Theorem C.5 is equal to $\overline{\nu}_{uv}(t) = \frac{l_{uv}}{\delta t_{uv}}$ and thus satisfies*

$$\overline{\nu}_{uv}(t) > 0 \iff \delta t_{uv} > 0 \text{ and}$$
$$\overline{\nu}_{uv}(t) < 0 \iff \delta t_{uv} < 0.$$

4. *The flows $q_{uv}(t - \delta t_{uv})$ and $q_{uv}(t)$ linked to the velocity $\nu_{uv}(\cdot) = \frac{q_{vu}(\cdot)}{\alpha_{vu}}$ satisfy*

$$\begin{cases} q_{uv}(t - \delta t_{uv}), q_{uv}(t) \geq 0 & \text{if } \delta t_{uv} > 0 \\ q_{uv}(t - \delta t_{uv}), q_{uv}(t) \leq 0 & \text{if } \delta t_{uv} < 0 \end{cases}.$$

According to Lemma C.7, if a pass-through time $\delta t_{uv} > 0$ is positive, so does $\overline{\nu}_{uv}(t) > 0$ hold, and there is an average flow from the node $u \in \mathcal{N}(v)$ to the node $v \in V$ during the time $[t - \delta t_{uv}, t]$.[9]

---

[9]More precisely, since in this case, $q_{uv}(t - \delta t_{uv}), q_{uv}(t) \geq 0$ holds, we know that at time $t - \delta t_{uv} \in \mathbb{R}$ and $t \in \mathbb{R}$, there is a flow from $u \in \mathcal{N}_-(v, t - \delta t_{uv}) \cap \mathcal{N}_-(v, t)$ to $v \in V$. Note that we tacitly ignored the case of zero flows here due to remark E.1.

If in contrast, a pass-through time $\delta t_{uv} < 0$ is negative, so does $\overline{\nu}_{uv}(t) < 0$ hold, and there is an average flow from the node $v \in V$ to the node $u \in \mathcal{N}(v)$ during the time $[t, t - \delta t_{uv}]$.[10] In both cases, the time $\delta t_{uv} = \frac{l_{uv}}{\overline{\nu}_{uv}(t)}$ is exactly the time that any particle transported by the average flow $\overline{q}_{uv}(t) = \alpha_{uv}\,\overline{\nu}_{uv}(t)$ needs to travel along the edge $e_{uv}$ of length $l_{uv}$. Therefore, we call $\delta t_{uv}$ the *pass-through time*. The cases are intuitively connected by the equality[11]

$$\int_{t-\delta t_{uv}}^{t} \nu_{uv}(s)\,ds = \int_{t}^{t-\delta t_{uv}} -\nu_{uv}(s)\,ds = \int_{t}^{t-\delta t_{uv}} \nu_{vu}(s)\,ds,$$

where in dependence of the sign of the pass-through time $\delta t_{uv} \in \mathbb{R}_{\neq 0}$, we use the oriented of the two integrals on the left- or right-hand side for interpretation. We can make similar observations for self-loop times.

**Lemma C.8** (Self-loop time). *Let $v \in V$, $u \in \mathcal{N}(v)$ and $t \in \mathbb{R}$ hold. If there exists a self-loop time $\delta t_{uv} \in \mathbb{R}_{\neq 0}$ w.r.t. $e_{uv} \in E$ and $t \in \mathbb{R}$, the following properties hold:*

1. *There exists no other self-loop time of the same sign (i.e., there exists either exactly one or no positive self-loop time, and there exists exactly one or no negative self-loop time).*

2. *The mean velocity*

$$\overline{\nu}_{uv}(t) := \overline{\nu}_{uv}(t - \delta t_{uv}, t) = \fint_{t-\delta t_{uv}}^{t} \nu_{uv}(s)\,ds = \frac{1}{\delta t_{uv}} \int_{t-\delta t_{uv}}^{t} \nu_{uv}(s)\,ds$$

   *during the time interval $[\min\{t - \delta t_{uv}, t\}, \max\{t - \delta t_{uv}, t\}]$ as defined in Theorem C.5 satisfies $\overline{\nu}_{uv}(t) = 0$.*

3. *The flows $q_{uv}(t - \delta t_{uv})$ and $q_{uv}(t)$ linked to the velocity $\nu_{uv}(\cdot) = \frac{q_{vu}(\cdot)}{\alpha_{vu}}$ satisfy*

$$\begin{cases} q_{uv}(t - \delta t_{uv}) \geq 0,\ q_{uv}(t) \leq 0 & \text{if } \delta t_{uv} > 0 \\ q_{uv}(t - \delta t_{uv}) \leq 0,\ q_{uv}(t) \geq 0 & \text{if } \delta t_{uv} < 0 \end{cases}.$$

According to Lemma C.8, independent on whether a self-loop time $\delta t_{uv} \in \mathbb{R}_{\neq 0}$ is positive or negative, $\overline{\nu}_{uv}(t) = 0$ holds. If a self-loop time $\delta t_{uv} > 0$ is positive, there is an average flow from the node $u \in \mathcal{N}(v)$ to itself during the time $[t - \delta t_{uv}, t]$.[12] If in contrast, a self-loop time $\delta t_{uv} < 0$ is negative, there is an average flow from the node $u \in \mathcal{N}(v)$ to itself during the time $[t, t - \delta t_{uv}]$.[13] In both cases, the time $\delta t_{uv}$ is exactly the time that any particle transported by the average flow $\overline{q}_{uv}(t) = \alpha_{uv}\,\overline{\nu}_{uv}(t) = 0$ needs to travel from $u$ along some part of the edge $e_{uv}$ until the flow changes its sign and the particle travels back to the node $u$ it came from. Therefore, we call $\delta t_{uv}$ the *self-loop time*.

As another result that will turn out to be useful in practise, by the symmetry of the underlying graph $g$ and the conservation of flows (Eq. (4)), an inverse pass-through time induces a pass-through time and vice versa, while an inverse self-loop time induces a self-loop time and vice versa.

**Lemma C.9** (Inverse pass-through time). *Let $v \in V$, $u \in \mathcal{N}(v)$ and $t \in \mathbb{R}$ hold. $\delta t_{uv} \in \mathbb{R}$ is an inverse pass-through time w.r.t. $e_{uv} \in E$ and $t \in \mathbb{R}$ if and only if (iff) $\delta t_{vu} := \delta t_{uv} \in \mathbb{R}$ is a pass-through time w.r.t. $e_{vu} \in E$ and $t \in \mathbb{R}$*

**Lemma C.10** (Inverse self-loop time). *Let $v \in V$, $u \in \mathcal{N}(v)$ and $t \in \mathbb{R}$ hold. $\delta t_{uv} \in \mathbb{R}$ is an inverse self-loop time w.r.t. $e_{uv} \in E$ and $t \in \mathbb{R}$ iff $\delta t_{vu} := \delta t_{uv} \in \mathbb{R}$ is a self-loop time w.r.t. $e_{vu} \in E$ and $t \in \mathbb{R}$*

---

[10]More precisely, since in this case, $q_{uv}(t - \delta t_{uv}), q_{uv}(t) \leq 0$ holds, we know that at time $t \in \mathbb{R}$ and $t - \delta t_{uv} \in \mathbb{R}$, there is a flow from $v \in V$ to $u \in \mathcal{N}_+(v, t) \cap \mathcal{N}_+(v, t - \delta t_{uv})$. Note that we tacitly ignored the case of zero flows here due to remark E.1.

[11]The proof follows similar arguments as given in, e.g., the proof of Lemma C.9.

[12]More precisely, since in this case, $q_{uv}(t - \delta t_{uv}) \geq 0$ and $q_{uv}(t) \leq 0$ holds, we know that at time $t - \delta t_{uv} \in \mathbb{R}$, there is a flow from $u \in \mathcal{N}_-(v, t - \delta t_{uv})$ to $v \in V$, while at time $t \in \mathbb{R}$, there is a flow from $v \in V$ to $u \in \mathcal{N}_+(v, t)$. Note that we tacitly ignored the case of zero flows here due to remark E.2.

[13]More precisely, since in this case, $q_{uv}(t) \geq 0$ and $q_{uv}(t - \delta t_{uv}) \leq 0$ holds, we know that at time $t \in \mathbb{R}$, there is a flow from $u \in \mathcal{N}_-(v, t)$ to $v \in V$, while at time $t - \delta t_{uv} \in \mathbb{R}$, there is a flow from $v \in V$ to $u \in \mathcal{N}_+(v, t - \delta t_{uv})$. Note that we tacitly ignored the case of zero flows here due to Remark E.2.

We now collected all relevant properties which help us to prove the main results of this section as motivated above: Given a pass-through time $\delta t_{uv} \in \mathbb{R}$ w.r.t. $e_{uv} \in E$ and $t \in \mathbb{R}$, we can re-write the concentrations $c_{uv}(t, l_{uv})$ and $c_{vu}(t - \delta t_{uv}, l_{vu})$ at the end $z = l_{uv} = l_{vu}$ of the edges $e_{uv}$ and $e_{vu}$ using the concentrations $c_u : \mathbb{R} \to \mathbb{R}_{\geq 0}$ and $c_v : \mathbb{R} \to \mathbb{R}_{\geq 0}$, respectively:

**Theorem C.11** (Pass-through time). *Let $v \in V$, $u \in \mathcal{N}(v)$ and $t \in \mathbb{R}$ hold. If the function $c_{uv}$ obeys Equation* (21) *and there exists a pass-through time $\delta t_{uv} \in \mathbb{R}_{\neq 0}$ w.r.t. $e_{uv} \in E$ and $t \in \mathbb{R}$, we obtain*

$$c_{uv}(t, l_{uv}) = c_{uv}(t - \delta t_{uv}, 0) = c_{uv}\left(t - \frac{l_{uv}}{\overline{\nu}_{uv}(t)}, 0\right).$$

*Even more, the concentrations $c_{uv}(\cdot, l_{uv})$ and $c_{vu}(\cdot, l_{vu})$ at the end $z = l_{uv} = l_{vu}$ of the edges $e_{uv} \in E$ and $e_{vu} \in E$ are connected to the concentrations $c_u$ and $c_v$ of the nodes of that edge, respectively, by*

$$\begin{cases} c_{uv}(t, l_{uv}) = c_u(t - \delta t_{uv}) & \text{if } \delta t_{uv} > 0 \\ c_{vu}(t - \delta t_{uv}, l_{vu}) = c_v(t) & \text{if } \delta t_{uv} < 0 \end{cases}.$$

According to Theorem C.11, if a pass-through time $\delta t_{uv} > 0$ is positive, the concentration $c_{uv}(t, l_{uv})$ at time $t \in \mathbb{R}$ and at the end of the edge $e_{uv} \in E$, along which there is an average flow from the node $u \in \mathcal{N}(v)$ to the node $v \in V$ during the time $[t - \delta t_{uv}, t]$, is equal to the concentration $c_u(t - \delta t_{uv})$ at the node $u \in \mathcal{N}(v)$ and $-\delta t_{uv} < 0$ units of time *earlier* than $t$.

If in contrast, a pass-through time $\delta t_{uv} < 0$ is negative, the concentration $c_{vu}(t - \delta t_{uv}, l_{vu})$ at time $t - \delta t_{uv} \in \mathbb{R}$ and at the end of the edge $e_{vu} \in E$, along which there is an average flow from the node $v \in V$ to the node $u \in \mathcal{N}(v)$ during the time $[t, t - \delta t_{uv}]$, is equal to the concentration $c_v(t)$ at the node $v \in V$ and $+\delta t_{uv} < 0$ units of time *earlier* than $t - \delta t_{uv}$.

Analogously, given a self-loop time $\delta t_{uv} \in \mathbb{R}$ w.r.t. $e_{uv} \in E$ and $t \in \mathbb{R}$, we can re-write the concentration $c_{vu}(t, l_{vu})$ at the end $z = l_{vu} = l_{uv}$ of the edge $e_{vu}$ using the concentration $c_u : \mathbb{R} \to \mathbb{R}_{\geq 0}$:

**Theorem C.12** (Self-loop time). *Let $v \in V$, $u \in \mathcal{N}(v)$ and $t \in \mathbb{R}$ hold. If the function $c_{uv}$ obeys Equation* (21) *and there exists a self-loop time $\delta t_{uv} \in \mathbb{R}_{\neq 0}$ w.r.t. $e_{uv} \in E$ and $t \in \mathbb{R}$, we obtain*

$$c_{uv}(t, 0) = c_{uv}(t - \delta t_{uv}, 0).$$

*Even more, the concentration $c_{vu}(\cdot, l_{vu})$ at the end $z = l_{vu} = l_{uv}$ of the edge $e_{vu} \in E$ is connected to the concentration $c_u$ of the node of that edge by*

$$\begin{cases} c_{vu}(t, l_{vu}) = c_u(t - \delta t_{uv}) & \text{if } \delta t_{uv} > 0 \\ c_{vu}(t - \delta t_{uv}, l_{vv}) = c_u(t) & \text{if } \delta t_{uv} < 0 \end{cases}.$$

According to Theorem C.12, if a self-loop time $\delta t_{uv} > 0$ is positive, the concentration $c_{vu}(t, l_{vu})$ at time $t \in \mathbb{R}$ and at the end of the edge $e_{vu} \in E$, along which there is an average flow from the node $u \in \mathcal{N}(v)$ to itself during the time $[t - \delta t_{uv}, t]$, is equal to the concentration $c_u(t - \delta t_{uv})$ at the node $u \in \mathcal{N}(v)$ and $-\delta t_{uv} < 0$ units of time *earlier* than $t$.

If in contrast, a self-loop time $\delta t_{uv} < 0$ is negative, the concentration $c_{vu}(t - \delta t_{uv}, l_{vu})$ at time $t - \delta t_{uv} \in \mathbb{R}$ and at the end of the edge $e_{vu} \in E$, along which there is an average flow from the node $u \in \mathcal{N}(v)$ to itself during the time $[t, t - \delta t_{uv}]$, is equal to the concentration $c_u(t)$ at the node $u \in \mathcal{N}(v)$ and $+\delta t_{uv} < 0$ units of time *earlier* than $t - \delta t_{uv}$.

In summary, Theorem C.11 and Theorem C.12 illustrate the time shift required to pass information from one node to another. Therefore we call these shifts *transport times*, and the underlying concepts serve as a basis for our advection-message-generation function $\phi$ (cf. Eq. (13) and (3)). To be able to properly define it, we need to make sure that for each pair of nodes $v \in V$, $u \in \mathcal{N}(v)$ and time $t \in \mathbb{R}$, such a transport time exists and is unique. For the sake of brevity, for such a pair, we define the function $z$ as given in Equation (7). Then we obtain the following existence and uniqueness result:

**Theorem C.13** (Existence and uniqueness of positive transport times)**.** *Let* $v \in V$, $u \in \mathcal{N}(v)$ *and* $t \in \mathbb{R}$ *hold. If* $\nu_{uv}(t) \neq 0$ *holds and the set*

$$A_{uv}(t) := \left\{ \delta t \in (0, \infty) \,\middle|\, \int_{t-\delta t}^{t} \nu_{uv}(s) \, ds \in \{-l_{uv}, 0, l_{uv}\} \right\}$$

$$= \left\{ \delta t \in (0, \infty) \,\middle|\, z(\delta t) \in \{-l_{uv}, 0, l_{uv}\} \right\}$$

*is not empty, the following properties hold:*

1. $\delta t_{uv} := \min A_{uv}(t) = \min\{\delta t \in (0, \infty) \,|\, z(\delta t) \in \{-l_{uv}, 0, l_{uv}\}\} \in (0, \infty)$ *exists.*

2. $\delta t_{uv}$ *is a positive pass-through time, inverse pass-through time, self-loop time or inverse self-loop time. More specifically,* $\delta t_{uv}$ *is (always w.r.t.* $e_{uv}$ *and* $t$*) ...*

   | | | |
   |---|---|---|
   | *... an inverse pass-through time* | $\Longleftrightarrow$ | $z(\delta t_{uv}) = -l_{uv},$ |
   | *... a self-loop time or an inverse self-loop time* | $\Longleftrightarrow$ | $z(\delta t_{uv}) = 0,$ |
   | *... a pass-through time* | $\Longleftrightarrow$ | $z(\delta t_{uv}) = l_{uv}.$ |

3. *There exists no other positive pass-through time, inverse pass-through time, self-loop time or inverse self-loop time than* $\delta t_{uv}$*.*

Theorem C.13 guarantees that the mapping

$$s : \mathbb{R} \times E \times F(\mathbb{R}, \mathbb{R}^{n_e}) \longrightarrow (0, \infty), \ (t, e_{uv}, \boldsymbol{\nu}) \longmapsto s(t, e_{uv}, \boldsymbol{\nu}) := \delta t_{uv} \qquad (31)$$

is well-defined and therefore allows to define the advection-message-generation function in accordance to the results of Theorems C.11 and C.12 as defined in Equation (9). Throughout most parts of this work, $t \in \mathbb{R}$, $e_{uv} \in E$ and $\boldsymbol{\nu} \in F(\mathbb{R}, \mathbb{R}^{n_n})$ are known from the context and we use the notation $\delta t_{uv}$ as a shortcut for $s(t, e_{uv}, \boldsymbol{\nu})$ to ease notation.

*Remark* C.14 (Double assignment phenomenon)*.* Note that through the relation[14]

$$\int_{t-\delta t_{uv}}^{t} \nu_{vu}(s) \, ds = - \int_{t-\delta t_{uv}}^{t} \nu_{uv}(s) \, ds,$$

two neighboring nodes $v \in V$ and $u \in \mathcal{N}(v)$ will be assigned to the same transport time $\delta t_{uv}$, corresponding to either pass-through and inverse pass-through times (cf. Lemma C.9) or self-loop and inverse self-loop times (cf. Lemma C.10). We call this the *double-assignment phenomenon*.

In the following Section C.2, we integrate our advection-message-generation function $\phi$ into the mixing at nodes (Eq. (22)) and show that the resulting representation of such corresponds to our advection-message-passing algorithm as presented in Section 3 (Eq. (11)), which automatically handles the double-assignment phenomenon.

## C.2 MESSAGE AGGREGATION AND UPDATE: MIXING AT NODES

As investigated in the previous Section C.1, for a node $v \in V$ and a time $t \in \mathbb{R}$, we are interested in a representation of the mixing at nodes (Eq. (22)) that does not depend on the concentrations $c_{uv}(t, l_{uv})$ at the end $z = l_{uv}$ of the edges $e_{uv}$ for each $u \in \mathcal{N}_-(v, t)$, but on the concentrations $c_u(t - \delta t_{uv})$ and $c_v(t - \delta t_{uv})$ shifted by suitable *transport times* $\delta t_{uv} \in \mathbb{R}$. Since in practice, we can only assume to have knowledge of the past, the suitability refers to the requirement that $\delta t_{uv} > 0$ should hold. Theorem C.13 guarantees the existence and uniqueness of such.

In this subsection, we prove that the message-generation function as defined in Equation (9) indeed assigns to each time $t \in \mathbb{R}$ and each edge $e_{uv} \in E$ a message $\phi(t, e_{uv}, \mathbf{c}, \boldsymbol{\nu})$ that determines the correct representation of $c_{uv}(t, l_{uv})$ from the correct source under consideration of the concentrations $\mathbf{c}$ and the flow field $\boldsymbol{\nu}$. The following lemma will help reaching this goal.

---

[14]See, e.g., the proof of Lemma C.9.

**Lemma C.15** (Transport times). *Let $v \in V$, $u \in \mathcal{N}(v)$ and $t \in \mathbb{R}$ hold. In the setting of Theorem C.13, the following property holds:*

1. *$\delta t_{uv}$ is (always w.r.t. $e_{uv}$ and $t$) ...*

   | | |
   |---|---|
   | *... a pass-through time* | $\iff z(\delta t_{uv}) \in \{-l_{uv}, l_{uv}\}$ *and* $q_{uv}(t) > 0$, |
   | *... an inverse pass-through time* | $\iff z(\delta t_{uv}) \in \{-l_{uv}, l_{uv}\}$ *and* $q_{uv}(t) < 0$, |
   | *... a self-loop time* | $\iff z(\delta t_{uv}) = 0$ *and* $q_{uv}(t) < 0$, |
   | *... an inverse self-loop time* | $\iff z(\delta t_{uv}) = 0$ *and* $q_{uv}(t) > 0$. |

Lemma C.15 justifies why in the definition of the advection-message-generation function $\phi$ (Eq. (9)), we do not need to distinguish between pass-through and inverse pass-through or self-loop and inverse self-loop times and though automatically handle the double assignment phenomenon (cf. Remark C.14): Taking into account the sign of the flow rate $q_{uv}(t)$ at time $t \in \mathbb{R}$, corresponding counterparts will vanish in the mixing at nodes due to the summation over inflow neighbors $u \in \mathcal{N}_{-}(v, t)$ only (cf. Eq. (22)) and in our advection-message-passing algorithm due to the ReLU-function (cf. Eq. (11)).

Moreover, the assumptions of Theorem C.13 and Lemma C.15 are naturally satisfied in the contexts where they are applied: On the one hand, summands where the velocity $\nu_{uv}(t)$ is equal to zero do not contribute by construction of the mixing at nodes Equation (22). On the other hand, whenever the set $A_{uv}(t)$ is empty (that is, no transport time exists), we assume to have knowledge of the values $c_v(t)$ for the respective times $t$: In cases of initial value problems, these values are known by definition (cf. remark (3.5)); for boundary value problems we assume an initial state of zero at non-boundary nodes, these values $c_v(t)$ for the respective times $t$ are then also zero by definition.

Consequently, we can combine the results from the advective transport along edges (Section C.1) with the results of this Section to obtain a message-passing representation of the mixing at nodes (Eq. (22)), which indeed coincides with our final message-passing scheme (11):

**Theorem C.16** (Mixing at nodes leveraging advective transport along edges). *Let $v \in V$, $u \in \mathcal{N}(v)$ and $t \in \mathbb{R}$ hold. Let $\phi$ be as defined in Equation (9). We assume that for each $u \in \mathcal{N}(v)$, the set $A_{uv}(t)$ as defined in Theorem C.15 is not empty. If the function $c_v$ obeys Equation (22) and if for each $u \in \mathcal{N}(v)$, the function $c_{uv}$ obeys Equation (21), we obtain*

$$
\begin{aligned}
c_v(t) &= \frac{1}{s} \left( \sum_{\substack{u \in \mathcal{N}(v) \\ z(\delta t) \in \{-l_{uv}, l_{uv}\}}} \mathrm{ReLU}(q_{uv}(t)) \cdot c_u(t - \delta t_{uv}) \right. \\
&\qquad \left. + \sum_{\substack{u \in \mathcal{N}(v) \\ z(\delta t) = 0}} \mathrm{ReLU}(q_{uv}(t)) \cdot c_v(t - \delta t_{uv}) \right) \\
&= \frac{1}{s} \sum_{u \in \mathcal{N}(v)} \mathrm{ReLU}(q_{uv}(t)) \cdot \phi(t, e_{uv}, \mathbf{c}, \boldsymbol{\nu}) \\
&= \frac{1}{s} \sum_{u \in \mathcal{N}_{-}(v)} q_{uv}(t) \cdot \phi(t, e_{uv}, \mathbf{c}, \boldsymbol{\nu})
\end{aligned}
\tag{32}
$$

*with scaling factor*

$$
s = \sum_{u \in \mathcal{N}(v)} \mathrm{ReLU}(q_{uv}(t)) = \sum_{u \in \mathcal{N}_{-}(v)} q_{uv}(t)
$$

*and transport time $\delta t_{uv} = \min A_{uv}(t)$ as defined in Theorem C.13.*

# D    DETAILS ON THE EXPERIMENTS

## D.1    EXPERIMENT 1: LEARNING ADVECTION AND REACTION ON A REAL-WORLD METRIC GRAPH

**Domain**    The two WDSs we are working with are Hanoi (Vrachimis et al., 2018), displayed in Figure 7, and L-Town (Vrachimis et al., 2020), displayed in Figure 3 and 8.

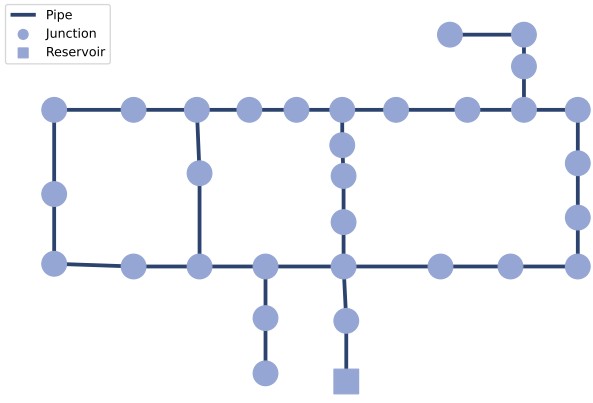

Figure 7: The WDS Hanoi with 32 nodes and 34 pipes.

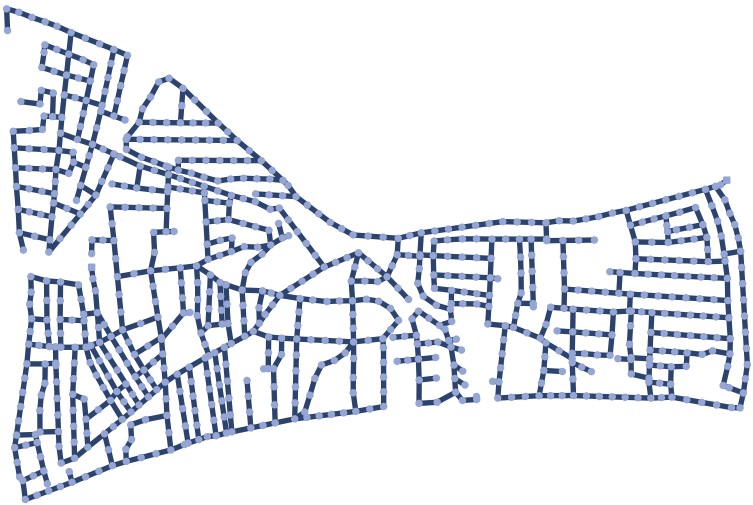

Figure 8: The WDS L-Town with 784 nodes and 908 pipes.

In both networks, we model chlorine decay, which is part of the edge dynamics in Section 5.1, as a first-order reaction over time and also include interactions with the pipe wall, which depend on the pipe diameter $\rho_{uv} \in \mathbb{R}_{\geq 0}$. This is formally defined as

$$f_r(t, z, e_{uv}) = -\left( k_b + \frac{4k_w}{\rho_{uv}} \right) c_{uv}(t, z),$$
(33)

where the parameters $k_b \in \mathbb{R}_{\geq 0}$ and $k_w \in \mathbb{R}_{\geq 0}$ control the intensity of chlorine decay over time and due to wall reactions, respectively.

**Dataset**    We generate datasets of 512 training samples and 128 testing samples for each WDSs with 700 time steps per sample using a sampling frequency of $dt = 60\,[s]$. The samples differ in

edge flows, edge lengths, and edge capacities, which are randomly samples as described below. As stated in Section 5.1, we use EPyT-Flow (Artelt et al., 2024) based on EPANET-MSX (Shang et al., 2023) as the simulator. The injection pattern (Dirichlet boundary condition) at which chlorine is injected at boundary nodes $V_b = V_r$ (all reservoirs) is

$$c_v(t_i) = \sum_{j=1}^{2} \frac{3}{j} \left(1 + \sin(2\pi t f_j)\right)$$

for all $v \in V_r$, $t_i \in \mathbb{R}$, $i \in I$ and with $f_1 = 0.507$ and $f_2 = -0.118$. The pattern is visualized in Figure 9.

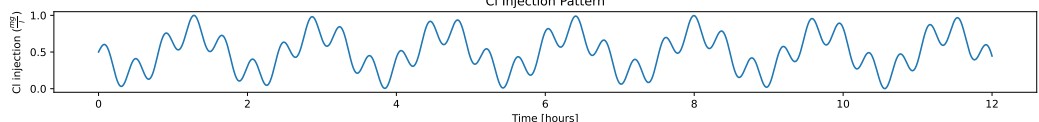

Figure 9: Pattern used for the chlorine injection at reservoir nodes.

The values for the reaction coefficients in Equation (33) are set to $k_b = 0.04$ and $k_w = 0.034$. We sample pipe lengths $l \sim U(0.1, 80)$ and pipe diameters $\rho \sim U(0.1, 80)$ uniformly per sample at random. To maintain the typical WDS topology of larger pipes at reservoirs and successively smaller pipes towards leaf nodes, diameters are ordered from low to high, as in the original WDSs Hanoi and L-Town. Nodal demands $\mathbf{d}$ determine the flow field $\boldsymbol{\nu}$, and to achieve temporally interesting flow patterns $\boldsymbol{\nu}$, we generate demands for each node $v \in V$ and time steps $t_i \in [0, 700]$ by cumulative sum over noise:

$$d_v(t_i) = \left| \sum_{s=1}^{t} \epsilon_s + z_{v,s} \right|, \quad \epsilon_s \sim \mathcal{N}(0, 1), \ z_{v,s} \sim \mathcal{N}(0, 0.04^2).$$

The resulting hydraulics show flow velocities $\boldsymbol{\nu}$ that vary in space and time and can also change direction. Sampling these parameters at random may result in physically implausible states that generate negative pressures, however, since the quality simulation only depends on flows and not on pressures, the resulting evolution of chlorine in the system is still valid.

**Model** The MLP that is intended to learn the reactions consists of two linear layers with 8 neurons each, and a SELU activation (Klambauer et al., 2017) in between. We train the model with the Adam optimizer (Kingma & Ba, 2014) over 250 epochs with a batch size of 256 using a learning rate of 0.001 that is reduced by a factor of 0.2 if the training loss did not decrease for the last 3 epochs. The loss function is the mean absolute error (MAE). Boundary nodes are masked.

**Baselines** The baselines we compare ourselves to are:

- **NNConv**: As another simple baseline, we use the kernel-based MPNN from (Gilmer et al., 2017). The inputs for the nodes $X \in \mathbb{R}^{|V| \times T}$, with $T = T_{hi} + T_{pr}$ is a feature matrix populated with the complete time series at boundary nodes, as well as the first time step as the *kown history* at all other nodes. The edge feature matrix $\mathcal{E} \in \mathbb{R}^{|E| \times 3T}$ contains the time series of scaled edge flows, edge capacities, and transport times. Node and edge features are first mappeed into a 128-dimensional latent space via two separate linear layers. After that three GG-NN convolutional layers (Gilmer et al., 2017) (NNConv implementation from torch-geometric) are applied. The message function is implemented by a two-layer MLP with a ReLU activation between the two layers. The first layer consis of 128 neurons, the second one by $128^2$ to form the linear transform $M \in \mathbb{R}^{128 \times 128}$ applied the the node

Table 3: Optuna Search Space for Different Model Types

| Parameter | NNConv | PDE-GNN | GPSConv | A-DGN |
|---|---|---|---|---|
| *hidden_dim* | {64, 96, 128, 160} | | | |
| *num_layers* | [2, 6] | | | |
| *learning_rate* | $[10^{-4}, 3 \times 10^{-3}]$ (log) | | | |
| *weight_decay* | $[10^{-6}, 10^{-1}]$ (log) | | | |
| *batch_size* | {64, 128, 256} | | | |
| *clip_grad_norm* | – | $\{10^{-4}, 0.5, 1.0, 2.0, -\}$ | – | – |
| $k_{pr}$ (training) | 700 | {32, 100, 150} | 700 | 700 |
| *dropout* | – | [0.0, 0.3] | [0.0, 0.3] | – |
| *num_heads* | – | – | {2,4,8} | – |
| *attn_dropout* | – | – | [0.0, 0.3] | – |
| *step_size* / $\epsilon$ | – | {1/4, 1/3, 1/2, 1} | – | { 1.0, 0.1, 0.01, 0.001 } |
| $\gamma$ | – | – | – | { 1.0, 0.1, 0.01, 0.001 } |
| $\Phi$ | – | – | – | { NNConv, GINE, GatedGCN } |
| *num_iters* | – | – | – | { 1, 2, 5, 10, 20, 50 } |
| **Fixed Hyperparameters** | | | | |
| $k_{hi}$ | 1 | | | |
| $k_{pr}$ (evaluation) | 700 | | | |
| *loss_fn* | L1 | | | |
| *epochs* | 60 | | | |

features. The messages are aggregated via sum aggregation and decoded by a single linear layer Calibrated manually, we decay the learning rate by 0.2 decay upon training loss plateau over 3 epochs, we set the initial learning rate to 0.001.

- **PDE-GNN** (Hermes et al., 2025) is a PDE-style GNN which solves initial and boundary value problems and has been proposed for chlorine estimation in WDSs. The model learns a PDE function that is integrated by Euler integration. Calibrated manually, we apply the same scheduler with 0.2 learning rate decay upon training loss plateau over 3 epochs, we set the initial learning rate to 0.001. The time step used for Euler integration is set to $dt = 0.25$. Other hyperparameters are left as specified in (Hermes et al., 2025).

- **GPSConv** (Rampášek et al., 2022) is a graph transformer that we include as a representative of an architecture capable of capturing global dependencies. As positional encodings, we utilize the graph Laplacian eigenvectors, since positional encodings based on random walks would degenerate to zero if the initial conditions are zero. The Laplacian positional encoding is truncated to 30 components, allowing the transfer-learning experiment from Hanoi (32 nodes) to the L-Town (785 nodes), and embedded to 8 dimensions via a linear layer. We use residual GatedGCN (Bresson & Laurent, 2017) as the message-passing module.

- **A-DGN** (Gravina et al., 2023) is a dynamical-systems based GNN, where the message-passing operation acts as a differential operator, similar to an ODE. As such, it computes updates $\frac{\partial \mathbf{X}}{\partial l}$ of the node features $\mathbf{X}$ that are integrated over several layers $l$ via Euler integration with a step size of $\epsilon$. This model allows deep architectures that are stable and non-dissipative, mediated through anti-symmetric weight matrices applied to the output of any message passing layer $\Phi$. We use this model in the weight-sharing style and select options for $\Phi$ that accommodate edge features, specifically NNConv, GINE, and GatedGCN (Gilmer et al., 2017; Hu* et al., 2020; Bresson & Laurent, 2017). As generalization is a focus here, we evaluate two variations A-DGN and A-DGN$_{Dia}$, to account for the difference in size of the training and evaluation graph: The first is a hyperparameter tuned baseline, while for the second variant, we change the number of iterations of the trained model to

the graph diameter. We have found that the latter works much better when the step size $\epsilon$ is adapted as well. Specifically, we set $\epsilon' = \epsilon \frac{num\_iters}{diameter(\mathcal{G})}$, and $num\_iters' = diameter(\mathcal{G})$.

- **PI DEEPONET** is a *LEGO*-like neural operator for metric graphs (Blechschmidt et al., 2025) originally developed for drift–diffusion equations. While this approach is highly effective for diffusive dynamics, its direct application to purely hyperbolic problems such as linear advection is not straightforward. First, the method assumes flow-directed edges, effectively imposing a fixed transport direction that is incompatible with more general advection fields. Second, at inner nodes the coupling is enforced through Kirchhoff–Neumann conditions, which conserve diffusive gradient fluxes $\partial_z c$. For hyperbolic dynamics, the flux is defined as $\nu(t, z)c(t, z)$ which would require a different coupling law such as mixing. For these reasons we omit this work as a baseline for now.

- **PINN-based approaches** on metric graphs (Blechschmidt et al., 2022; Laczkó et al., 2025) are effective for learning solutions of specific PDE settings on metric graphs and are suited for inverse problems. However, PINNs are not operator-learning frameworks: A change in PDE parameters or boundary conditions typically requires retraining. Since our focus is on flexible models that generalize across varying flow conditions and topologies, PINN-based approaches are not an appropriate baseline in our setting.

For a fair comparison of MeGA-MP to the other methods, we run a small-scale hyperparameter tuning for every baseline method using Optuna (Akiba et al., 2019). The search space is specified in Table 3. Every configuration is trained for 60 epochs using the Adam optimizer. Trials are terminated at an earlier epoch if the trial's result is worse than the median of results of previous trials at the same epoch (median pruning). New configurations are sampled based on the tree-structured Parzen Estimator (TPE) (Watanabe, 2025) implemented in optuna.

**Results** In Figures 10 to 13, we compare MeGA-MP to all baselines and the ground truth (the simulator EPANET) for both the advection-reaction dynamical system and the advection dynamical system on both the training WDS Hanoi and the test WDS L-Town. In Figures 14 to 17, we compare MeGA-MP with and without a reaction MLP for both the advection-reaction dynamical system and the advection dynamical system on both the training WDS Hanoi and the test WDS L-Town. An overview of these figures and their differences can also be found in Table 4.

Table 4: Overview of the results.

|  |  | Hanoi | L-Town |
|---|---|---|---|
| Comparison to baselines | Advection-reaction dynamics (MeGA-MP with MLP) | Figure 10 | Figure 11 |
|  | Advection dynamics (MeGA-MP without MLP) | Figure 12 | Figure 13 |
| Ablation studies | Advection-reaction dynamics (MeGA-MP with and without MLP) | Figure 14 | Figure 15 |
|  | Advection dynamics (MeGA-MP with and without MLP) | Figure 16 | Figure 17 |

One can see that MeGA-MP is the only model that performs consistently accurate for both the advection-reaction dynamics and the reaction dynamics over multiple nodes and generalizes well from the training WDS Hanoi to the test WDS L-Town (Figure 10 to 13). The baselines show different generalization behavior, emphasizing the challenges in modeling advection-dominated dynamics on metric graphs. For example, while PDE-GNN performs well on the training WDS Hanoi (Figure 10 and 12), it shows smoothing behavior for the advection-reaction dynamics, but explodes for the advection dynamics on the test WDS L-Town. We assume that this is the case because the reaction components (chlorine decay) in the data introduce a natural bias in dampening the predictions of the model (Figure 11). In contrast, without this dampening factor and due to the iterative architecture of PDE-GNN, for the advection dynamics, errors accumulate over time (Figure 13).

The ablation study shows that MeGA-MP without learning components performs poorly on the advection-reaction dynamics as compared to the one with learning components (Figure 14 and 15) while MeGA-MP without learning components performs equally well on the advection dynamics

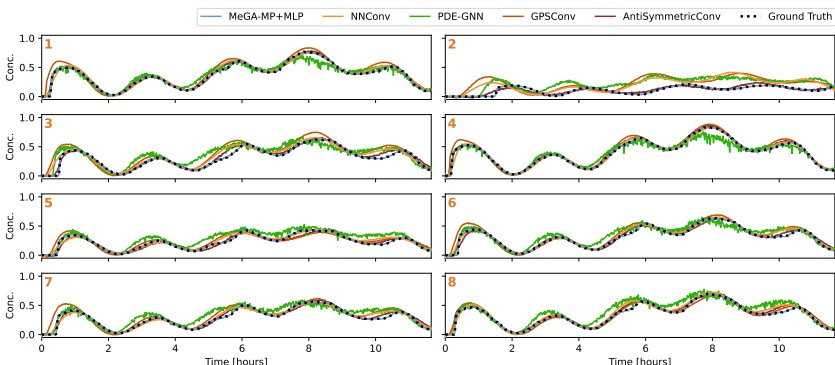

Figure 10: Predictions of all baselines and MeGA-MP for the **advection-reaction** dynamical system at a subset of nodes of the WDS **Hanoi**.

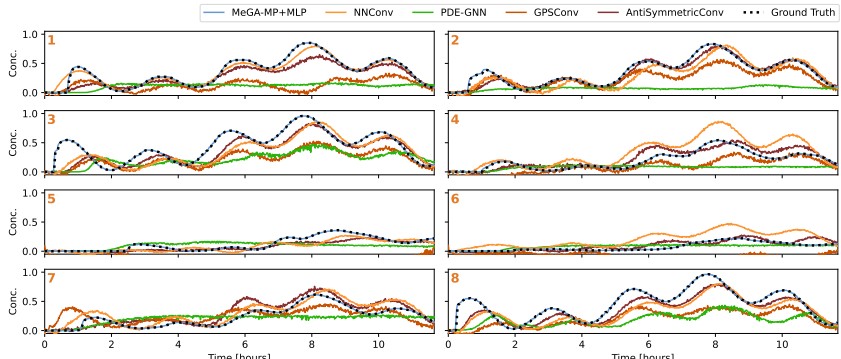

Figure 11: Predictions of all baselines and MeGA-MP for the **advection-reaction** dynamical system at a subset of nodes of the WDS **L-Town**.

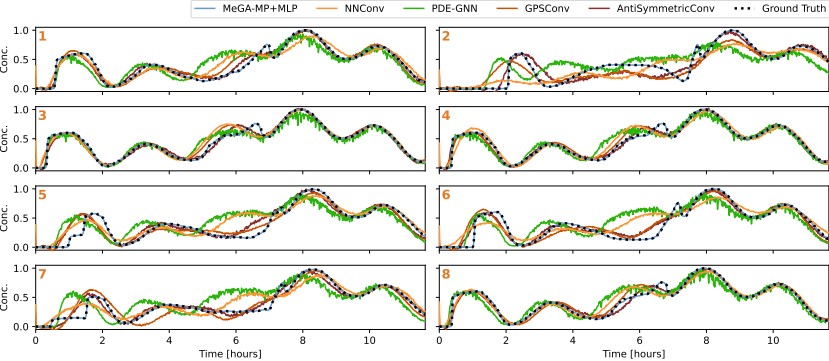

Figure 12: Predictions of all baselines and MeGA-MP for the **advection** dynamical system at a subset of nodes of the WDS **Hanoi**.

as compared to the one with learning components (Figure 16 and 17). This shows that indeed, MeGA-MP does not require any learning components to model advection alone accurately while the learning components are necessary to model more complex, advection-dominated dynamics.

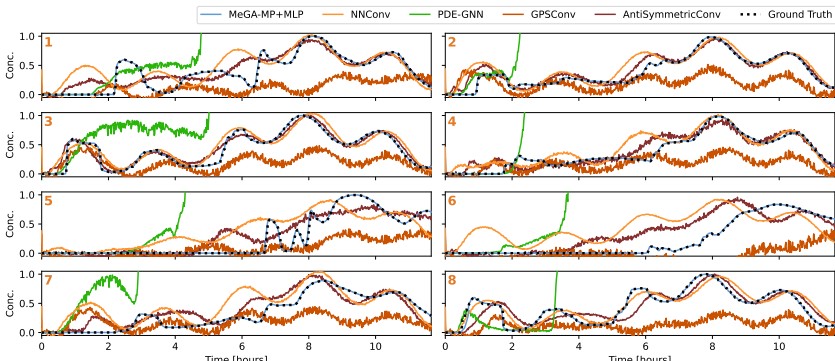

Figure 13: Predictions of all baselines and MeGA-MP for the **advection** dynamical system at a subset of nodes of the WDS **L-Town**.

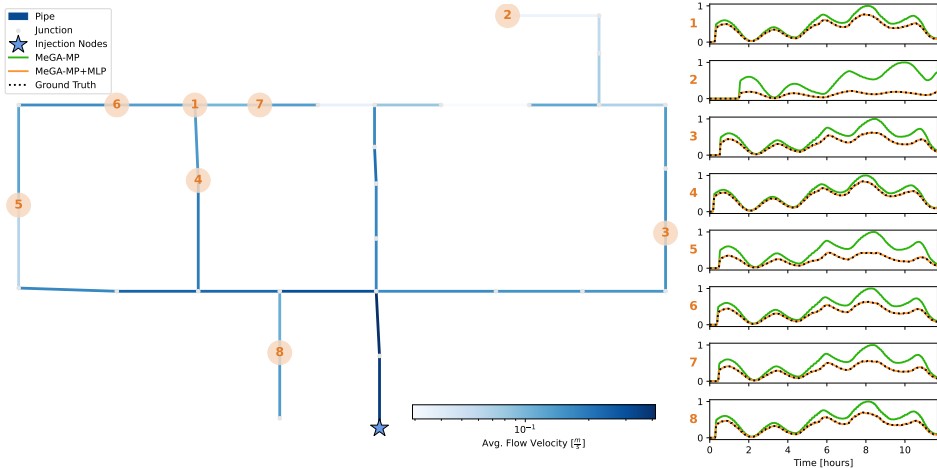

Figure 14: Predictions of MeGA-MP with and without a reaction MLP for the **advection-reaction** dynamical system at a subset of nodes of the WDS **Hanoi**. Hanoi is the graph on which all models are trained. Edge color corresponds to the flow velocity averaged across all time steps of the sample.

### D.2 EXPERIMENT 2: ADVECTION ON A 1D EUCLIDEAN DOMAIN

**Domain and Dataset** The boundary value problem which we analyse in Section 5.2 is given by

$$\frac{\partial c(t,z)}{\partial t} = -\nu(t)\,\frac{\partial c(t,z)}{\partial z} \quad \text{with} \quad \nu(t) = 0.3 \cdot \sin\left(\frac{2\pi t}{100}\right) + 0.3 \text{ and}$$
$$c(t,0) = \exp\left(\frac{1}{2}\left(\frac{t - \mu_t}{\sigma_t}\right)^2\right) \tag{34}$$

for all $t \in [0, 100]$ and $z \in [0, 100]$ and with $\mu_t = 38$ and $\sigma_t = 18$. Its analytical solution is computed analogously to the solution of the initial value problem with boundary condition, as discussed next.

Additionally, in this Appendix, we analyze the initial value problem with boundary condition given by

$$\frac{\partial c(t,z)}{\partial t} = -\nu(t)\,\frac{\partial c(t,z)}{\partial z} \quad \text{with} \quad \nu(t) = 0.3 \cdot \sin\left(\frac{2\pi t}{100}\right) + 0.3 \text{ and}$$
$$c(0,z) = \exp\left(\frac{1}{2}\left(\frac{z - \mu_z}{\sigma_z}\right)^2\right), \quad c(t,0) = \exp\left(\frac{1}{2}\left(\frac{t - \mu_t}{\sigma_t}\right)^2\right) \tag{35}$$

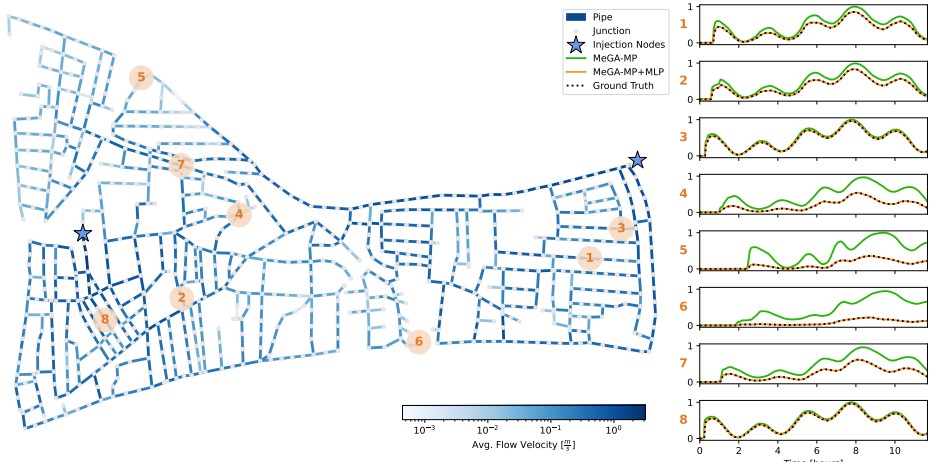

Figure 15: Predictions of MeGA-MP with and without a reaction MLP for the **advection-reaction** dynamical system at a subset of nodes of the WDS **L-Town**. The L-Town graph is not part of the training set. Edge color corresponds to the flow velocity averaged across all time steps of the sample.

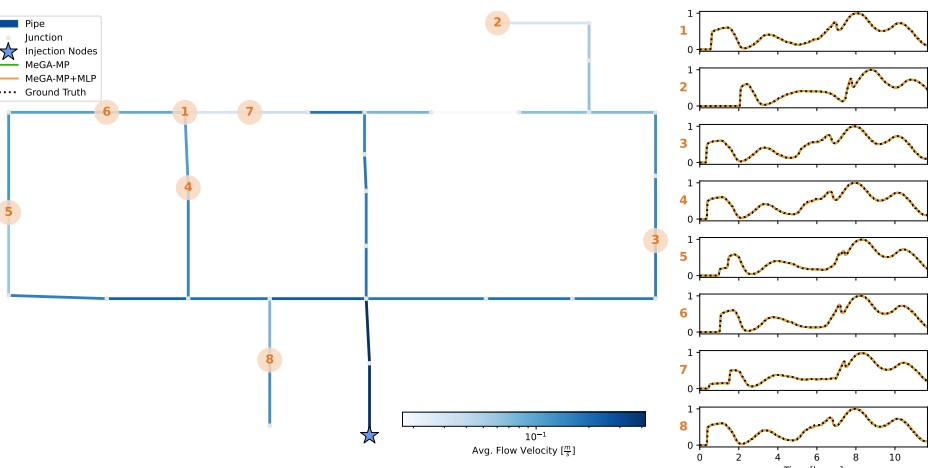

Figure 16: Predictions of of MeGA-MP with and without a reaction MLP for the **advection** dynamical system at a subset of nodes of the WDS **Hanoi**. Hanoi is the graph on which all models are trained. Edge color corresponds to the flow velocity averaged across all time steps of the sample.

for all $t \in [0, 100]$ and $z \in [0, 100]$ and with $\mu_z = 8$, $\sigma_z = 2.5$, $\mu_t = 38$ and $\sigma_t = 18$.

To compute an analytical solution, we can solve the initial value problem $c_{ivp}(t, z)$ and the boundary value problem $c_{bvp}(t, z)$ separately and sum the individual results. This decomposition is possible for $\nu(t) > 0$, where characteristic lines do not cross and each point of the solution corresponds to a value of either the initial value function or the boundary value function.

The initial value problem is solved by integrating the velocity field to compute the spatial shift of the Gaussian $c(0, z)$ for each $t \in [0, 100]$, via $t \longmapsto z_0(t) := \int_0^t \nu(s) \, ds$. The antiderivative of $\nu(t)$ is given by

$$\bar{\nu}(t) = 0.3 \left( t - \frac{\cos(4\pi t)}{4\pi} + \frac{1}{4\pi} \right),$$

and the solution of the initial value problem is $c_{ivp}(t, z) = c(0, z + z_0(t))$.

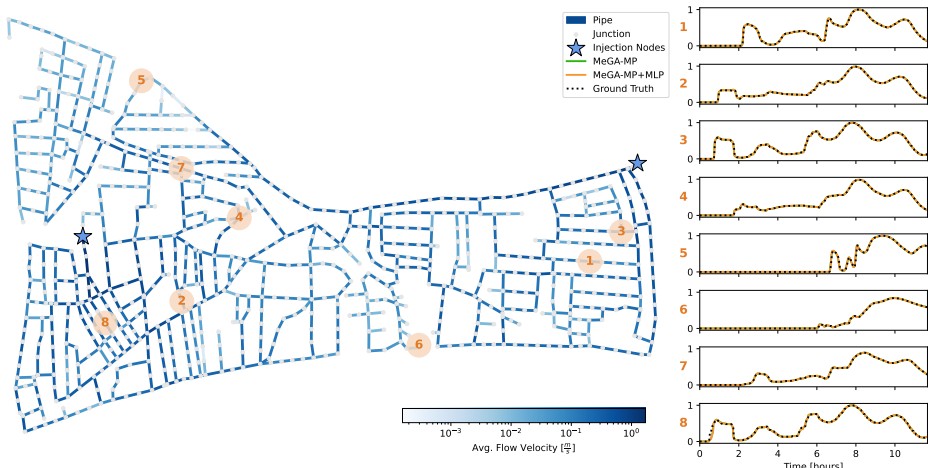

Figure 17: Predictions of of MeGA-MP with and without a reaction MLP for the **advection** dynamical system at a subset of nodes of the WDS **L-Town**. The L-Town graph is not part of the training set. Edge color corresponds to the flow velocity averaged across all time steps of the sample.

To solve the boundary value problem, we find the transport time $\delta t(z)$ from the boundary at $z = 0$ to every other point $z$ in the domain. We leverage the same procedure as for our MeGA-MP (cf. Eq. 8) and compute $\delta t(z)$ as the unique solution of $\int_0^{\delta t(z)} \nu(s)ds = z$. Since $\bar{\nu}(t)$ is strictly increasing the solution is unique and well-defined. According to Theorem C.11, the coordinate $(t + \delta t(z), z)$ lies on the characteristic curve that intersects the boundary at $(t, 0)$. We can shift the boundary value Gaussian accordingly and obtain $c_{bvp}(t, z) = c(t + \delta t(z), 0)$. The solution to the initial value problem with boundary condition is then

$$c(t, z) = c_{ivp}(t, z) + c_{bvp}(t, z)$$

In practice, we compute the respective values numerically at each discrete spatial and temporal grid point.

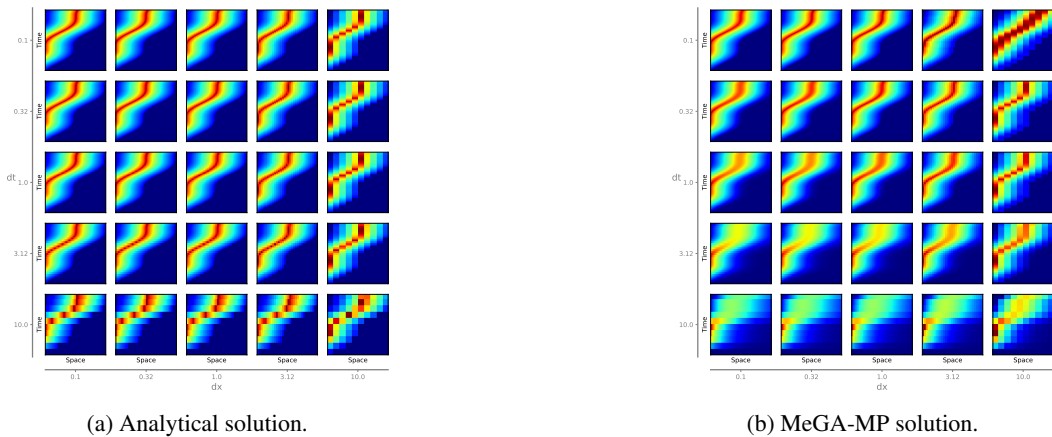

(a) Analytical solution.    (b) MeGA-MP solution.

Figure 18: Solutions of the boundary value problem (34) for different temporal and spatial discretizations and on the whole resulting grid

**Baselines and Results**  As an addition to Figure 4, in Figure 18, the analytical solution of the boundary value problem, which serves as ground truth, and the MeGA-MP solution are visualized at different discretizations and on the whole resulting grid. Moreover, in Figure 19, the analytical solution of the initial value problem with boundary values, which serves as ground truth, and the MeGA-MP solution are visualized at different discretizations and on the whole resulting grid.

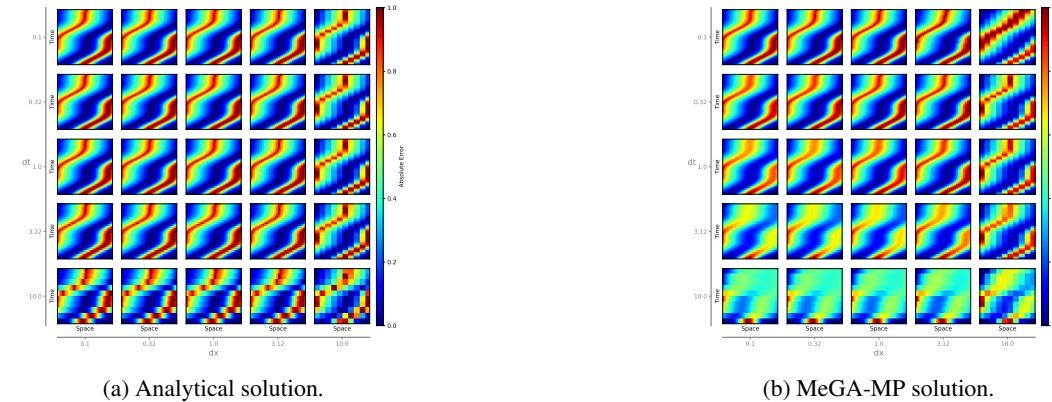

(a) Analytical solution.

(b) MeGA-MP solution.

Figure 19: Solutions of initial value problem with boundary condition (35) for different temporal and spatial discretizations and on the whole resulting grid.

Next to the analytical solution, the numerical and analytical solvers we compare ourselves to are:

- **Runge-Kutta-4** is the classical order-four Runge-Kutta (RK4) numerical method for solving PDEs (and ODEs) of the kind

$$\frac{\partial c(t, z)}{\partial t} = f(t, z, c)$$

with the update rule

$$c(t_{n+1}, z) = c(t_n, z) + \frac{dt}{6} \left( k_1 + 2k_2 + 2k_3 + k_4 \right)$$

where $n \in \mathbb{N}$ is the iteration step, $dt$ is the step-size and $k_1 = f(t_n, z, c)$, $k_2 = f(t_n + \frac{dt}{2}, c(t_n, z) + \frac{dt}{2} k_1)$, $k_3 = f(t_n + \frac{dt}{2}, c(t_n, z) + \frac{dt}{2} k_2)$ and $k_4 = f(t_n + dt, c(t_n, z) + dt k_3)$.

- **Semi-Lagrangian scheme**: We use a custom semi-Lagrangian solver. For every grid point $z \in \Omega$ and iteration step $n + 1$, the method computes the departure point $z'$ of a particle arriving at $z$ by tracing backward along the velocity field $\nu(t, z)$. Since the departure point $z'$ is a continuous coordinate, we interpolate spatially to compute the new solution $c(t_{n+1}, z)$. If we assume a spatially constant and temporally piecewise constant flow field, the departure point is simply $z' = z - dt \cdot \nu(t_n, z)$. Otherwise, it is approximated by integrating the characteristic ODE $\frac{dz}{dt} = -\nu(t, z)$ via the RK4 method. The result is then $c(t_{n+1}, z) = \mathcal{I}[c(t_n, \cdot)](z')$, where $\mathcal{I}$ is the spatial interpolation operator like the linear interpolation in Eq. (15).

As an extension of Table 2, in Table 5, we compare our method to these numerical solvers at different discretizations. We also report the runtime of each method in Table 6. It can be observed that a larger number of edges (smaller $dx$) results in longer runtime, which is expected. Note that these values are only meant to convey a rough idea of time complexity since there may be several ways to improve the performance of the baselines. We ran all methods on the CPU.

Additionally, in Figure 20, we visualize the solutions of the initial value problem with boundary conditions (35) for different solvers and different temporal and spatial discretizations at the last step $t = 100$.

Table 5: Mean absolute error in-between analytical ground truth and solver solution over all temporal and spatial grid points.

| $dt$ | $dx$ | semi- Lagr. | RK4 | MeGA-MP |
|---|---|---|---|---|
| 0.1 | 0.10 | **0.0013**$_{\pm0.0050}$ | – | 0.0025$_{\pm0.0080}$ |
| | 0.32 | 0.0043$_{\pm0.0114}$ | **0.0013**$_{\pm0.0042}$ | 0.0014$_{\pm0.0057}$ |
| | 1.00 | 0.0106$_{\pm0.0226}$ | 0.0032$_{\pm0.0073}$ | **0.0027**$_{\pm0.0082}$ |
| | 3.12 | 0.0214$_{\pm0.0419}$ | **0.0104**$_{\pm0.0152}$ | 0.0174$_{\pm0.0289}$ |
| | 10.00 | 0.0461$_{\pm0.0802}$ | **0.0311**$_{\pm0.0425}$ | 0.1099$_{\pm0.1435}$ |
| 0.32 | 0.10 | 0.0021$_{\pm0.0058}$ | – | 0.0074$_{\pm0.0160}$ |
| | 0.32 | **0.0029**$_{\pm0.0076}$ | 0.0042$_{\pm0.0093}$ | 0.0053$_{\pm0.0121}$ |
| | 1.00 | 0.0085$_{\pm0.0181}$ | 0.0041$_{\pm0.0080}$ | **0.0033**$_{\pm0.0092}$ |
| | 3.12 | 0.0201$_{\pm0.0394}$ | 0.0110$_{\pm0.0159}$ | **0.0037**$_{\pm0.0126}$ |
| | 10.00 | 0.0450$_{\pm0.0790}$ | 0.0321$_{\pm0.0427}$ | **0.0062**$_{\pm0.0174}$ |
| 1.0 | 0.10 | 0.0117$_{\pm0.0175}$ | – | 0.0161$_{\pm0.0290}$ |
| | 0.32 | 0.0112$_{\pm0.0161}$ | – | 0.0152$_{\pm0.0269}$ |
| | 1.00 | 0.0116$_{\pm0.0159}$ | **0.0106**$_{\pm0.0137}$ | 0.0120$_{\pm0.0199}$ |
| | 3.12 | 0.0197$_{\pm0.0317}$ | 0.0101$_{\pm0.0147}$ | **0.0093**$_{\pm0.0162}$ |
| | 10.00 | 0.0433$_{\pm0.0767}$ | 0.0276$_{\pm0.0397}$ | **0.0106**$_{\pm0.0198}$ |
| 3.12 | 0.10 | 0.0366$_{\pm0.0444}$ | – | 0.0371$_{\pm0.0530}$ |
| | 0.32 | **0.0360**$_{\pm0.0430}$ | – | 0.0363$_{\pm0.0518}$ |
| | 1.00 | 0.0343$_{\pm0.0403}$ | – | **0.0342**$_{\pm0.0485}$ |
| | 3.12 | 0.0321$_{\pm0.0391}$ | 0.0303$_{\pm0.0310}$ | **0.0280**$_{\pm0.0387}$ |
| | 10.00 | 0.0424$_{\pm0.0673}$ | 0.0339$_{\pm0.0416}$ | **0.0235**$_{\pm0.0326}$ |
| 10.0 | 0.10 | 0.1106$_{\pm0.1265}$ | – | **0.0876**$_{\pm0.1026}$ |
| | 0.32 | 0.1100$_{\pm0.1255}$ | – | **0.0871**$_{\pm0.1021}$ |
| | 1.00 | 0.1080$_{\pm0.1223}$ | – | **0.0856**$_{\pm0.1005}$ |
| | 3.12 | 0.1028$_{\pm0.1174}$ | – | **0.0803**$_{\pm0.0938}$ |
| | 10.00 | 0.0894$_{\pm0.1091}$ | 0.1332$_{\pm0.1345}$ | **0.0656**$_{\pm0.0788}$ |

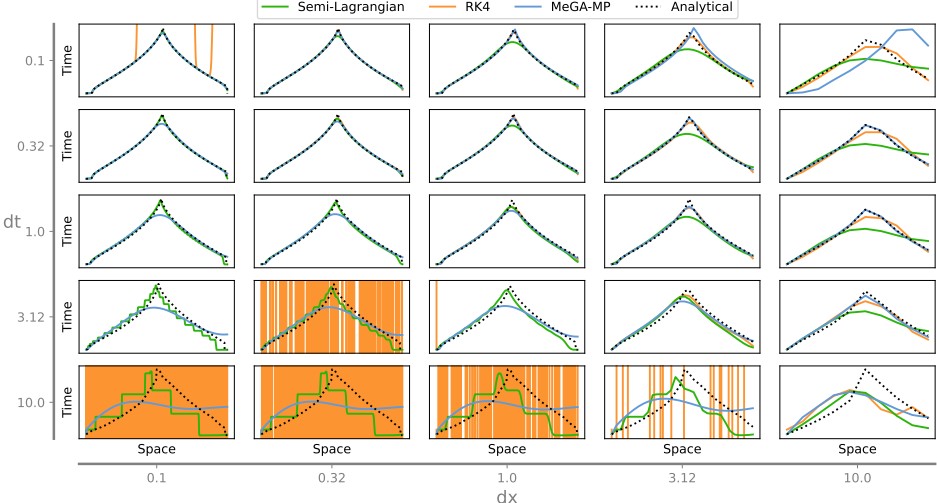

Figure 20: Solutions of the initial value problem with boundary condition (34) for different solvers and different temporal and spatial discretizations at the last step $t = 100$.

Table 6: Time in seconds that it takes each method to solve the BVP. The time it takes the preprocessing to convert flows into transport times (s. Algorithm 1) is included in these values.

| $dt$ | $dx$ | semi- Lagr. | RK4 | MeGA-MP | Analytical |
|------|------|-------------|------|---------|------------|
| 0.1 | 0.10 | 69.0965 | – | **28.1989** | 0.4697 |
| | 0.32 | 21.4509 | **1.3616** | 5.2283 | 0.3434 |
| | 1.00 | 6.6336 | **1.2051** | 1.2221 | 0.3255 |
| | 3.12 | 2.2019 | 1.1867 | **0.4049** | 0.3486 |
| | 10.00 | 0.7894 | 1.1848 | **0.2116** | 0.2757 |
| 0.32 | 0.10 | 22.2474 | – | **19.2572** | 0.0350 |
| | 0.32 | 6.7440 | **0.4245** | 3.5078 | 0.0092 |
| | 1.00 | 2.0956 | **0.3805** | 0.4689 | 0.0024 |
| | 3.12 | 0.7039 | 0.3802 | **0.1126** | 0.0011 |
| | 10.00 | 0.2494 | 0.3666 | **0.0309** | 0.0007 |
| 1.0 | 0.10 | **7.0511** | – | 7.2094 | 0.0051 |
| | 0.32 | 2.0747 | – | **1.5679** | 0.0019 |
| | 1.00 | 0.6643 | **0.1193** | 0.3306 | 0.0010 |
| | 3.12 | 0.2225 | 0.1175 | **0.0701** | 0.0006 |
| | 10.00 | 0.0781 | 0.1160 | **0.0251** | 0.0005 |
| 3.12 | 0.10 | **2.1983** | – | 5.0147 | 0.0019 |
| | 0.32 | **0.6536** | – | 0.8929 | 0.0009 |
| | 1.00 | 0.2034 | – | **0.1936** | 0.0007 |
| | 3.12 | 0.0684 | **0.0368** | 0.0582 | 0.0004 |
| | 10.00 | 0.0246 | 0.0361 | **0.0231** | 0.0004 |
| 10.0 | 0.10 | **0.6470** | – | 2.9019 | 0.0010 |
| | 0.32 | **0.1932** | – | 0.5289 | 0.0006 |
| | 1.00 | **0.0601** | – | 0.1365 | 0.0004 |
| | 3.12 | **0.0202** | – | 0.0450 | 0.0006 |
| | 10.00 | **0.0074** | 0.0111 | 0.0189 | 0.0004 |

# E  PROOFS

## E.1  PROOF OF LEMMA 3.1

**Lemma** (Lemma 3.1). *Let $v \in V$ and $u \in \mathcal{N}(v)$ hold. If the function $c_{uv}$ obeys Equation (21), for each $t_0 \in \mathbb{R}$ and $z_0 \in \mathbb{R}$, the curve*

$$\gamma_{t_0,z_0} : T_{t_0,z_0} \longrightarrow \mathbb{R}_{\geq 0}$$
$$t \longmapsto c_{uv}(t, z_0 + \int_{t_0}^{t} \nu_{uv}(s)\, ds)$$

*is constant on*

$$T_{t_0,z_0} = \left\{ t \in \mathbb{R} \mid 0 \leq z_0 + \int_{t_0}^{t'} \nu_{uv}(s)\, ds \leq l_{uv} \ \forall t' \in \big[\min\{t_0,t\}, \max\{t_0,t\}\big] \right\}.$$

*Proof.* Let $c_{uv} : \mathbb{R} \times (0, l_{uv}) \to \mathbb{R}_{\geq 0}$ obey Equation (21) (note that Equation (21) is only defined for $c_{uv|(0,l_{uv})} : \mathbb{R} \times (0, l_{uv}) \to \mathbb{R}_{\geq 0}$). We can consider its space coordinate $z$ as a function $z : \mathbb{R} \to (0, l_{uv})$ parametrized by time. Consequently, we can use the method of characteristics (cf. (Habermann, 2013)) to derive the following system of ODEs from Equation (21) for the composed, now only time-dependent function $c_{uv} : \mathbb{R} \to \mathbb{R}_{\geq 0}, c_{uv}(t) := c_{uv}(t, z(t))$:

$$\frac{dc_{uv}(t)}{dt} = 0, \quad \frac{dz(t)}{dt} = \nu_{uv}(t), \quad \frac{dt}{dt} = 1.$$

By the fundamental theorem of calculus, we observe that for each $t_0 \in \mathbb{R}$ and $z_0 \in \mathbb{R}$, the function

$$z : \mathbb{R} \longrightarrow \mathbb{R}, \ t \longmapsto z(t) := z_0 + \int_{t_0}^{t} \nu_{uv}(s)\, ds$$

satisfies the second ODE. Therefore, by the first ODE, we can conclude that the function $c_{uv}$ is constant along characteristic curves $(t, z(t)) = (t, z_0 + \int_{t_0}^t \nu_{uv}(s)\, ds)$ along the time $\mathbb{R}$ as well as along the space interval $(0, l_{uv})$, i.e., *within* the edge $e_{uv}$ of length $l_{uv} > 0$. Therefore, taking into account the domain and co-domain of $c_{uv}$, $\nu_{uv}$ and $z$ as a function, $t_0, t \in \mathbb{R}$ and $z(t^{'}) \in (0, l_{uv})$ need to hold for all $t^{'} \in [\min\{t_0, t\}, \max\{t_0, t\}]$.[15]

In summary, the curve $\gamma_{z_0} : T_{t_0, z_0} \to \mathbb{R}_{\geq 0}, t \mapsto c_{uv}(t, z_0 + \int_{t_0}^t \nu_{uv}(s)\, ds)$ is constant on

$$T_{t_0, z_0} = \left\{ t \in \mathbb{R} \mid 0 < z_0 + \int_{t_0}^{t^{'}} \nu_{uv}(s)\, ds < l_{uv} \,\, \forall t^{'} \in \big[ \min\{t_0, t\}, \max\{t_0, t\} \big] \right\}.$$

$\square$

### E.2 PROOF OF THEOREM B.1

**Theorem** (Theorem B.1). *Let $\mathcal{T} = (A, E)$ be a finite tree with root node $a_0 \in A$, **int**ernal nodes $A_{leaf}$ and **leaf** nodes $A_{leaf}$ (that is, $A = \{a_0\} \sqcup A_{int} \sqcup A_{leaf}$ holds). If each node $a \in A$ is associated with a magnitude $x_a \in \mathbb{R}$ that satisfies*

$$x_a = \sum_{b \in \mathcal{C}(a)} \omega_{ab} \cdot x_b \text{ for all } a \in \{a_0\} \sqcup A_{int}, \tag{36}$$

*where the set $\mathcal{C}(a) \subset A$ consists of the children of $a \in A$ and $w_{ab}$ is a weight of the edge $e_{ab} \in E$ that depends on both the parent $a \in A$ and its child $b \in \mathcal{C}(a)$, then the magnitude $x_0 := x_{a_0}$ of the root node $a_0 \in A$ is determined by the magnitudes $x_{l_p} := x_{a_{l_p}}$ of the leaf nodes $a_{l_p} \in A_{leaf}$ by*

$$x_0 = \sum_{p = (a_0, \ldots, a_{l_p}) \in \mathcal{P}(a_0, A_{leaf})} \omega(a_0, \ldots, a_{l_p}) \cdot x_{a_{l_p}}, \tag{37}$$

*where $\mathcal{P}(a_0, A_{leaf})$ be the set of paths connecting the root node $a_0$ to one of its leaf nodes $A_{leaf}$ and*

$$\omega(a_0, \ldots, a_{l_p}) := \prod_{l=0}^{l_p - 1} \omega_{a_l a_{l+1}} \tag{38}$$

*is the product of weights along a(n) (arbitrary) path $p = (a_0, \ldots, a_{l_p}) \in \mathcal{P}$ with length $l_p \in \mathbb{N}$.*

*Proof.* For the sake of brevity, in this proof, we use the abbreviation $\mathcal{P} = \mathcal{P}(a_0, A_{\text{leaf}})$.

*Step 1: For each $l \in \mathbb{N}$, we obtain*

$$x_{a_0} = \sum_{\substack{p = (a_0, \ldots, a_{l_p}) \in \mathcal{P}: \\ l_p \leq l}} \omega(a_0, \ldots, a_{l_p}) \cdot x_{a_{l_p}} + \sum_{\substack{p = (a_0, \ldots, a_{l_p}) \in \mathcal{P}: \\ l_p > l}} \omega(a_0, \ldots, a_l) \cdot x_{a_l}. \tag{39}$$

We prove the claim by induction to $l \in \mathbb{N}$.
*Induction base:* If $l = 1$, by (1) Equation (36) applied to $x_{a_0}$, (2) basic transformations, (3) a change in notation, (4) the definition of $\omega$ (cf. (38)) and (5) the choice of $l = 1$, we obtain

$$x_{a_0} \overset{(1)}{=} \sum_{b \in \mathcal{C}(a_0)} \omega_{a_0 b} \cdot x_b$$

$$\overset{(2,3)}{=} \sum_{a_1 \in \mathcal{C}(a_0) \cap A_{\text{leaf}}} \omega_{a_0 a_1} \cdot x_{a_1} + \sum_{a_1 \in \mathcal{C}(a_0) \cap A_{\text{int}}} \omega_{a_0 a_1} \cdot x_{a_1}$$

$$\overset{(2)}{=} \sum_{\substack{p = (a_0, a_1) \in \mathcal{P}: \\ l_p = 1}} \omega_{a_0 a_1} \cdot x_{a_1} + \sum_{\substack{p = (a_0, a_1, \ldots, a_{l_p}) \in \mathcal{P}: \\ l_p > 1}} w_{a_0 a_1} \cdot x_{a_1}$$

---

[15]Especially to mention, the requirement $z_0 + \int_{t_0}^t \nu_{uv}(s)\, ds \in (0, l_{uv})$ does not suffices, cf. remark C.3.

$$\overset{(4)}{=} \sum_{\substack{p=(a_0,a_1)\in\mathcal{P}: \\ l_p=1}} \omega(a_0,a_1) \cdot x_{a_1} + \sum_{\substack{p=(a_0,...,a_{l_p})\in\mathcal{P}: \\ l_p>1}} \omega(a_0,a_1) \cdot x_{a_l}$$

$$\overset{(2,5)}{=} \sum_{\substack{p=(a_0,...,a_{l_p})\in\mathcal{P}: \\ l_p\leq l}} \omega(a_0,...,a_{l_p}) \cdot x_{a_{l_p}} + \sum_{\substack{p=(a_0,...,a_{l_p})\in\mathcal{P}: \\ l_p>l}} \omega(a_0,...,a_l) \cdot x_{a_l}.$$

*Induction hypothesis:* Equation (39) holds for all $\tilde{l} \leq l \in \mathbb{N}$.

*Induction step:* By (1) the induction hypothesis, we obtain

$$x_{a_0} \overset{(1)}{=} \sum_{\substack{p=(a_0,...,a_{l_p})\in\mathcal{P}: \\ l_p\leq l}} \omega(a_0,...,a_{l_p}) \cdot x_{a_{l_p}} + \sum_{\substack{p=(a_0,...,a_{l_p})\in\mathcal{P}: \\ l_p>l}} \omega(a_0,...,a_l) \cdot x_{a_l}.$$

Similar to the induction base, by (1) Equation (36) applied to $x_{a_l}$, (note that since $l < l_p$, $a_l \in A_{\mathrm{int}}$ must hold), (2) basic transformations, (3) a change in notation and (4) the definition of $\omega$ (cf. (38)), we transform the second summand to

$$\sum_{\substack{p=(a_0,...,a_{l_p})\in\mathcal{P}: \\ l_p>l}} \omega(a_0,...,a_l) \cdot x_{a_l}$$

$$\overset{(1)}{=} \sum_{\substack{p=(a_0,...,a_{l_p})\in\mathcal{P}: \\ l_p>l}} \omega(a_0,...,a_l) \cdot \left( \sum_{b\in\mathcal{C}(a_l)} \omega_{a_l b} \cdot x_b \right)$$

$$\overset{(2,3)}{=} \sum_{\substack{p=(a_0,...,a_{l_p})\in\mathcal{P}: \\ l_p\geq l+1}} \omega(a_0,...,a_l) \cdot \left( \sum_{a_{l+1}\in\mathcal{C}(a_l)\cap A_{\mathrm{leaf}}} \omega_{a_l a_{l+1}} \cdot x_{a_{l+1}} + \sum_{a_{l+1}\in\mathcal{C}(a_l)\cap A_{\mathrm{in}}} \omega_{a_l a_{l+1}} \cdot x_{a_{l+1}} \right)$$

$$\overset{(2)}{=} \sum_{\substack{p=(a_0,...,a_{l_p})\in\mathcal{P}: \\ l_p\geq l+1}} \sum_{a_{l+1}\in\mathcal{C}(a_l)\cap A_{\mathrm{leaf}}} \omega(a_0,...,a_l) \cdot \omega_{a_l a_{l+1}} \cdot x_{a_{l+1}}$$

$$+ \sum_{\substack{p=(a_0,...,a_{l_p})\in\mathcal{P}: \\ l_p\geq l+1}} \sum_{a_{l+1}\in\mathcal{C}(a_l)\cap A_{\mathrm{in}}} \omega(a_0,...,a_l) \cdot \omega_{a_l a_{l+1}} \cdot x_{a_{l+1}}$$

$$\overset{(2)}{=} \sum_{\substack{p=(a_0,...,a_{l_p})\in\mathcal{P}: \\ l_p=l+1}} \omega(a_0,...,a_l) \cdot \omega_{a_l a_{l+1}} \cdot x_{a_{l+1}}$$

$$+ \sum_{\substack{p=(a_0,...,a_{l_p})\in\mathcal{P}: \\ l_p>l+1}} \omega(a_0,...,a_l) \cdot \omega_{a_l a_{l+1}} \cdot x_{a_{l+1}}$$

$$\overset{(4)}{=} \sum_{\substack{p=(a_0,...,a_{l_p})\in\mathcal{P}: \\ l_p=l+1}} \omega(a_0,...,a_l,a_{l+1}) \cdot x_{a_{l+1}} + \sum_{\substack{p=(a_0,...,a_{l_p})\in\mathcal{P}: \\ l_p>l+1}} \omega(a_0,...,a_l,a_{l+1}) \cdot x_{a_{l+1}}.$$

Bringing it all together, again by (1) basic transformations, we obtain

$$x_{a_0} = \sum_{\substack{p=(a_0,...,a_{l_p})\in\mathcal{P}: \\ l_p\leq l}} \omega(a_0,...,a_{l_p}) \cdot x_{a_{l_p}} + \sum_{\substack{p=(a_0,...,a_{l_p})\in\mathcal{P}: \\ l_p>l}} \omega(a_0,...,a_l) \cdot x_{a_l}$$

$$= \sum_{\substack{p=(a_0,...,a_{l_p})\in\mathcal{P}: \\ l_p\leq l}} \omega(a_0,...,a_{l_p}) \cdot x_{a_{l_p}}$$

$$+ \sum_{\substack{p=(a_0,...,a_{l_p})\in\mathcal{P}: \\ l_p=l+1}} \omega(a_0,...,a_l,a_{l+1}) \cdot x_{a_{l+1}}$$

$$+ \sum_{\substack{p=(a_0,...,a_{l_p})\in\mathcal{P}: \\ l_p>l+1}} \omega(a_0,...,a_l,a_{l+1}) \cdot x_{a_{l+1}}$$

$$\stackrel{(1)}{=} \sum_{\substack{p=(a_0,...,a_{l_p})\in\mathcal{P}: \\ l_p<l+1}} \omega(a_0,...,a_{l_p}) \cdot x_{a_{l_p}}$$

$$+ \sum_{\substack{p=(a_0,...,a_{l_p})\in\mathcal{P}: \\ l_p=l+1}} \omega(a_0,...,a_{l_p}) \cdot x_{a_{l_p}}$$

$$+ \sum_{\substack{p=(a_0,...,a_{l_p})\in\mathcal{P}: \\ l_p>l+1}} \omega(a_0,...,a_{l+1}) \cdot x_{a_{l+1}}$$

$$\stackrel{(1)}{=} \sum_{\substack{p=(a_0,...,a_{l_p})\in\mathcal{P}: \\ l_p\leq l+1}} \omega(a_0,...,a_{l_p}) \cdot x_{a_{l_p}} \quad + \sum_{\substack{p=(a_0,...,a_{l_p})\in\mathcal{P}: \\ l_p>l+1}} \omega(a_0,...,a_{l+1}) \cdot x_{a_{l+1}},$$

which concludes the induction step.

*Step 2: Equation* (37) *holds.*

Since $\mathcal{T}$ is finite, that is, $\mathcal{T}$ an acyclic graph with a finite number of nodes $|A| < \infty$, also each path $p \in \mathcal{P}$ has to be finite. Therefore, we can choose $l = \max_{p\in\mathcal{P}} l_p \in \mathbb{N}$. Consequently, the second sum in *Step 1* is empty and the condition in the first sum in *Step 1* is naturally satisfied. This yields Equation (37). $\qquad\square$

### E.3 PROOF OF THEOREM B.3

**Theorem** (Theorem B.3). *Let $v \in V$ and $t \in \mathbb{R}$ hold. Let $\mathcal{T}(v,t)$ be the inflow tree of $v$ at time $t$ as defined in Definition B.2. If $c_v(t)$ is defined by Equation (11), then*

$$c_v(t) = \sum_{p=((v_0,t_0)=(v,t),(u_1,v_1,t_1)...,(u_{l_p},v_{l_p},t_{l_p}))\in\mathcal{P}(v,t)} \left( \prod_{l=0}^{l_p-1} w_{u_{l+1}v_l}(t_l) \right) \cdot c_{v_{l_p}}(t_{l_p})$$

*with weights $w_{u_{l+1}v_l}(t_l)$ according to Equation (10) holds.*

*Proof.* For each layer $l \in \mathbb{N}_0$ of the tree, we now associate each node $a_l = (u_l, v_l, t_l) \in A$ of that layer with the magnitude $x_l := x_{a_l} := c_{v_l}(t_l)$.

*Step 1: Together with these magnitudes, $\mathcal{T}(v,t)$ defines a tree in terms of Theorem B.1.*

In order to prove the claim, we first of all investigate on the term $\phi(t_l, e_{u_{l+1}v_l}, \mathbf{c}, \boldsymbol{\nu})$ for a given parent node $a_l = (u_l, v_l, t_l)$ in the $l$-th layer and its child node $a_{l+1} = (u_{l+1}, v_{l+1}, t_{l+1}) \in \mathcal{C}(a_l)$.[16] Indeed, by (1) Equation (9) and (2) the definition of the inflow tree $\mathcal{T}(v,t)$ (Definition B.2), for any

---

[16]Especially to mention, by definition of the inflow tree $\mathcal{T}(v,t)$ (Definition B.2), $u_{l+1} \in \mathcal{N}_-(v_t, t_l)$ holds, that is, the edge $e_{u_{l+1}v_l} \in E$ exists.

$l \in \mathbb{N}_0$, $\phi(t_l, e_{u_{l+1}v_l}, \mathbf{c}, \boldsymbol{\nu})$ is equal to

$$\phi(t_l, e_{u_{l+1}v_l}, \mathbf{c}, \boldsymbol{\nu})$$

$$\overset{(1)}{=} \begin{cases} c_{u_{l+1}}(t_l - \mathrm{s}(t_l, e_{u_{l+1}v_l}, \boldsymbol{\nu})) & \text{if } \mathrm{z}(\mathrm{s}(t_l, e_{u_{l+1}v_l}, \boldsymbol{\nu})) \in \{-l_{uv}, l_{uv}\}, \\ c_{v_l}(t_l - \mathrm{s}(t_l, e_{u_{l+1}v_l}, \boldsymbol{\nu})) & \text{if } \mathrm{z}(\mathrm{s}(t_l, e_{u_{l+1}v_l}, \boldsymbol{\nu})) = 0 \end{cases}$$

$$\overset{(2)}{=} \begin{cases} c_{u_{l+1}}(t_{l+1}) & \text{if } \mathrm{z}(\mathrm{s}(t_l, e_{u_{l+1}v_l}, \boldsymbol{\nu})) \in \{-l_{uv}, l_{uv}\}, \\ c_{v_l}(t_{l+1}) & \text{if } \mathrm{z}(\mathrm{s}(t_l, e_{u_{l+1}v_l}, \boldsymbol{\nu})) = 0 \end{cases} \tag{40}$$

$$\overset{(2)}{=} c_{v_{l+1}}(t_{l+1})$$

$$\overset{(2)}{=} x_{l+1}.$$

Consequently, again by (1) the definition of the inflow tree $\mathcal{T}(v, t)$ (Definition B.2), (2) Equation (11) and (3) Equation (40), for any $a_l \in \{a_0\} \sqcup A_{\text{int}}$, we obtain

$$x_{a_l} := x_l \overset{(1)}{=} c_{v_l}(t_l)$$

$$\overset{(2)}{=} \sum_{u_{l+1} \in \mathcal{N}_-(v_l, t_l)} w_{u_{l+1}v_l}(t_l) \cdot \phi(t_l, e_{u_{l+1}v_l}, \mathbf{c}, \boldsymbol{\nu})$$

$$\overset{(3)}{=} \sum_{u_{l+1} \in \mathcal{N}_-(v_l, t_l)} w_{u_{l+1}v_l}(t_l) \cdot x_{l+1}$$

$$\overset{(1)}{=} \sum_{a_{l+1} \in \mathcal{C}(a_l)} \omega_{a_l a_{l+1}} \cdot x_{a_{l+1}}$$

with weights

$$\omega_{a_l a_{l+1}} := w_{u_{l+1}v_l}(t_l). \tag{41}$$

Consequently, $\mathcal{T}(v, t)$ is a tree in terms of Theorem B.1.

*Step 2: Equation* (18) *holds.*

Again by (1) the definition of the inflow tree $\mathcal{T}(v, t)$ (Definition B.2), (2) Theorem B.1 (which we can apply according to *Step 1*) and (3) Equation (41), we obtain

$$c_v(t) \overset{(1)}{=} c_{v_0}(t_0) \overset{(1)}{=} x_0$$

$$\overset{(2)}{=} \sum_{p=(a_0,...,a_{l_p}) \in \mathcal{P}(a_0, A_{\text{leaf}})} \omega(a_0, ..., a_{l_p}) \cdot x_{l_p}$$

$$\overset{(1,3)}{=} \sum_{p=((v_0,t_0)=(v,t),(u_1,v_1,t_1)...,(u_{l_p},v_{l_p},t_{l_p})) \in \mathcal{P}(v,t)} \left( \prod_{l=0}^{l_p-1} w_{u_{l+1}v_l}(t_l) \right) \cdot c_{v_{l_p}}(t_{l_p}),$$

which was to show. $\qquad\square$

### E.4 PROOF OF THEOREM B.4

**Theorem** (Theorem B.4). *Let $v \in V$ and $t \in T_{hi} \cup T_{pr}$ hold. Let $\hat{\mathcal{T}}(v, t)$ be the interpolation inflow tree of $v$ at time $t$ as defined in Definition B.2. Let $k \geq \max_{p \in \hat{\mathcal{P}}(v,t)} l_p$ hold. If $\hat{c}_v^{(k)}(t)$ is defined by Equation* (12), *then*

$$\hat{c}_v^{(k)}(t) = \sum_{p=((v_0,t_0)=(v,t),(u_1,v_1,t_1)...,(u_{l_p},v_{l_p},t_{l_p})) \in \hat{\mathcal{P}}(v,t)} \left( \prod_{l=0}^{l_p-1} w_{u_{l+1}v_l}(t_l) \cdot a(t_{l+1}) \right) \cdot \hat{c}_{v_{l_p}}^{(k-l_p)}(t_{l_p})$$

*with weights $w_{u_{l+1}v_l}(t_l)$ and $a(t_{l+1})$ according to Equation* (10) *and* (15), *respectively, holds.*

*Proof.* Similar to the proof of Theorem B.3, for each layer $l \in \mathbb{N}_0$ of the tree, we now associate each node $a_l = (u_l, v_l, t_l) \in A$ of that layer with the magnitude $x_l := x_{a_l} := \hat{c}_{v_l}^{(k-l)}(t_l)$.

*Step 1: Together with these magnitudes, $\hat{\mathcal{T}}(v, t)$ defines a tree in terms of Theorem B.1.*

In order to prove the claim, we first of all investigate on the term $\varphi(t_l, e_{u_{l+1}v_l}, \hat{\mathbf{c}}^{(k-(l+1))}, \boldsymbol{\nu})$ for a given parent node $a_l = (u_l, v_l, t_l)$ in the $l$-th layer and its child node $a_{l+1} = (u_{l+1}, v_{l+1}, t_{l+1}) \in \mathcal{C}(a_l)$. Indeed, by (1) Equation (16), for any $l \in \mathbb{N}_0$, $\varphi(t_l, e_{u_{l+1}v_l}, \hat{\mathbf{c}}^{(k-(l+1))}, \boldsymbol{\nu})$ is equal to

$$
\varphi(t_l, e_{u_{l+1}v_l}, \hat{\mathbf{c}}^{(k-(l+1))}, \boldsymbol{\nu})
$$
$$
\overset{(1)}{=} \begin{cases} a(t_j) \cdot \hat{c}_{u_{l+1}}^{(k-(l+1))}(t_j) + a(t_{j+1}) \cdot \hat{c}_{u_{l+1}}^{(k-(l+1))}(t_{j+1}) & \text{if } \mathbb{z}(\mathbb{s}(t_l, e_{u_{l+1}v_l}, \boldsymbol{\nu})) \in \{-l_{uv}, l_{uv}\}, \\ a(t_j) \cdot \hat{c}_{v_l}^{(k-(l+1))}(t_j) + a(t_{j+1}) \cdot \hat{c}_{v_l}^{(k-(l+1))}(t_{j+1}) & \text{if } \mathbb{z}(\mathbb{s}(t_l, e_{u_{l+1}v_l}, \boldsymbol{\nu})) = 0 \end{cases}
$$

for the $j \in I$ for which $t_l - \mathbb{s}(t_l, e_{u_{l+1}v_l}, \boldsymbol{\nu}) \in [t_j, t_{j+1})$ holds. Following (1) the definition of the interpolation inflow tree $\hat{\mathcal{T}}(v, t)$ (Definition B.2), we obtain

$$
\varphi(t_l, e_{u_{l+1}v_l}, \hat{\mathbf{c}}^{(k-(l+1))}, \boldsymbol{\nu})
$$
$$
\overset{(1)}{=} \begin{cases} \sum_{\substack{t_{l+1} \in \{t_j, t_{j+1}\}: \\ t_l - \mathbb{s}(t_l, e_{u_{l+1}v_l}, \boldsymbol{\nu}) \in [t_j, t_{j+1})}} a(t_{l+1}) \cdot \hat{c}_{u_{l+1}}^{(k-(l+1))}(t_{l+1}) & \text{if } \mathbb{z}(\mathbb{s}(t_l, e_{u_{l+1}v_l}, \boldsymbol{\nu})) \in \{-l_{uv}, l_{uv}\}, \\ \sum_{\substack{t_{l+1} \in \{t_j, t_{j+1}\}: \\ t_l - \mathbb{s}(t_l, e_{u_{l+1}v_l}, \boldsymbol{\nu}) \in [t_j, t_{j+1})}} a(t_{l+1}) \cdot \hat{c}_{v_l}^{(k-(l+1))}(t_{l+1}) & \text{if } \mathbb{z}(\mathbb{s}(t_l, ee_{u_{l+1}v_l}, \boldsymbol{\nu})) = 0 \end{cases}
$$
$$
\overset{(1)}{=} \sum_{\substack{t_{l+1} \in \{t_j, t_{j+1}\}: \\ t_l - \mathbb{s}(t_l, e_{u_{l+1}v_l}, \boldsymbol{\nu}) \in [t_j, t_{j+1})}} a(t_{l+1}) \cdot \hat{c}_{v_{l+1}}^{(k-(l+1))}(t_{l+1}) \tag{42}
$$
$$
\overset{(1)}{=} \sum_{\substack{t_{l+1} \in \{t_j, t_{j+1}\}: \\ t_l - \mathbb{s}(t_l, e_{u_{l+1}v_l}, \boldsymbol{\nu}) \in [t_j, t_{j+1})}} a(t_{l+1}) \cdot x_{l+1}.
$$

Consequently, again by (1) the definition of the interpolation inflow tree $\mathcal{T}(v, t)$ (Definition B.2), (2) Equation (12), (3) Equation (42) and (4) basic transformations, for any $a_l \in \{a_0\} \sqcup A_{\text{int}}$, we obtain

$$
x_{a_l} := x_l \overset{(1)}{=} \hat{c}_{v_l}^{(k-l)}(t_l)
$$
$$
\overset{(2)}{=} \sum_{u_{l+1} \in \mathcal{N}_-(v_l, t_l)} w_{u_{l+1}v_l}(t_l) \cdot \varphi(t_l, e_{v_{l+1}v_l}, \hat{\mathbf{c}}^{(k-(l+1))}, \boldsymbol{\nu})
$$
$$
\overset{(3)}{=} \sum_{u_{l+1} \in \mathcal{N}_-(v_l, t_l)} w_{u_{l+1}v_l}(t_l) \cdot \left( \sum_{\substack{t_{l+1} \in \{t_j, t_{j+1}\}: \\ t_l - \mathbb{s}(t_l, e_{u_{l+1}v_l}, \boldsymbol{\nu}) \in [t_j, t_{j+1})}} a(t_{l+1}) \cdot x_{l+1} \right)
$$
$$
\overset{(4)}{=} \sum_{u_{l+1} \in \mathcal{N}_-(v_l, t_l)} \sum_{\substack{t_{l+1} \in \{t_j, t_{j+1}\}: \\ t_l - \mathbb{s}(t_l, e_{(l+1)l}, \boldsymbol{\nu}) \in [t_j, t_{j+1})}} w_{u_{l+1}v_l}(t_l) \cdot a(t_{l+1}) \cdot x_{l+1}
$$
$$
\overset{(1)}{=} \sum_{a_{l+1} \in \mathcal{C}(a_l)} \omega_{a_l a_{l+1}} \cdot x_{a_{l+1}}
$$

with weights

$$
\omega_{a_l a_{l+1}} := w_{u_{l+1}v_l}(t_l) \cdot a(t_{l+1}). \tag{43}
$$

Consequently, $\hat{\mathcal{T}}(v, t)$ is a tree in terms of Theorem B.1.

*Step 2: Equation (19) holds.*

Again by (1) the definition of the interpolation inflow tree $\hat{\mathcal{T}}(v, t)$ (Definition B.2), (2) Theorem B.1 (which we can apply according to *Step 1*) and (3) Equation (43), we obtain

$$
\begin{aligned}
\hat{c}_v^{(k)}(t) &\overset{(1)}{=} \hat{c}_{v_0}^{(k-0)}(t_0) \overset{(1)}{=} x_{a_0} \\
&\overset{(2)}{=} \sum_{p=(a_0,...,a_{l_p})\in\mathcal{P}(a_0,A_{\text{leaf}})} \omega(a_0,...,a_{l_p}) \cdot x_{l_p} \\
&\overset{(1,3)}{=} \sum_{p=((v_0,t_0)=(v,t),(u_1,v_1,t_1)...,(u_{l_p},v_{l_p},t_{l_p}))\in\hat{\mathcal{P}}(v,t)} \left( \prod_{l=0}^{l_p-1} w_{u_{l+1}v_l}(t_l) \cdot a(t_{l+1}) \right) \cdot \hat{c}_{v_{l_p}}^{(k-l_p)}(t_{l_p}),
\end{aligned}
$$

which was to show. $\qquad\square$

### E.5 PROOF OF THEOREM 3.3

**Theorem** (Theorem 3.3). *There exists a $\kappa \in \mathbb{N}$, such that $\hat{c}_v^{(k)}(t)$ as defined in Equation (12) is equal to zero for all $k > \kappa$, $v \in V$ and $t \in T_{hi} \cup T_{pr}$.*

*Proof.* Let $v \in V = V_{\text{b}} \sqcup (V \setminus V_{\text{b}})$ and $t \in T_{\text{hi}} \sqcup T_{\text{pr}}$ hold.

*Case 1:* If $t \in T_{\text{hi}}$ or $t \in T_{\text{pr}}$ and $v \in V_{\text{b}}$ holds, by Equation (12), $\hat{c}_v^{(k)}(t)$ is equal to zero if $k > 0$ and equal to $c_v^{(k)}(t)$ if $k = 0$.

*Case 2.1:* If $t \in T_{\text{pr}}$, $v \in V \setminus V_{\text{b}}$ and $k \geq \max_{p\in\hat{\mathcal{P}}(v,t)} l_p$ holds, by Theorem B.3, we can express $\hat{c}_v^{(k)}(t)$ as functions of $\hat{c}_{v_{l_p}}^{(k-l_p)}(t_{l_p})$ for paths $p = ((v_0,t_0) = (v,t),(u_1,v_1,t_1)...,(u_{l_p},v_{l_p},t_{l_p})) \in \hat{\mathcal{P}}(v,t)$. By definition of $\hat{\mathcal{P}}(v,t)$, $t_{l_p} \in T_{\text{hi}}$ or $t_{l_p} \in T_{\text{pr}}$ and $v_{l_p} \in V_{\text{b}}$ holds and even more, $t_{l_p}$ is the first of the times $t_0 = t > t_1 > ... > t_{l_p}$ which satisfies this criterion (that is, $t_l \in T_{\text{pr}}$ and $v_l \in V \setminus V_{\text{b}}$ holds for all $l = 0,...,l_p - 1$). Therefore, by Equation (12), for each of these different lengths $l_p \leq k$, $\hat{c}_{v_{l_p}}^{(k-l_p)}(t_{l_p})$ is equal to zero if $k > l_p$ and equal to $c_{v_{l_p}}(t_{l_p})$ for $k = l_p$.

*Case 2.2:* If $t \in T_{\text{pr}}$, $v \in V \setminus V_{\text{b}}$ and $k < \max_{p\in\hat{\mathcal{P}}(v,t)} l_p$ holds, we can only express $\hat{c}_v^{(k)}(t)$ as functions of $\hat{c}_{v_{l_p}}^{(k-l_p)}(t_{l_p})$ for paths $p = ((v_0,t_0) = (v,t),(u_1,v_1,t_1)...,(u_{l_p},v_{l_p},t_{l_p})) \in \hat{\mathcal{P}}(v,t)$ which additionally satisfy $l_p \leq k$. For these paths, the same findings as in *Case 2.1* hold.

For all remaining paths which satisfy $l_p > k$, we can express $\hat{c}_v^{(k)}(t)$ as functions of $\hat{c}_{v_l}^{(k-l)}(t_l) = \hat{c}_{v_l}^{(0)}(t_l)$ for $l = k < l_p$. Since in this case, $t_l \in T_{\text{pr}}$ and $v_l \in V \setminus V_{\text{b}}$ holds, by Equation (12), $\hat{c}_{v_l}^{(k-l)}(t_l) = \hat{c}_{v_l}^{(0)}(t_l)$ is equal to zero.

*Case 2:* In summary, if $t \in T_{\text{pr}}$ and $v \in V \setminus V_{\text{b}}$ holds, we can express $\hat{c}_v^{(k)}(t)$ as functions of $\hat{c}_{v_{l_p}}^{(k-l_p)}(t_{l_p})$ which are only non-zero for $k \in \{l_p \mid p \in \hat{\mathcal{P}}(v,t)\}$.

Choosing $\kappa = \max_{v\in V} \max_{t\in T_{\text{hi}}\sqcup T_{\text{pr}}} \max_{p\in\hat{\mathcal{P}}(v,t)} l_p$, the claim follows. $\qquad\square$

### E.6 PROOF OF THEOREM 3.4 AND THEOREM B.8

**Theorem** (Theorem 3.4 and Theorem B.8). *Let $v \in V$ and $t \in T_{hi} \cup T_{pr}$ hold. Let $\mathcal{T}(v,t)$ and $\hat{\mathcal{T}}(v,t)$ be the inflow tree and the interpolation inflow tree of $v$ at time $t$ as defined in Definition B.2, respectively. Let Assumption B.5 and B.6 hold. Then the prediction error in-between $c_v(t)$ and $\hat{c}_v(t)$ is given by*

$$
|c_v(t) - \hat{c}_v(t)|
$$

$$
\leq \sum_{\tilde{p}\in\hat{\mathcal{P}}(v,t)} a(\hat{t}_1,...,\hat{t}_{l_p}) \cdot \left( \underbrace{|c_{v_{l_p}}(t_{l_p}) - c_{v_{l_p}}(\hat{t}_{l_p})|}_{\textit{Temporal interpolation error}} + |c_{v_{l_p}}(t_{l_p})| \cdot \underbrace{|w(t_0,...,t_{l_p-1}) - w(\hat{t}_0,...,\hat{t}_{l_p-1})|}_{\textit{Accumulated weighting error}} \right)
$$

*with joint paths $\tilde{p} = ((v_0, t_0) = (v, t), (u_1, v_1, t_1, \hat{t}_1), ..., (u_{l_p}, v_{l_p}, t_{l_p}, \hat{t}_{l_p})) \in \hat{\mathcal{P}}(v, t)$ according to Definition B.7,*

$$a(\hat{t}_1, ..., \hat{t}_{l_p}) := \prod_{l=0}^{l_p-1} a(\hat{t}_{l+1}) \quad and \quad w(\hat{t}_0, ..., \hat{t}_{l_p-1}) := \prod_{l=0}^{l_p-1} w_{u_{l+1}v_l}(\hat{t}_l)$$

*with weights $w_{u_{l+1}v_l}(t_l)$ and $a(t_{l+1})$ according to Equation (10) and (15), respectively. If the temporal interpolation error is zero, we obtain $|c_v(t) - \hat{c}_v(t)| = 0$.*

*Proof.* By Theorem B.3 and B.4,

$$c_v(t) = \sum_{p=((v_0,t_0)=(v,t),(u_1,v_1,t_1)...,(u_{l_p},v_{l_p},t_{l_p}))\in\mathcal{P}(v,t)} \left(\prod_{l=0}^{l_p-1} w_{u_{l+1}v_l}(t_l)\right) \cdot c_{v_{l_p}}(t_{l_p}) \text{ and}$$

$$\hat{c}_v^{(k)}(t) = \sum_{p=((v_0,t_0)=(v,t),(u_1,v_1,t_1)...,(u_{l_p},v_{l_p},t_{l_p}))\in\hat{\mathcal{P}}(v,t)} \left(\prod_{l=0}^{l_p-1} w_{u_{l+1}v_l}(t_l) \cdot a(t_{l+1})\right) \cdot \hat{c}_{v_{l_p}}^{(k-l_p)}(t_{l_p})$$

holds for all $k \geq \max_{p\in\hat{\mathcal{P}}(v,t)} l_p$. As already discussed in the proof of Theorem 3.3, for a $k < \max_{p\in\hat{\mathcal{P}}(v,t)} l_p$, paths $p \in \hat{\mathcal{P}}(v,t)$ with $l_p \leq k$ contribute with the same contribution $\hat{c}_{v_{l_p}}^{(k-l_p)}(t_{l_p}) = c_{v_{l_p}}(t_{l_p})$ to $\hat{c}_v^{(k)}(t)$ as they would for a larger $k$, while the ones with $l_p > k$ contribute with $\hat{c}_{v_l}^{(k-l)}(t_l) = 0$ (for $l = k < l_p$). More specifically, considering a flexible $k \in \mathbb{N}$, each path contributes only once to $\hat{c}_v^{(k)}(t)$, namely for $k = l_p$. Therefore, $\hat{c}_{v_{l_p}}^{(k-l_p)}(t_{l_p}) = \delta_{kl_p} \cdot c_{v_{l_p}}(t_{l_p})$ holds for all $k \geq l_p$ and also gives a meaning for cases $k < l_p$. Consequently, by (1) Equation (12) and (2) the findings above, we can express $\hat{c}_v(t)$ by

$$\hat{c}_v(t) \stackrel{(1)}{=} \sum_{k=0}^{\infty} \hat{c}_v^{(k)}(t)$$

$$\stackrel{(2)}{=} \sum_{k=0}^{\infty} \sum_{p=((v_0,t_0)=(v,t),(u_1,v_1,t_1)...,(u_{l_p},v_{l_p},t_{l_p}))\in\hat{\mathcal{P}}(v,t)} \left(\prod_{l=0}^{l_p-1} w_{u_{l+1}v_l}(t_l) \cdot a(t_{l+1})\right) \cdot \hat{c}_{v_{l_p}}^{(k-l_p)}(t_{l_p})$$

$$\stackrel{(2)}{=} \sum_{p=((v_0,t_0)=(v,t),(u_1,v_1,t_1)...,(u_{l_p},v_{l_p},t_{l_p}))\in\hat{\mathcal{P}}(v,t)} \left(\prod_{l=0}^{l_p-1} w_{u_{l+1}v_l}(t_l) \cdot a(t_{l+1})\right) \cdot c_{v_{l_p}}(t_{l_p}).$$

Now we want to use the shared paths $\tilde{p}$ from Equation (20) to express $c_v(t)$ and $\hat{c}_v(t)$ using the same summands. To achieve this, it is important to note that as an artifact of the interpolation, there are multiple paths $\hat{p} \in \hat{\mathcal{P}}(v, t)$ that are identifiable with a single path $p \in \mathcal{P}(v, t)$. However, due to the fact that certain interpolation weights $a(t_{l+1})$ add up to one, a single contribution $c_{v_{l_p}}(t_{l_p})$ of a path $p \in \mathcal{P}(v, t)$ to $c_v(t)$, weighted by the weight

$$\left(\prod_{l=0}^{l_p-1} w_{u_{l+1}v_l}(t_l)\right),$$

can be distributed to the multiple contributions $c_{v_{l_p}}(t_{l_p})$ of a path $p \in \mathcal{P}(v, t)$ to $c_v(t)$, weighted by the weights

$$\left(\prod_{l=0}^{l_p-1} w_{u_{l+1}v_l}(t_l) \cdot a(t_{l+1})\right)$$

associated with corresponding paths $\hat{p} \in \hat{\mathcal{P}}(v,t)$ that each are identifiable with the path $p \in \mathcal{P}(v,t)$. Using the shared paths $\tilde{p}$ notation from Equation (20), this allows to express both $c_v(t)$ and $\hat{c}_v(t)$ as

$$c_v(t) = \sum_{\tilde{p}=((v_0,t_0)=(v,t),(u_1,v_1,t_1,\hat{t}_1),\dots,(u_{l_p},v_{l_p},t_{l_p},\hat{t}_{l_p})) \in \hat{\mathcal{P}}(v,t)} \left( \prod_{l=0}^{l_p-1} w_{u_{l+1}v_l}(t_l) \cdot a(\hat{t}_{l+1}) \right) \cdot c_{v_{l_p}}(t_{l_p}) \text{ and}$$

$$\hat{c}_v^{(k)}(t) = \sum_{\tilde{p}=((v_0,t_0)=(v,t),(u_1,v_1,t_1,\hat{t}_1),\dots,(u_{l_p},v_{l_p},t_{l_p},\hat{t}_{l_p})) \in \hat{\mathcal{P}}(v,t)} \left( \prod_{l=0}^{l_p-1} w_{u_{l+1}v_l}(\hat{t}_l) \cdot a(\hat{t}_{l+1}) \right) \cdot c_{v_{l_p}}(\hat{t}_{l_p}),$$

making the sums perfectly comparable. Therefore, by (1) basic transformations and triangle-inequality and (2) the fact that $|w_1 c_1 - w_2 c_2| = |w_2 c_2 - w_2 c_1 + w_2 c_1 + w_1 c_1| \leq |c_1 - c_2| + |c_1||w_1 - w_2|$ holds for any $w_1, w_2 \in [0,1]$ and $c_1, c_2 \in \mathbb{R}$, we obtain

$$|c_v(t) - \hat{c}_v(t)|$$

$$\overset{(1)}{\leq} \sum_{\tilde{p} \in \hat{\mathcal{P}}(v,t)} \left( \prod_{l=0}^{l_p-1} a(\hat{t}_{l+1}) \right) \cdot \left| \left( \prod_{l=0}^{l_p-1} \underbrace{w_{u_{l+1}v_l}(t_l)}_{\in [0,1]} \right) \cdot c_{v_{l_p}}(t_{l_p}) - \left( \prod_{l=0}^{l_p-1} \underbrace{w_{u_{l+1}v_l}(\hat{t}_l)}_{\in [0,1]} \right) \cdot c_{v_{l_p}}(\hat{t}_{l_p}) \right|$$

$$\overset{(2)}{\leq} \sum_{\tilde{p} \in \hat{\mathcal{P}}(v,t)} \left( \prod_{l=0}^{l_p-1} a(\hat{t}_{l+1}) \right) \cdot \Bigg( \quad |c_{v_{l_p}}(t_{l_p}) - c_{v_{l_p}}(\hat{t}_{l_p})|$$

$$+ \quad |c_{v_{l_p}}(t_{l_p})| \cdot \left| \left( \prod_{l=0}^{l_p-1} w_{u_{l+1}v_l}(t_l) \right) - \left( \prod_{l=0}^{l_p-1} w_{u_{l+1}v_l}(\hat{t}_l) \right) \right| \Bigg).$$

If the temporal interpolation error is zero, that is, if $\phi = \varphi$ holds, all interpolation times $\hat{t}_l$ corresponding to left interpolation bound will be equal to $t_l$ hand have interpolation weights $a(\hat{t}_l) = 1$. Interpolation times $\hat{t}_l$ corresponding to right interpolation bounds have interpolation weights $a(\hat{t}_l) = 0$ (cf. Eq. (15). Therefore, in this case, the only non-zero summands in the error bound are the ones where $t_l = \hat{t}_l$ holds for all $l = 0, \dots, l_p$, in which case $|c_v(t) - \hat{c}_v(t)| = 0$ holds. This proves the claim.

$\square$

### E.7 PROOF OF LEMMA C.4

**Lemma** (Lemma C.4). *Let $v \in V$, $u \in \mathcal{N}(v)$, $t_0 \in \mathbb{R}$, $z_0 \in \mathbb{R}$ hold and let $T_{t_0,z_0}$ be defined as in Lemma 3.1.*

1. *If $z_0 \notin (0, l_{uv})$ holds, then $T_{t_0,z_0} = \emptyset$ holds.*

2. *If $z_0 \in (0, l_{uv})$ holds, then $T_{t_0,z_0} \neq \emptyset$ holds. Even more, $t_0 \in T_{t_0,z_0}$ holds.*

3. *If $z_0 \in (0, l_{uv})$ holds, there exist $t_l \in [-\infty, t_0)$ and $t_r \in (t_0, \infty]$ such that*

    (a) *if $t_l \neq -\infty$ holds, $z_0 + \int_{t_0}^{t_l} \nu_{uv}(s)\, ds \in \{0, l_{uv}\}$ holds,*

    (b) *if $t_r \neq +\infty$ holds, $z_0 + \int_{t_0}^{t_r} \nu_{uv}(s)\, ds \in \{0, l_{uv}\}$ holds and*

    (c) *$T_{t_0,z_0} = (t_l, t_l) = (t_0 - (t_0 - t_l), t_0 + (t_r - t_0))$ holds.*

    *Equivalently, there exist $\delta t_l, \delta t_r \in (0, \infty]$ such that $T_{t_0,z_0} = (t_0 - \delta t_l, t_0 + \delta t_l)$ holds.*

*Proof.* Similarly to Remark C.3, we define the function

$$z : \mathbb{R} \longrightarrow \mathbb{R}, \ t \longmapsto z(t) := z_0 + \int_{t_0}^{t} \nu_{uv}(s)\, ds.$$

By the fundamental theorem of calculus, the function $z$ is differentiable and therefore, continuous. Moreover, using the function $z$, we can re-write the set $T_{t_0, z_0}$ as

$$T_{t_0, z_0} = \left\{ t \in \mathbb{R} \mid z(t^{'}) \in (0, l_{uv}) \, \forall t^{'} \in \left[ \min\{t_0, t\}, \max\{t_0, t\} \right] \right\}.$$

*1.:* If $z_0 \notin (0, l_{uv})$ holds, let us assume that $T_{t_0, z_0} \neq \emptyset$ holds. By definition of $T_{t_0, z_0}$, there exists a $t \in \mathbb{R}$, such that $z(t^{'}) \in (0, l_{uv})$ holds for all $t^{'} \in [\min\{t_0, t\}, \max\{t_0, t\}]$. However, for $t^{'} = t_0 \in [\min\{t_0, t\}, \max\{t_0, t\}]$, we obtain $z(t^{'}) = z(t_0) = z_0 + \int_{t_0}^{t_0} \nu_{uv}(s) \, ds = z_0 \notin (0, l_{uv})$ – a contradiction.

*2.:* If $z_0 \in (0, l_{uv})$ holds, we obtain $z(t_0) = z_0 + \int_{t_0}^{t_0} \nu_{uv}(s) \, ds = z_0 \in (0, l_{uv})$ and therefore, trivially, $t_0 \in T_{t_0, z_0}$ and $T_{t_0, z_0} \neq \emptyset$ holds.

*3.:* We define the sets

$$A_l := \{ t^{'} \in (-\infty, t_0] \mid z(t^{'}) \notin (0, l_{uv}) \} \text{ and}$$

$$A_r := \{ t^{'} \in [t_0, +\infty) \mid z(t^{'}) \notin (0, l_{uv}) \}.$$

Based on these sets, we choose

$$t_l := \begin{cases} \max A_l & \text{if } A_l \neq \emptyset \\ -\infty & \text{if } A_l = \emptyset \end{cases} \quad \text{and} \quad t_r := \begin{cases} \min A_r & \text{if } A_r \neq \emptyset \\ +\infty & \text{if } A_r = \emptyset \end{cases}.$$

If $z_0 \in (0, l_{uv})$ holds, as seen above, $z(t_0) = z_0 \in (0, l_{uv})$ holds. Therefore, we observe that $t_0 \notin A_l$ and $t_0 \notin A_r$, or equivalently, $A_l \subset (-\infty, t_0)$ and $A_r \subset (t_0, +\infty)$ hold. Consequently, our choices of $t_l$ and $t_r$ indeed satisfy $t_l \in [-\infty, t_0)$ and $t_r \in (t_0, \infty]$ and even more, $t_0 \in (t_l, t_r)$ holds.

We now want to show that for these choices, indeed, *5(a) - 5(c)* hold.

*3(a), 3(b):* If $t_l \neq -\infty$ holds, $z_0 + \int_{t_0}^{t_l} \nu_{uv}(s) \, ds \in \{0, l_{uv}\}$ holds and if $t_r \neq +\infty$ holds, $z_0 + \int_{t_0}^{t_r} \nu_{uv}(s) \, ds \in \{0, l_{uv}\}$ holds.

The claims are equivalent to show that if $A_l \neq \emptyset$ holds, $z(t_l) \in \{0, l_{uv}\}$ holds and if $A_r \neq \emptyset$ holds, $z(t_r) \in \{0, l_{uv}\}$ holds.

If $A_l \neq \emptyset$ holds, let us assume that $z(t_l) \notin \{0, l_{uv}\}$ holds. Since $t_l = \max A_l \in A_l$, $z(t_l) > l_{uv}$ or $z(t_l) < 0$ must hold. Without loss of generality (W.l.o.g.), let us assume that $z(t_l) < 0$ holds. Since $z$ is continuous and $z(t_0) = z_0 \in (0, l_{uv})$, i.e., $0 < z(t_0)$ holds, according to the intermediate value theorem, there exists a $t^{'} \in (t_l, t_0)$ such that $z(t^{'}) = 0$ holds. Therefore, we can conclude that $\max A_l = t_l < t^{'} \in A_l$ holds – a contradiction to the choice of $t_l = \min A_l$.

If $A_r \neq \emptyset$ holds, let us assume $z(t_r) \notin \{0, l_{uv}\}$ holds. Analogously, analogous arguments lead to a contradiction.

*3(c), step 1: $T_{t_0, z_0} \subset (t_l, t_l)$ holds.*

Let $t \in T_{t_0, z_0} \subset \mathbb{R}$ and let us assume that $t \notin (t_l, t_r)$, or equivalently, $t \leq t_l < t_0$ or $t \geq t_r > t_0$ holds.
*Case 1:* If $t \leq t_0$ and $A_l \neq \emptyset$ holds, we observe that $t \leq t_l = \max A_l < t_0$ must hold. By definition of $T_{t_0, z_0}$, $z(t^{'}) \in (0, l_{uv})$ holds for all $t^{'} \in [\min\{t_0, t\}, \max\{t_0, t\}] = [t, t_0]$. However, by definition of $A_l$, for $t^{'} = t_l \in [t, t_0) \subset [t, t_0]$, $z(t^{'}) = z(t_l) \notin (0, l_{uv})$ holds – a contradiction to the first conclusion.
*Case 2:* If $t \leq t_0$ and $A_l = \emptyset$ holds, we observe that $t \leq t_l = -\infty$ must hold – a contradiction.
*Case 3:* If $t \geq t_0$ and $A_l \neq \emptyset$ holds, we observe that $t \geq t_r = \min A_r > t_0$ must hold. By definition of $T_{t_0, z_0}$, $z(t^{'}) \in (0, l_{uv})$ holds for all $t^{'} \in [\min\{t_0, t\}, \max\{t_0, t\}] = [t_0, t]$. However, by definition of $A_r$, for $t^{'} = t_r \in (t_0, t] \subset [t, t_0]$, $z(t^{'}) = z(t_r) \notin (0, l_{uv})$ holds – a contradiction to the first conclusion.
*Case 4:* If $t \geq t_0$ and $A_l = \emptyset$ holds, we observe that $t \geq t_r = +\infty$ must hold – a contradiction.

*5(c), step 2:* $(t_l, t_l) \subset T_{t_0,z_0}$ *holds.*

Let $t \in (t_1, t_1) \subset \mathbb{R}$ and let us assume that $t \notin T_{t_0,z_0}$ holds.

*Case 1:* If $t \leq t_0$ and $A_1 \neq \emptyset$ holds, by definition of $T_{t_0,z_0}$, there exists a $t' \in [\min\{t_0, t\}, \max\{t_0, t\}] = [t, t_0] \subset (t_1, t_0] = (\max A_1, t_0]$ such that $z(t') \notin (0, l_{uv})$ holds. Therefore, we can conclude that $\max A_1 = t_1 < t' \in A_1$ holds – a contradiction to the choice of $t_1 = \max A_1$.

*Case 2:* If $t \leq t_0$ and $A_1 \neq \emptyset$ holds, by definition of $T_{t_0,z_0}$, there exists a $t' \in [\min\{t_0, t\}, \max\{t_0, t\}] = [t, t_0] \subset (t_1, t_0] = (-\infty, t_0]$ such that $z(t') \notin (0, l_{uv})$ holds. Therefore, we can conclude that $t' \in A_1$ holds – a contradiction to $A_1 \neq \emptyset$.

*Case 3:* If $t \geq t_0$ and $A_r \neq \emptyset$ holds, by definition of $T_{t_0,z_0}$, there exists a $t' \in [\min\{t_0, t\}, \max\{t_0, t\}] = [t_0, t] \subset [t_0, t_r) = [t_0, \min A_r)$ such that $z(t') \notin (0, l_{uv})$ holds. Therefore, we can conclude that $\min A_r = t_r > t' \in A_r$ holds – a contradiction to the choice of $t_r = \min A_1$.

*Case 4:* If $t \geq t_0$ and $A_r \neq \emptyset$ holds, by definition of $T_{t_0,z_0}$, there exists a $t' \in [\min\{t_0, t\}, \max\{t_0, t\}] = [t_0, t] \subset [t_0, t_r) = [t_0, +\infty)$ such that $z(t') \notin (0, l_{uv})$ holds. Therefore, we can conclude that $t' \in A_r$ holds – a contradiction to $A_r \neq \emptyset$. □

## E.8 Proof of Theorem C.5

**Theorem** (Theorem C.5). *Let $v \in V$ and $u \in \mathcal{N}(v)$ hold. If the function $c_{uv}$ obeys Equation (21), for each $t_0 \in \mathbb{R}$ and $z_0 \in (0, l_{uv})$,*

$$c_{uv}\left(t, z_0 + \int_{t_0}^{t} \nu_{uv}(s)\, ds\right) = c_{uv}\left(t, z_0 + \overline{\nu}_{uv}(t_0, t)\, (t - t_0)\right) = c_{uv}(t_0, z_0)$$

*holds for all $t \in T_{t_0,z_0}$ with $T_{t_0,z_0}$ as defined in Lemma 3.1 and for the mean velocity*

$$\overline{\nu}_{uv}(t_0, t) := \begin{cases} \dashint_{t_0}^{t} \nu_{uv}(s)\, ds & \text{if } t \neq t_0 \\ 0 & \text{if } t = t_0 \end{cases} = \begin{cases} \frac{1}{t - t_0} \int_{t_0}^{t} \nu_{uv}(s)\, ds & \text{if } t \neq t_0 \\ 0 & \text{if } t = t_0 \end{cases}.$$

*Proof.* For each $t_0 \in \mathbb{R}$ and $z_0 \in (0, l_{uv}) \subset \mathbb{R}$, by Lemma 3.1, $\gamma_{t_0,z_0}(t) = c_{uv}(t, z_0 + \int_{t_0}^{t} \nu_{uv}(s)\, ds) = c$ holds for all $t \in T_{t_0,z_0}$ and a constant $c \in \mathbb{R}_{\geq 0}$. Thus, it suffices to show that $c = c_{uv}(t_0, z_0)$ holds.

Let us assume that $c \neq c_{uv}(t_0, z_0)$ holds. Since $z_0 \in (0, l_{uv})$, by Lemma C.4, $t_0 \in T_{t_0,z_0}$ holds. However, for $t = t_0 \in T_{t_0,z_0}$, we obtain $c = \gamma_{t_0,z_0}(t) = c_{uv}(t, z_0 + \int_{t_0}^{t_0} \nu_{uv}(s)\, ds) = c_{uv}(t_0, z_0)$ – a contradiction. Therefore, $c_{uv}(t, z_0 + \int_{t_0}^{t} \nu_{uv}(s)\, ds) = c_{uv}(t_0, z_0)$ holds. Finally, by definition of $\overline{\nu}_{uv}(t_0, t)$, for $t \neq t_0$, we obtain

$$\overline{\nu}_{uv}(t_0, t) \cdot (t - t_0) = \frac{t - t_0}{t - t_0} \int_{t_0}^{t} \nu_{uv}(s)\, ds = \int_{t_0}^{t} \nu_{uv}(s)\, ds,$$

allowing to conclude that $c_{uv}(t, z_0 + \int_{t_0}^{t} \nu_{uv}(s)\, ds) = c_{uv}(t, z_0 + \overline{\nu}_{uv}(t_0, t)\, (t - t_0))$ holds. □

## E.9 Proof of Lemma C.7

**Lemma** (Lemma C.7). *Let $v \in V$, $u \in \mathcal{N}(v)$ and $t \in \mathbb{R}$ hold. If there exists a pass-through time $\delta t_{uv} \in \mathbb{R}$ w.r.t. $e_{uv} \in E$ and $t \in \mathbb{R}$, the following properties hold:*

1. *$\delta t_{uv} \in \mathbb{R}_{\neq 0}$ and thus, $t - \delta t_{uv} \neq t$ holds.*

2. *There exists no other pass-through time of the same sign (i.e., there exists either exactly one or no positive pass-through time, and there exists exactly one or no negative pass-through time).*

3. *The mean velocity*

$$\overline{\nu}_{uv}(t) := \overline{\nu}_{uv}(t - \delta t_{uv}, t) = \fint_{t-\delta t_{uv}}^{t} \nu_{uv}(s) \, ds = \frac{1}{\delta t_{uv}} \int_{t-\delta t_{uv}}^{t} \nu_{uv}(s) \, ds$$

*during the time interval* $[\min\{t - \delta t_{uv}, t\}, \max\{t - \delta t_{uv}, t\}]$ *as defined in Theorem C.5 is equal to* $\overline{\nu}_{uv}(t) = \frac{l_{uv}}{\delta t_{uv}}$ *and thus satisfies*

$$\overline{\nu}_{uv}(t) > 0 \iff \delta t_{uv} > 0 \text{ and}$$
$$\overline{\nu}_{uv}(t) < 0 \iff \delta t_{uv} < 0.$$

4. *The flows* $q_{uv}(t - \delta t_{uv})$ *and* $q_{uv}(t)$ *linked to the velocity* $\nu_{uv}(\cdot) = \frac{q_{vu}(\cdot)}{\alpha_{vu}}$ *satisfy*

$$\begin{cases} q_{uv}(t - \delta t_{uv}), q_{uv}(t) \geq 0 & \text{if } \delta t_{uv} > 0 \\ q_{uv}(t - \delta t_{uv}), q_{uv}(t) \leq 0 & \text{if } \delta t_{uv} < 0 \end{cases}.$$

*Proof. 1.:* Let us assume that $\delta t_{uv} = 0$ holds. Consequently, (1) by basic results from analysis, (2) condition (23) and (3) definition of the pipe length $l_{uv} > 0$,

$$0 \overset{(1)}{=} \int_{t-0}^{t} \nu_{uv}(s) \, ds \overset{(1)}{=} \int_{t-\delta t_{uv}}^{t} \nu_{uv}(s) \, ds \overset{(2)}{=} l_{uv} \overset{(3)}{>} 0$$

holds – a contradiction.

*2.:* Let us assume that there exist two different pass-through times $\delta t_{1uv}, \delta t_{2uv} \in \mathbb{R}_{\neq 0}$ with the same sign. If $\delta t_{1uv}, \delta t_{2uv} > 0$ holds, without loss of generality (w.l.o.g.), let $0 < \delta t_{1uv} < \delta t_{2uv}$ hold. Consequently, for $t' = t - \delta t_{1uv} \in (t - \delta t_{2uv}, t) = (\min\{t - \delta t_{2uv}, t\}, \max\{t - \delta t_{2uv}, t\})$, by (1) by basic results from analysis and (2) condition (23), we obtain

$$\int_{t-\delta t_{2uv}}^{t'} \nu_{uv}(s) \, ds \overset{(1)}{=} \int_{t-\delta t_{2uv}}^{t} \nu_{uv}(s) \, ds - \int_{t-\delta t_{1uv}}^{t} \nu_{uv}(s) \, ds \overset{(2)}{=} l_{uv} - l_{uv} \overset{(1)}{=} 0$$

– a contradiction to condition (24).

If $\delta t_{1uv}, \delta t_{2uv} < 0$ holds, w.l.o.g., let $\delta t_{2uv} < \delta t_{1uv} < 0$ hold. Consequently, for $t' = t - \delta t_{1uv} \in (t, t - \delta t_{2uv}) = (\min\{t - \delta t_{2uv}, t\}, \max\{t - \delta t_{2uv}, t\})$, we can use the same calculation to again obtain a contradiction.

*3.:* By (1) definition of the mean velocity $\overline{\nu}_{uv}(t - \delta t_{uv}, t)$ as defined in Theorem C.5 and (2) condition (23), we indeed obtain

$$\overline{\nu}_{uv}(t) := \overline{\nu}_{uv}(t - \delta t_{uv}, t) \overset{(1)}{=} \fint_{t-\delta t_{uv}}^{t} \nu_{uv}(s) \, ds \overset{(1)}{=} \frac{1}{\delta t_{uv}} \int_{t-\delta t_{uv}}^{t} \nu_{uv}(s) \, ds \overset{(2)}{=} \frac{l_{uv}}{\delta t_{uv}}.$$

The second claim then follows by the fact that by definition of the pipe length, $l_{uv} > 0$ holds.

*4.:* If $\delta t_{uv} > 0$ holds, let us assume that $q_{uv}(t - \delta t_{uv}) < 0$, and consequently, $\nu_{uv}(t - \delta t_{uv}) < 0$ holds. Since $\nu_{uv} \in C^0(\mathbb{R}, \mathbb{R})$ is continuous, there exists a $t' \in (t - \delta t_{uv}, t) = (\min\{t - \delta t_{uv}, t\}, \max\{t - \delta t_{uv}, t\})$ such that $\nu_{uv}(s) < 0$ holds for all $s \in (t - \delta t_{uv}, t')$. Consequently, by (1) basic results from analysis, we obtain

$$\int_{t-\delta t_{uv}}^{t'} \underbrace{\nu_{uv}(s)}_{<0} \, ds \overset{(1)}{<} 0$$

– a contradiction to condition (24).

Similarly, if $\delta t_{uv} > 0$ holds, let us assume that $q_{uv}(t) < 0$, and consequently, $\nu_{vu}(t) < 0$ holds. Since $\nu_{vu} \in C^0(\mathbb{R}, \mathbb{R})$ is continuous, there exists a $t' \in (t - \delta t_{uv}, t) = (\min\{t - \delta t_{uv}, t\}, \max\{t -$

$\delta t_{uv}, t\})$ such that $\nu_{uv}(s) < 0$ holds for all $s \in (t^{'}, t)$. Consequently, by (1) basic results from analysis and (2) condition (23), we obtain

$$\int_{t-\delta t_{uv}}^{t^{'}} \nu_{uv}(s)\, ds \overset{(1)}{=} \underbrace{\int_{t-\delta t_{uv}}^{t} \nu_{uv}(s)\, ds}_{\overset{(2)}{=} l_{uv}} - \underbrace{\underbrace{\int_{t^{'}}^{t} \underbrace{\nu_{uv}(s)}_{<0}\, ds}_{\overset{(1)}{<0}}}_{>0} > l_{uv}$$

– again a contradiction to condition (24).

In contrast, if $\delta t_{uv} < 0$ holds, let us assume that $q_{uv}(t - \delta t_{uv}) > 0$, and consequently, $\nu_{uv}(t - \delta t_{uv}) > 0$ holds. Since $\nu_{uv} \in C^0(\mathbb{R}, \mathbb{R})$ is continuous, there exists a $t^{'} \in (t, t - \delta t_{uv}) = (\min\{t - \delta t_{uv}, t\}, \max\{t - \delta t_{uv}, t\})$ such that $\nu_{uv}(s) > 0$ holds for all $s \in (t^{'}, t - \delta t_{uv})$. Consequently, by (1) basic results from analysis, we obtain

$$\int_{t-\delta t_{uv}}^{t^{'}} \nu_{uv}(s)\, ds \overset{(1)}{=} - \underbrace{\int_{t^{'}}^{t-\delta t_{uv}} \underbrace{\nu_{uv}(s)}_{>0}\, ds}_{\overset{(1)}{>0}} < 0$$

– again a contradiction to condition (24).

Similarly, if $\delta t_{uv} < 0$ holds, let us assume that $q_{uv}(t) > 0$, and consequently, $\nu_{uv}(t) > 0$ holds. Since $\nu_{uv} \in C^0(\mathbb{R}, \mathbb{R})$ is continuous, there exists a $t^{'} \in (t, t - \delta t_{uv}) = (\min\{t - \delta t_{uv}, t\}, \max\{t - \delta t_{uv}, t\})$ such that $\nu_{uv}(s) > 0$ holds for all $s \in (t, t^{'})$. Consequently, by (1) basic results from analysis and (2) condition (23), we obtain

$$\int_{t-\delta t_{uv}}^{t^{'}} \nu_{uv}(s)\, ds \overset{(1)}{=} \int_{t-\delta t_{uv}}^{t} \nu_{uv}(s)\, ds - \int_{t^{'}}^{t} \nu_{uv}(s)\, ds$$

$$\overset{(1)}{=} \underbrace{\int_{t-\delta t_{uv}}^{t} \nu_{uv}(s)\, ds}_{\overset{(2)}{=} l_{uv}} + \underbrace{\underbrace{\int_{t}^{t^{'}} \underbrace{\nu_{uv}(s)}_{>0}\, ds}_{\overset{(1)}{>0}}}_{>0} > l_{uv}$$

– again a contradiction to condition (24).

*Remark* E.1. Note that if one of the flow rates discussed above are zero, that they can only be zero in this point of time, but not on a whole sub-interval of the time interval $[\min\{t - \delta t_{uv}, t\}, \max\{t - \delta t_{uv}, t\}]$.

More precisely, if $\delta t_{uv} > 0$ holds, let us assume that there exists an $\epsilon > 0$ such that for all $s \in [t - \delta t_{uv}, t - \delta t_{uv} + \epsilon), q_{uv}(s) = 0$, and consequently, $\nu_{uv}(s) = 0$, holds. Thus, for $t^{'} = \min\{t - \delta t_{uv} + \epsilon, t\} \in (t - \delta t_{uv}, t) = (\min\{t - \delta t_{uv}, t\}, \max\{t - \delta t_{uv}, t\})$, by (1) basic results from analysis, we obtain

$$\int_{t-\delta t_{uv}}^{t^{'}} \underbrace{\nu_{uv}(s)}_{=0}\, ds \overset{(1)}{=} 0$$

– a contradiction to condition (24). In combination with the findings above, therefore, $q_{uv}(t - \delta t_{uv}) \geq 0$ needs to hold, and if the (unlikely) case $q_{uv}(t - \delta t_{uv}) = 0$ indeed applies, we observe that for all $\epsilon > 0$ there exists an $s \in [t - \delta t_{uv}, t - \delta t_{uv} + \epsilon)$ such that $q_{uv}(s) \neq 0$, and consequently, $\nu_{uv}(s) \neq 0$, holds. Along the same lines, one can also show that indeed, $q_{uv}(s) > 0$, and consequently, $\nu_{uv}(s) > 0$, holds (otherwise, we would again obtain a contradiction to condition (28)).

The other cases are analogous adaptations of this discussion and the findings above.

□

### E.10 PROOF OF LEMMA C.8

**Lemma** (Lemma C.8). *Let $v \in V$, $u \in \mathcal{N}(v)$ and $t \in \mathbb{R}$ hold. If there exists a self-loop time $\delta t_{uv} \in \mathbb{R}_{\neq 0}$ w.r.t. $e_{uv} \in E$ and $t \in \mathbb{R}$, the following properties hold:*

1. *There exists no other self-loop time of the same sign (i.e., there exists either exactly one or no positive self-loop time, and there exists exactly one or no negative self-loop time).*

2. *The mean velocity*

$$\overline{\nu}_{uv}(t) := \overline{\nu}_{uv}(t - \delta t_{uv}, t) = \fint_{t-\delta t_{uv}}^{t} \nu_{uv}(s)\, ds = \frac{1}{\delta t_{uv}} \int_{t-\delta t_{uv}}^{t} \nu_{uv}(s)\, ds$$

   *during the time interval $[\min\{t - \delta t_{uv}, t\}, \max\{t - \delta t_{uv}, t\}]$ as defined in Theorem C.5 satisfies $\overline{\nu}_{uv}(t) = 0$.*

3. *The flows $q_{uv}(t - \delta t_{uv})$ and $q_{uv}(t)$ linked to the velocity $\nu_{uv}(\cdot) = \frac{q_{vu(\cdot)}}{\alpha_{vu}}$ satisfy*

$$\begin{cases} q_{uv}(t - \delta t_{uv}) \geq 0,\ q_{uv}(t) \leq 0 & \text{if } \delta t_{uv} > 0 \\ q_{uv}(t - \delta t_{uv}) \leq 0,\ q_{uv}(t) \geq 0 & \text{if } \delta t_{uv} < 0 \end{cases}.$$

The proof of Lemma C.8 is very similar to the proof of Lemma C.7 and only requires a few changes. The interested reader can compare both proofs and investigate why the changes are required, and how they induce the different results in Lemma C.8.3 as compared to Lemma C.7.4.

*Proof. 1.:* Let us assume that there exist two different self-loop times $\delta t_{1uv}, \delta t_{2uv} \in \mathbb{R}_{\neq 0}$ with the same sign. If $\delta t_{1uv}, \delta t_{2uv} > 0$ holds, w.l.o.g., let $0 < \delta t_{1uv} < \delta t_{2uv}$ hold. Consequently, for $t' = t - \delta t_{1uv} \in (t - \delta t_{2uv}, t) = (\min\{t - \delta t_{2uv}, t\}, \max\{t - \delta t_{2uv}, t\})$, by (1) by basic results from analysis and (2) condition (27), we obtain

$$\int_{t-\delta t_{2uv}}^{t'} \nu_{uv}(s)\, ds \overset{(1)}{=} \int_{t-\delta t_{2uv}}^{t} \nu_{uv}(s)\, ds - \int_{t-\delta t_{1uv}}^{t} \nu_{uv}(s)\, ds \overset{(2)}{=} 0 - 0 \overset{(1)}{=} 0$$

– a contradiction to condition (24).

If $\delta t_{1uv}, \delta t_{2uv} < 0$ holds, w.l.o.g., let $\delta t_{2uv} < \delta t_{1uv} < 0$ hold. Consequently, for $t' = t - \delta t_{1uv} \in (t, t - \delta t_{2uv}) = (\min\{t - \delta t_{2uv}, t\}, \max\{t - \delta t_{2uv}, t\})$, we can use the same calculation to again obtain a contradiction.

*2.:* By (1) definition of the mean velocity $\overline{\nu}_{uv}(t - \delta t_{uv}, t)$ as defined in Theorem C.5 and (2) condition (27), we indeed obtain

$$\overline{\nu}_{uv}(t) := \overline{\nu}_{uv}(t - \delta t_{uv}, t) \overset{(1)}{=} \fint_{t-\delta t_{uv}}^{t} \nu_{uv}(s)\, ds \overset{(1)}{=} \frac{1}{\delta t_{uv}} \int_{t-\delta t_{uv}}^{t} \nu_{uv}(s)\, ds \overset{(2)}{=} 0.$$

*3.:* If $\delta t_{uv} > 0$ holds, let us assume that $q_{uv}(t - \delta t_{uv}) < 0$, and consequently, $\nu_{uv}(t - \delta t_{uv}) < 0$ holds. Since $\nu_{uv} \in C^0(\mathbb{R}, \mathbb{R})$ is continuous, there exists a $t' \in (t - \delta t_{uv}, t) = (\min\{t - \delta t_{uv}, t\}, \max\{t - \delta t_{uv}, t\})$ such that $\nu_{uv}(s) < 0$ holds for all $s \in (t - \delta t_{uv}, t')$. Consequently, by (1) basic results from analysis, we obtain

$$\int_{t-\delta t_{uv}}^{t'} \underbrace{\nu_{uv}(s)}_{<0}\, ds \overset{(1)}{<} 0$$

– a contradiction to condition (28).

Similarly, if $\delta t_{uv} > 0$ holds, let us assume that $q_{uv}(t) > 0$, and consequently, $\nu_{uv}(t) > 0$ holds. Since $\nu_{uv} \in C^0(\mathbb{R}, \mathbb{R})$ is continuous, there exists a $t^{'} \in (t - \delta t_{uv}, t) = (\min\{t - \delta t_{uv}, t\}, \max\{t - \delta t_{uv}, t\})$ such that $\nu_{uv}(s) > 0$ holds for all $s \in (t^{'}, t)$. Consequently, by (1) basic results from analysis and (2) condition (27), we obtain

$$\int_{t - \delta t_{uv}}^{t^{'}} \nu_{uv}(s) \, ds \overset{(1)}{=} \underbrace{\int_{t - \delta t_{uv}}^{t} \nu_{uv}(s) \, ds}_{\overset{(2)}{=} 0} - \underbrace{\int_{t^{'}}^{t} \underbrace{\nu_{uv}(s)}_{\overset{}{>0}} \, ds}_{\overset{(1)}{>0}} < 0$$

$$\underbrace{\phantom{xxxxxxxxxxxxxxxxxxxxxxxxxxxxxxxxxxxxxxxxxxxxxxxxxxx}}_{<0}$$

– again a contradiction to condition (28).

In contrast, if $\delta t_{uv} < 0$ holds, let us assume that $q_{uv}(t - \delta t_{uv}) > 0$, and consequently, $\nu_{uv}(t - \delta t_{uv}) > 0$ holds. Since $\nu_{uv} \in C^0(\mathbb{R}, \mathbb{R})$ is continuous, there exists a $t^{'} \in (t, t - \delta t_{uv}) = (\min\{t - \delta t_{uv}, t\}, \max\{t - \delta t_{uv}, t\})$ such that $\nu_{uv}(s) > 0$ holds for all $s \in (t^{'}, t - \delta t_{uv})$. Consequently, by (1) basic results from analysis, we obtain

$$\int_{t - \delta t_{uv}}^{t^{'}} \nu_{uv}(s) \, ds \overset{(1)}{=} - \underbrace{\int_{t^{'}}^{t - \delta t_{uv}} \underbrace{\nu_{uv}(s)}_{>0} \, ds}_{\overset{(1)}{>0}} < 0$$

– again a contradiction to condition (28).

Similarly, if $\delta t_{uv} < 0$ holds, let us assume that $q_{uv}(t) < 0$, and consequently, $\nu_{uv}(t) < 0$ holds. Since $\nu_{uv} \in C^0(\mathbb{R}, \mathbb{R})$ is continuous, there exists a $t^{'} \in (t, t - \delta t_{uv}) = (\min\{t - \delta t_{uv}, t\}, \max\{t - \delta t_{uv}, t\})$ such that $\nu_{uv}(s) < 0$ holds for all $s \in (t, t^{'})$. Consequently, by (1) basic results from analysis and (2) condition (27), we obtain

$$\int_{t - \delta t_{uv}}^{t^{'}} \nu_{uv}(s) \, ds \overset{(1)}{=} \int_{t - \delta t_{uv}}^{t} \nu_{uv}(s) \, ds - \int_{t^{'}}^{t} \nu_{uv}(s) \, ds$$

$$\overset{(1)}{=} \underbrace{\int_{t - \delta t_{uv}}^{t} \nu_{uv}(s) \, ds}_{\overset{(2)}{=} 0} + \underbrace{\int_{t}^{t^{'}} \underbrace{\nu_{uv}(s)}_{<0} \, ds}_{\overset{(1)}{<0}} < 0$$

$$\underbrace{\phantom{xxxxxxxxxxxxxxxxxxxxxxxxxxxxxxxxxxxxxxxxxxxxxxxxxxx}}_{<0}$$

– again a contradiction to condition (28).

*Remark* E.2. Note that if one of the flow rates discussed above are zero, that they can only be zero in this point of time, but not on a whole sub-interval of the time interval $[\min\{t - \delta t_{uv}, t\}, \max\{t - \delta t_{uv}, t\}]$.

More precisely, if $\delta t_{uv} > 0$ holds, let us assume that there exists an $\epsilon > 0$ such that for all $s \in [t - \delta t_{uv}, t - \delta t_{uv} + \epsilon), q_{uv}(s) = 0$, and consequently, $\nu_{uv}(s) = 0$, holds. Thus, for $t^{'} = \min\{t - \delta t_{uv} + \epsilon, t\} \in (t - \delta t_{uv}, t) = (\min\{t - \delta t_{uv}, t\}, \max\{t - \delta t_{uv}, t\})$, by (1) basic results from analysis, we obtain

$$\int_{t - \delta t_{uv}}^{t^{'}} \underbrace{\nu_{uv}(s)}_{=0} \, ds \overset{(1)}{=} 0$$

– a contradiction to condition (28). In combination with the findings above, therefore, $q_{uv}(t - \delta t_{uv}) \geq 0$ needs to hold, and if the (unlikely) case $q_{uv}(t - \delta t_{uv}) = 0$ indeed applies, we observe

that for all $\epsilon > 0$ there exists an $s \in [t - \delta t_{uv}, t - \delta t_{uv} + \epsilon)$ such that $q_{uv}(s) \neq 0$, and consequently, $\nu_{uv}(s) \neq 0$, holds. Along the same lines, one can also show that indeed, $q_{uv}(s) > 0$, and consequently, $\nu_{uv}(s) > 0$, holds (otherwise, we would again obtain a contradiction to condition (28)).

The other cases are analogous adaptations of this discussion and the findings above.

$\square$

### E.11 PROOF OF LEMMA C.9

**Lemma** (Lemma C.9). *Let $v \in V$, $u \in \mathcal{N}(v)$ and $t \in \mathbb{R}$ hold. $\delta t_{uv} \in \mathbb{R}$ is an inverse pass-through time w.r.t. $e_{uv} \in E$ and $t \in \mathbb{R}$ iff $\delta t_{vu} := \delta t_{uv} \in \mathbb{R}$ is a pass-through time w.r.t. $e_{vu} \in E$ and $t \in \mathbb{R}$*

*Proof.* "$\Rightarrow$" If $\delta t_{uv} \in \mathbb{R}$ is an inverse pass-through time w.r.t. $e_{uv} \in E$ and $t \in \mathbb{R}$, by (1) the fact that the geometric pipe features are symmetric in the sense that $l_{vu} = l_{uv}$ and $\alpha_{vu} = \alpha_{uv}$ holds for all $v \in V$ and $u \in \mathcal{N}(v)$, (2) definition of the flow velocity $\nu_{vu}(\cdot) = \frac{q_{vu}(\cdot)}{\alpha_{vu}}$, (3) the conservation of flows (cf. Eq. (4)) and (4) condition (25), we obtain

$$\int_{t-\delta t_{uv}}^{t} \nu_{vu}(s) \, ds \overset{(1,2,3)}{=} -\int_{t-\delta t_{uv}}^{t} \nu_{uv}(s) \, ds \overset{(4)}{=} -(-l_{uv}) \overset{(1)}{=} l_{vu},$$

that is, condition (23) is satisfied for $\delta t_{vu} := \delta t_{uv}$ and $\nu_{vu}$.

Similarly, by additionally using (5) condition (26), we obtain

$$\int_{t-\delta t_{uv}}^{t'} \nu_{vu}(s) \, ds \overset{(1,2,3)}{=} -\int_{t-\delta t_{uv}}^{t} \nu_{uv}(s) \, ds \overset{(5)}{\in} (0, l_{uv}) \overset{(1)}{=} (0, l_{vu}),$$

that is, condition (24) is satisfied for $\delta t_{vu} := \delta t_{uv}$ and $\nu_{vu}$. Therefore, $\delta t_{vu} := \delta t_{uv} \in \mathbb{R}$ is a pass-through time w.r.t. $e_{vu} \in E$ and $t \in \mathbb{R}$.

"$\Leftarrow$" If $\delta t_{vu} := \delta t_{uv} \in \mathbb{R}$ is a pass-through time w.r.t. $e_{vu} \in E$ and $t \in \mathbb{R}$, by (1) the fact that the geometric pipe features are symmetric in the sense that $l_{vu} = l_{uv}$ and $\alpha_{vu} = \alpha_{uv}$ holds for all $v \in V$ and $u \in \mathcal{N}(v)$, (2) definition of the flow velocity $\nu_{uv}(\cdot) = \frac{q_{uv}(\cdot)}{\alpha_{uv}}$, (3) the conservation of flows (cf. Eq. (4)) and (4) condition (23), we obtain

$$\int_{t-\delta t_{uv}}^{t} \nu_{uv}(s) \, ds \overset{(1,2,3)}{=} -\int_{t-\delta t_{uv}}^{t} \nu_{vu}(s) \, ds \overset{(4)}{=} -(l_{vu}) \overset{(1)}{=} -l_{uv},$$

that is, condition (25) is satisfied for $\delta t_{uv} := \delta t_{vu}$ and $\nu_{uv}$.

Similarly, by additionally using (5) condition (24), we obtain

$$\int_{t-\delta t_{uv}}^{t'} \nu_{uv}(s) \, ds \overset{(1,2,3)}{=} -\int_{t-\delta t_{uv}}^{t} \nu_{vu}(s) \, ds \overset{(5)}{\in} (-l_{vu}, 0) \overset{(1)}{=} (-l_{uv}, 0),$$

that is, condition (26) is satisfied for $\delta t_{uv} := \delta t_{vu}$ and $\nu_{uv}$. Therefore, $\delta t_{uv} := \delta t_{vu} \in \mathbb{R}$ is an inverse pass-through time w.r.t. $e_{uv} \in E$ and $t \in \mathbb{R}$. $\square$

### E.12 PROOF OF LEMMA C.10

**Lemma** (Lemma C.10). *Let $v \in V$, $u \in \mathcal{N}(v)$ and $t \in \mathbb{R}$ hold. $\delta t_{uv} \in \mathbb{R}$ is an inverse self-loop time w.r.t. $e_{uv} \in E$ and $t \in \mathbb{R}$ iff $\delta t_{vu} := \delta t_{uv} \in \mathbb{R}$ is a self-loop time w.r.t. $e_{vu} \in E$ and $t \in \mathbb{R}$*

The proof of Lemma C.10 is very similar to the proof of Lemma C.9 and only requires a few changes. The interested reader can compare both proofs and investigate why the changes are required, and how they induce the different results in Lemma C.10 as compared to Lemma C.10.

*Proof.* "⇒" If $\delta t_{uv} \in \mathbb{R}$ is an inverse self-loop time w.r.t. $e_{uv} \in E$ and $t \in \mathbb{R}$, by (1) the fact that the geometric pipe features are symmetric in the sense that $l_{vu} = l_{uv}$ and $\alpha_{vu} = \alpha_{uv}$ holds for all $v \in V$ and $u \in \mathcal{N}(v)$, (2) definition of the flow velocity $\nu_{vu}(\cdot) = \frac{q_{vu(\cdot)}}{\alpha_{vu}}$, (3) the conservation of flows (cf. Eq. (4)) and (4) condition (29), we obtain

$$\int_{t-\delta t_{uv}}^{t} \nu_{vu}(s)\, ds \overset{(1,2,3)}{=} -\int_{t-\delta t_{uv}}^{t} \nu_{uv}(s)\, ds \overset{(4)}{=} -0 = 0,$$

that is, condition (27) is satisfied for $\delta t_{vu} := \delta t_{uv}$ and $\nu_{vu}$.

Similarly, by additionally using (5) condition (30), we obtain

$$\int_{t-\delta t_{uv}}^{t'} \nu_{vu}(s)\, ds \overset{(1,2,3)}{=} -\int_{t-\delta t_{uv}}^{t} \nu_{uv}(s)\, ds \overset{(5)}{\in} (0, l_{uv}) \overset{(1)}{=} (0, l_{vu}),$$

that is, condition (28) is satisfied for $\delta t_{vu} := \delta t_{uv}$ and $\nu_{vu}$. Therefore, $\delta t_{vu} := \delta t_{uv} \in \mathbb{R}$ is a self-loop time w.r.t. $e_{vu} \in E$ and $t \in \mathbb{R}$.

"⇐" If $\delta t_{vu} := \delta t_{uv} \in \mathbb{R}$ is a self-loop time w.r.t. $e_{vu} \in E$ and $t \in \mathbb{R}$, by (1) the fact that the geometric pipe features are symmetric in the sense that $l_{vu} = l_{uv}$ and $\alpha_{vu} = \alpha_{uv}$ holds for all $v \in V$ and $u \in \mathcal{N}(v)$, (2) definition of the flow velocity $\nu_{uv}(\cdot) = \frac{q_{uv(\cdot)}}{\alpha_{uv}}$, (3) the conservation of flows (cf. Eq. (4)) and (4) condition (27), we obtain

$$\int_{t-\delta t_{uv}}^{t} \nu_{uv}(s)\, ds \overset{(1,2,3)}{=} -\int_{t-\delta t_{uv}}^{t} \nu_{vu}(s)\, ds \overset{(4)}{=} -(0) = 0,$$

that is, condition (29) is satisfied for $\delta t_{uv} := \delta t_{vu}$ and $\nu_{uv}$.

Similarly, by additionally using (5) condition (28), we obtain

$$\int_{t-\delta t_{uv}}^{t'} \nu_{uv}(s)\, ds \overset{(1,2,3)}{=} -\int_{t-\delta t_{uv}}^{t} \nu_{vu}(s)\, ds \overset{(5)}{\in} (-l_{vu}, 0) \overset{(1)}{=} (-l_{uv}, 0),$$

that is, condition (30) is satisfied for $\delta t_{uv} := \delta t_{vu}$ and $\nu_{uv}$. Therefore, $\delta t_{uv} := \delta t_{vu} \in \mathbb{R}$ is an inverse self-loop time w.r.t. $e_{uv} \in E$ and $t \in \mathbb{R}$. $\qquad\square$

### E.13 PROOF OF THEOREM C.11

**Theorem** (Theorem C.11). *Let $v \in V$, $u \in \mathcal{N}(v)$ and $t \in \mathbb{R}$ hold. If the function $c_{uv}$ obeys Equation (21) and there exists a pass-through time $\delta t_{uv} \in \mathbb{R}_{\neq 0}$ w.r.t. $e_{uv} \in E$ and $t \in \mathbb{R}$, we obtain*

$$c_{uv}(t, l_{uv}) = c_{uv}(t - \delta t_{uv}, 0) = c_{uv}\left(t - \frac{l_{uv}}{\overline{\nu}_{uv}(t)}, 0\right).$$

*Even more, the concentrations $c_{uv}(\cdot, l_{uv})$ and $c_{vu}(\cdot, l_{vu})$ at the end $z = l_{uv} = l_{vu}$ of the edges $e_{uv} \in E$ and $e_{vu} \in E$ are connected to the concentrations $c_u$ and $c_v$ of the nodes of that edge, respectively, by*

$$\begin{cases} c_{uv}(t, l_{uv}) = c_u(t - \delta t_{uv}) & \text{if } \delta t_{uv} > 0 \\ c_{vu}(t - \delta t_{uv}, l_{vu}) = c_v(t) & \text{if } \delta t_{uv} < 0 \end{cases}.$$

*Proof.* The main idea of this proof is to use the continuity of the function $c_{vu} \in C^0(\mathbb{R} \times [0, l_{vu}], \mathbb{R}_{\geq 0})$ to transfer the results from the advective transport along edges, i.e., Theorem C.5, to the ends of the pipe $e_{uv} \in E$, i.e., to the functions $c_u, c_v \in C^1(\mathbb{R}, \mathbb{R}_{\geq 0})$.

The crucial part is that we would like to apply Theorem C.5 for $t_0 = t - \delta t_{uv}$, $z_0 = 0$ and $t \in \mathbb{R}$ as given. However, the first problem is that $z_0 = 0 \notin (0, l_{uv})$ does not satisfy the condition of Theorem C.5. Moreover, since for this choice,

$$z_0 + \int_{t_0}^{t_0} \nu_{uv}(s)\, ds = z_0 + 0 = 0 \quad \text{and} \quad z_0 + \int_{t_0}^{t} \nu_{uv}(s)\, ds = z_0 + l_{uv} = l_{uv}$$

hold, $t \notin T_{t_0, z_0}$ does also not satisfy the condition of Theorem C.5. Intuitively, we first have to transfer the information from the boundary of the edge $e_{uv}$ to the inside of the edge using the continuity of the function $c_{vu} \in C^1(\mathbb{R} \times [0, l_{vu}], \mathbb{R}_{\geq 0})$. Afterwards, we can use Theorem C.5 for slightly modified $z_0$ and $t$.

To do so, we define the two functions

$$z : (\min\{0, \delta t_{uv}\}, \max\{0, \delta t_{uv}\}) \longrightarrow \mathbb{R}, \ \epsilon \longmapsto z(\epsilon) := \int_{t - \delta t_{uv}}^{t - \epsilon} \nu_{vu}(s) \, ds \text{ and}$$

$$\tilde{z} : (\min\{0, \delta t_{uv}\}, \max\{0, \delta t_{uv}\}) \longrightarrow \mathbb{R}, \ \epsilon \longmapsto \tilde{z}(e) := \frac{l_{uv} - z(\epsilon)}{2}.$$

We first have to prove some technical details regarding these functions (*Step 1*) in order to be able to apply Theorem C.5 as illustrated above (*Step 2*). Afterwards, we can use the results from Lemma C.7 about the sign of the flow at time $t - \delta t_{uv}$ in dependence of the sign of the pass-through time $\delta t_{uv}$ in combination with Definition C.1 to be able to transfer the results from the advective transport along edges to the end of the pipes (*Step 3*).

*Step 1: The functions $z$ and $\tilde{z}$ satisfy the following properties:*

  1. $z$, $\tilde{z}$ and $z + \tilde{z}$ are continuous.

  2. $\mathrm{im}(z) \subset (0, l_{vu})$, $\mathrm{im}(\tilde{z}) \subset (0, \frac{1}{2} l_{vu}) \subset (0, l_{vu})$ and $\mathrm{im}(z + \tilde{z}) \subset (\frac{1}{2}, l_{vu}) \subset (0, l_{vu})$.

  3. $\lim_{\epsilon \to 0} z(\epsilon) = l_{vu}$, $\lim_{\epsilon \to 0} \tilde{z}(\epsilon) = 0$ and $\lim_{\epsilon \to 0} z(\epsilon) + \tilde{z}(\epsilon) = l_{vu}$.

Note that the limit is to be understood for $\epsilon \in (\min\{0, \delta t_{uv}\}, \max\{0, \delta t_{uv}\})$ in the domain of $z$ and $\tilde{z}$, i.e., if $\delta t_{uv} > 0$, $(\min\{0, \delta t_{uv}\}, \max\{0, \delta t_{uv}\}) = (0, \delta t_{uv})$ holds, and the limit $\epsilon \to 0$ is to be understood as the limit $\epsilon \searrow 0, \epsilon < \delta t_{uv}$. In contrast, if $\delta t_{uv} < 0$, $(\min\{0, \delta t_{uv}\}, \max\{0, \delta t_{uv}\}) = (\delta t_{uv}, 0)$ holds, and the limit $\epsilon \to 0$ is to be understood as the limit $\epsilon \nearrow 0, \epsilon > \delta t_{uv}$.

*1.:* By the fundamental theorem of calculus, the function $t \mapsto \int_{t - \delta t_{uv}}^{t} \nu_{vu}(s) \, ds$ is differentiable and therefore, continuous. Therefore, as a composition of continuous functions, the functions $z$, $\tilde{z}$ and $z + \tilde{z}$ are continuous.

*2.:* By (1) basic transformations, (2) definition of $z$, $\tilde{z}$ and $z + \tilde{z}$, respectively, and (3) condition (24), we obtain

$$\mathrm{im}(z) \overset{(1)}{=} \{z(\epsilon) \mid \epsilon \in (\min\{0, \delta t_{uv}\}, \max\{0, \delta t_{uv}\})\}$$

$$\overset{(2)}{=} \left\{ \int_{t - \delta t_{uv}}^{t - \epsilon} \nu_{vu}(s) \, ds \ \middle| \ \epsilon \in (\min\{0, \delta t_{uv}\}, \max\{0, \delta t_{uv}\}) \right\}$$

$$\overset{(1)}{=} \left\{ \int_{t - \delta t_{uv}}^{t - \epsilon} \nu_{vu}(s) \, ds \ \middle| \ t - \epsilon \in (\min\{t, t - \delta t_{uv}\}, \max\{t, t - \delta t_{uv}\}) \right\}$$

$$\overset{(3)}{\subset} (0, l_{uv}).$$

By additionally using that therefore, (4) $z(\epsilon) \in (0, l_{uv})$ holds for all $\epsilon \in (\min\{0, \delta t_{uv}\}, \max\{0, \delta t_{uv}\})$, we obtain

$$\mathrm{im}(\tilde{z}) \overset{(1)}{=} \{\tilde{z}(\epsilon) \mid \epsilon \in (\min\{0, \delta t_{uv}\}, \max\{0, \delta t_{uv}\})$$

$$\overset{(2)}{=} \left\{ \frac{l_{uv} - z(\epsilon)}{2} \ \middle| \ \epsilon \in (\min\{0, \delta t_{uv}\}, \max\{0, \delta t_{uv}\}) \right\}$$

$$\overset{(4)}{\subset} (0, \tfrac{1}{2} l_{uv}) \quad \text{and}$$

$$\mathrm{im}(z + \tilde{z}) \overset{(1)}{=} \{z(\epsilon) + \tilde{z}(\epsilon) \mid \epsilon \in (\min\{0, \delta t_{uv}\}, \max\{0, \delta t_{uv}\})\}$$

$$\overset{(2)}{=} \left\{ z(\epsilon) + \frac{l_{uv} - z(\epsilon)}{2} \;\middle|\; \epsilon \in (\min\{0, \delta t_{uv}\}, \max\{0, \delta t_{uv}\}) \right\}$$

$$\overset{(1)}{=} \left\{ \frac{l_{uv} + z(\epsilon)}{2} \;\middle|\; \epsilon \in (\min\{0, \delta t_{uv}\}, \max\{0, \delta t_{uv}\}) \right\}$$

$$\overset{(4)}{\subset} (\tfrac{1}{2} l_{uv}, l_{uv}).$$

*3.:* By (1) continuity of $z$, $\tilde{z}$ and $z + \tilde{z}$, respectively, (2) definition of $z$, $\tilde{z}$ and $z + \tilde{z}$, respectively and (3) condition (23), we obtain

$$\lim_{\epsilon \to 0} z(\epsilon) \overset{(1)}{=} z(0) \overset{(2)}{=} \int_{t - \delta t_{uv}}^{t} \nu_{vu}(s)\, ds \overset{(3)}{=} l_{uv},$$

$$\lim_{\epsilon \to 0} \tilde{z}(\epsilon) \overset{(1)}{=} \tilde{z}(0) \overset{(2)}{=} \frac{l_{uv} - z(0)}{2} \overset{(2,3)}{=} 0,$$

$$\lim_{\epsilon \to 0} z(\epsilon) + \tilde{z}(\epsilon) \overset{(1)}{=} z(0) + \tilde{z}(0) \overset{(2,3)}{=} l_{uv}.$$

*Step 2:* $c_{uv}(t, l_{uv}) = c_{uv}(t - \delta t_{uv}, 0) = c_{uv}(t - \frac{l_{uv}}{\overline{\nu}_{uv}(t)}, 0)$ *holds.*

For each $\epsilon \in (\min\{0, \delta t_{uv}\}, \max\{0, \delta t_{uv}\})$, we can now choose $t_0 = t - \delta t_{uv} \in \mathbb{R}$ and – as shown in *Step 1* – $z_0 = \tilde{z}(\epsilon) \in (0, l_{vu})$. Consequently, for each $\epsilon \in (\min\{0, \delta t_{uv}\}, \max\{0, \delta t_{uv}\})$, by (1) the choices of $t_0 = t - \delta t_{uv} \in \mathbb{R}$ and $z_0 = \tilde{z}(\epsilon) \in (0, l_{vu})$, (2) the definition of $z$ and $\tilde{z}$ from *Step 1* and (3) the results from *Step 1*, we observe that

$$z_0 + \int_{t_0}^{t - \epsilon} \nu_{uv}(s)\, ds \overset{(1)}{=} \tilde{z}(\epsilon) + \int_{t - \delta t_{uv}}^{t - \epsilon} \nu_{uv}(s)\, ds \overset{(2)}{=} \tilde{z}(\epsilon) + z(\epsilon) \overset{(3)}{\in} (0, l_{uv}),$$

and therefore, $t - \epsilon \in T_{t_0, z_0} = T_{t - \delta t_{uv}, \tilde{z}(\epsilon)}$ holds.[17]

Therefore, by (1) the results from *Step 1*, (2) the continuity of the function $c_{vu} \in C^1(\mathbb{R} \times [0, l_{vu}], \mathbb{R}_{\geq 0})$, (3) definition of $z$ and (4) Theorem C.5 applied to $t_0 = t - \delta t_{uv}$, $z_0 = \tilde{z}(\epsilon) \in (0, l_{vu})$ and $t - \epsilon \in T_{t_0, z_0} = T_{t - \delta t_{uv}, \tilde{z}(\epsilon)}$, we obtain

$$c_{uv}(t, l_{uv}) \overset{(1)}{=} c_{uv}\left( \lim_{\epsilon \to 0} t - \epsilon, \lim_{\epsilon \to 0} z(\epsilon) + \tilde{z}(\epsilon) \right)$$

$$\overset{(2)}{=} \lim_{\epsilon \to 0} c_{uv}\left( t - \epsilon, z(\epsilon) + \tilde{z}(\epsilon) \right)$$

$$\overset{(3)}{=} \lim_{\epsilon \to 0} c_{uv}\left( t - \epsilon, \underbrace{\tilde{z}(\epsilon)}_{\overset{(1)}{\in}(0, l_{uv})} + \underbrace{\int_{t - \delta t_{uv}}^{t - \epsilon} \nu_{vu}(s)\, ds}_{= z(\epsilon) \overset{(1)}{\in} (0, l_{uv})} \right)$$

$$\underbrace{\phantom{xxxxxxxxxxxxxxxxxxxxxx}}_{= z(\epsilon) + \tilde{z}(\epsilon) \overset{(1)}{\in}(0, l_{uv})}$$

$$\overset{(4)}{=} \lim_{\epsilon \to 0} c_{uv}\left( t - \delta t_{uv}, \tilde{z}(\epsilon) \right)$$

$$\overset{(2)}{=} c_{uv}\left( t - \delta t_{uv}, \lim_{\epsilon \to 0} \tilde{z}(\epsilon) \right)$$

$$\overset{(1)}{=} c_{uv}(t - \delta t_{uv}, 0).$$

Finally, as seen in Lemma C.7.3, $\delta t_{uv} = \frac{l_{uv}}{\overline{\nu}_{uv}(t)}$ holds, and we can conclude

$$c_{uv}(t - \delta t_{uv}, 0) = c_{uv}\left( t - \frac{l_{uv}}{\overline{\nu}_{uv}(t)} \right).$$

---

[17]Note that all the technical work of constructing $z$ and $\tilde{z}$ in *Step 1* is because we would actually like to choose $z_0 = 0$ and $t \in \mathbb{R}$ as given, however, this $z_0$ and this $t$ do not satisfy the requirements of Theorem C.5.

*Step 3: If $\delta t_{uv} > 0$ holds, $c_{uv}(t, l_{uv}) = c_u(t - \delta t_{uv})$ holds. If $\delta t_{uv} < 0$ holds, $c_{uv}(t, l_{uv}) = c_{vu}(t - \delta t_{uv}, l_{vu}) = c_v(t)$ holds.*

If $\delta t_{uv} > 0$ holds, by (1) *Step 2* and (2) Definition C.1.2 together with the fact that by Lemma C.7.4, $q_{uv}(t - \delta t_{uv}) \geq 0$ holds, we obtain

$$c_{uv}(t, l_{uv}) \overset{(1)}{=} c_{uv}(t - \delta t_{uv}, 0) \overset{(2)}{=} c_u(t - \delta t_{uv}).$$

If instead, $\delta t_{uv} < 0$ holds, by (1) Definition C.1.1, (2) *Step 2* and (3) Definition C.1.2 together with the fact that by the conservation of flows (Eq. (4)) and Lemma C.7.4, $q_{vu}(t) = -q_{uv}(t) \geq 0$ holds, we obtain

$$c_{vu}(t - \delta t_{uv}, l_{vu}) \overset{(1)}{=} c_{uv}(t - \delta t_{uv}, 0) \overset{(2)}{=} c_{uv}(t, l_{uv}) \overset{(1)}{=} c_{vu}(t, 0) \overset{(3)}{=} c_v(t).$$

$\square$

### E.14 PROOF OF THEOREM C.12

**Theorem** (Theorem C.12). *Let $v \in V$, $u \in \mathcal{N}(v)$ and $t \in \mathbb{R}$ hold. If the function $c_{uv}$ obeys Equation (21) and there exists a self-loop time $\delta t_{uv} \in \mathbb{R}_{\neq 0}$ w.r.t. $e_{uv} \in E$ and $t \in \mathbb{R}$, we obtain*
$$c_{uv}(t, 0) = c_{uv}(t - \delta t_{uv}, 0).$$

*Even more, the concentration $c_{vu}(\cdot, l_{vu})$ at the end $z = l_{vu} = l_{uv}$ of the edge $e_{vu} \in E$ is connected to the concentration $c_u$ of the node of that edge by*
$$\begin{cases} c_{vu}(t, l_{vu}) = c_u(t - \delta t_{uv}) & \text{if } \delta t_{uv} > 0 \\ c_{vu}(t - \delta t_{uv}, l_{vv}) = c_u(t) & \text{if } \delta t_{uv} < 0 \end{cases}.$$

The proof of Theorem C.12 is very similar to the proof of Theorem C.11 and only requires a few changes. The interested reader can compare both proofs and investigate why the changes are required, and how they induce the different results in Theorem C.12 as compared to Theorem C.11.

*Proof.* The main idea of this proof is to use the continuity of the function $c_{vu} \in C^1(\mathbb{R} \times [0, l_{vu}], \mathbb{R}_{\geq 0})$ to transfer the results from the advective transport along edges, i.e., Theorem C.5, to the end of the pipe $e_{uv} \in E$, i.e., to the function $c_u \in C^1(\mathbb{R}, \mathbb{R}_{\geq 0})$.

The crucial part is that we would like to apply Theorem C.5 for $t_0 = t - \delta t_{uv}$, $z_0 = 0$ and $t \in \mathbb{R}$ as given. However, the first problem is that $z_0 = 0 \notin (0, l_{uv})$ does not satisfy the condition of Theorem C.5. Moreover, since for this choice,

$$z_0 + \int_{t_0}^{t_0} \nu_{uv}(s) \, ds = z_0 + 0 = 0 \quad \text{and} \quad z_0 + \int_{t_0}^{t} \nu_{uv}(s) \, ds = z_0 + 0 = 0$$

hold, $t \notin T_{t_0, z_0}$ does also not satisfy the condition of Theorem C.5. Intuitively, we first have to transfer the information from the boundary of the edge $e_{uv}$ to the inside of the edge using the continuity of the function $c_{vu} \in C^1(\mathbb{R} \times [0, l_{vu}], \mathbb{R}_{\geq 0})$. Afterwards, we can use Theorem C.5 for slightly modified $z_0$ and $t$.

To do so, we define the two functions

$$z : (\min\{0, \delta t_{uv}\}, \max\{0, \delta t_{uv}\}) \longrightarrow \mathbb{R}, \ \epsilon \longmapsto z(\epsilon) := \int_{t - \delta t_{uv}}^{t - \epsilon} \nu_{vu}(s) \, ds \text{ and}$$

$$\tilde{z} : (\min\{0, \delta t_{uv}\}, \max\{0, \delta t_{uv}\}) \longrightarrow \mathbb{R}, \ \epsilon \longmapsto \tilde{z}(e) := \frac{z(\epsilon)}{2}.$$

We first have to prove some technical details regarding these functions (*Step 1*) in order to be able to apply Theorem C.5 as illustrated above (*Step 2*). Afterwards, we can use the results from Lemma C.8 about the sign of the flow at time $t - \delta t_{uv}$ in dependence of the sign of the self-loop time $\delta t_{uv}$ in combination with Definition C.1 to be able to transfer the results from the advective transport along edges to the end of the pipe (*Step 3*).

*Step 1: The functions $z$ and $\tilde{z}$ satisfy the following properties:*

1. $z$, $\tilde{z}$ and $z + \tilde{z}$ are continuous.

2. $\operatorname{im}(z) \subset (0, l_{vu})$, $\operatorname{im}(\tilde{z}) \subset (0, \frac{1}{2}l_{vu}) \subset (0, l_{vu})$ and $\operatorname{im}(z + \tilde{z}) \subset (0, \frac{3}{2}l_{vu}) \subset (0, l_{vu})$.

3. $\lim_{\epsilon \to 0} z(\epsilon) = 0$, $\lim_{\epsilon \to 0} \tilde{z}(\epsilon) = 0$ and $\lim_{\epsilon \to 0} z(\epsilon) + \tilde{z}(\epsilon) = 0$.

Note that the limit is to be understood for $\epsilon \in (\min\{0, \delta t_{uv}\}, \max\{0, \delta t_{uv}\})$ in the domain of $z$ and $\tilde{z}$, i.e., if $\delta t_{uv} > 0$, $(\min\{0, \delta t_{uv}\}, \max\{0, \delta t_{uv}\}) = (0, \delta t_{uv})$ holds, and the limit $\epsilon \to 0$ is to be understood as the limit $\epsilon \searrow 0, \epsilon < \delta t_{uv}$. In contrast, if $\delta t_{uv} < 0$, $(\min\{0, \delta t_{uv}\}, \max\{0, \delta t_{uv}\}) = (\delta t_{uv}, 0)$ holds, and the limit $\epsilon \to 0$ is to be understood as the limit $\epsilon \nearrow 0, \epsilon > \delta t_{uv}$.

*1.:* By the fundamental theorem of calculus, the function $t \mapsto \int_{t - \delta t_{uv}}^{t} \nu_{vu}(s) \, ds$ differentiable and therefore, continuous. Therefore, as compositions of continuous functions, the functions $z$, $\tilde{z}$ and $z + \tilde{z}$ are continuous.

*2.:* By (1) basic transformations, (2) definition of $z$, $\tilde{z}$ and $z + \tilde{z}$, respectively, and (3) condition (28), we obtain

$$\operatorname{im}(z) \stackrel{(1)}{=} \{z(\epsilon) \mid \epsilon \in (\min\{0, \delta t_{uv}\}, \max\{0, \delta t_{uv}\})\}$$

$$\stackrel{(2)}{=} \left\{ \int_{t - \delta t_{uv}}^{t - \epsilon} \nu_{vu}(s) \, ds \;\middle|\; \epsilon \in (\min\{0, \delta t_{uv}\}, \max\{0, \delta t_{uv}\}) \right\}$$

$$\stackrel{(1)}{=} \left\{ \int_{t - \delta t_{uv}}^{t - \epsilon} \nu_{vu}(s) \, ds \;\middle|\; t - \epsilon \in (\min\{t, t - \delta t_{uv}\}, \max\{t, t - \delta t_{uv}\}) \right\}$$

$$\stackrel{(3)}{\subset} (0, l_{uv}).$$

By additionally using that therefore, (4) $z(\epsilon) \in (0, l_{uv})$ holds for all $\epsilon \in (\min\{0, \delta t_{uv}\}, \max\{0, \delta t_{uv}\})$, we obtain

$$\operatorname{im}(\tilde{z}) \stackrel{(1)}{=} \{\tilde{z}(\epsilon) \mid \epsilon \in (\min\{0, \delta t_{uv}\}, \max\{0, \delta t_{uv}\})$$

$$\stackrel{(2)}{=} \left\{ \frac{z(\epsilon)}{2} \;\middle|\; \epsilon \in (\min\{0, \delta t_{uv}\}, \max\{0, \delta t_{uv}\}) \right\}$$

$$\stackrel{(4)}{\subset} (0, \tfrac{1}{2}l_{uv}) \quad \text{and}$$

$$\operatorname{im}(z + \tilde{z}) \stackrel{(1)}{=} \{z(\epsilon) + \tilde{z}(\epsilon) \mid \epsilon \in (\min\{0, \delta t_{uv}\}, \max\{0, \delta t_{uv}\})\}$$

$$\stackrel{(2)}{=} \left\{ z(\epsilon) + \frac{z(\epsilon)}{2} \;\middle|\; \epsilon \in (\min\{0, \delta t_{uv}\}, \max\{0, \delta t_{uv}\}) \right\}$$

$$\stackrel{(1)}{=} \left\{ \frac{3}{2} z(\epsilon) \;\middle|\; \epsilon \in (\min\{0, \delta t_{uv}\}, \max\{0, \delta t_{uv}\}) \right\}$$

$$\stackrel{(4)}{\subset} (0, \tfrac{3}{2}l_{uv}).$$

*3.:* By (1) continuity of $z$, $\tilde{z}$ and $z + \tilde{z}$, respectively, (2) definition of $z$, $\tilde{z}$ and $z + \tilde{z}$, respectively and (3) condition (27), we obtain

$$\lim_{\epsilon \to 0} z(\epsilon) \stackrel{(1)}{=} z(0) \stackrel{(2)}{=} \int_{t - \delta t_{uv}}^{t} \nu_{vu}(s) \, ds \stackrel{(3)}{=} 0,$$

$$\lim_{\epsilon \to 0} \tilde{z}(\epsilon) \stackrel{(1)}{=} \tilde{z}(0) \stackrel{(2)}{=} \frac{z(0)}{2} \stackrel{(2,3)}{=} 0,$$

$$\lim_{\epsilon \to 0} z(\epsilon) + \tilde{z}(\epsilon) \stackrel{(1)}{=} z(0) + \tilde{z}(0) \stackrel{(2,3)}{=} 0.$$

*Step 2: $c_{uv}(t, 0) = c_{uv}(t - \delta t_{uv}, 0)$ holds.*

For each $\epsilon \in (\min\{0, \delta t_{uv}\}, \max\{0, \delta t_{uv}\})$, we can now choose $t_0 = t - \delta t_{uv} \in \mathbb{R}$ and – as shown in *Step 1* – $z_0 = \tilde{z}(\epsilon) \in (0, l_{vu})$. Consequently, for each $\epsilon \in (\min\{0, \delta t_{uv}\}, \max\{0, \delta t_{uv}\})$, by (1)

the choices of $t_0 = t - \delta t_{uv} \in \mathbb{R}$ and $z_0 = \tilde{z}(\epsilon) \in (0, l_{vu})$, (2) the definition of $z$ and $\tilde{z}$ from *Step 1* and (3) the results from *Step 1*, we observe that

$$z_0 + \int_{t_0}^{t-\epsilon} \nu_{uv}(s)\, ds \overset{(1)}{=} \tilde{z}(\epsilon) + \int_{t-\delta t_{uv}}^{t-\epsilon} \nu_{uv}(s)\, ds \overset{(2)}{=} \tilde{z}(\epsilon) + z(\epsilon) \overset{(3)}{\in} (0, l_{uv}),$$

and therefore, $t - \epsilon \in T_{t_0, z_0} = T_{t-\delta t_{uv}, \tilde{z}(\epsilon)}$ holds.[18]

Therefore, by (1) the results from *Step 1*, (2) the continuity of the function $c_{vu} \in C^1(\mathbb{R} \times [0, l_{vu}], \mathbb{R}_{\geq 0})$, (3) definition of $z$ and (4) Theorem C.5 applied to $t_0 = t - \delta t_{uv}, z_0 = \tilde{z}(\epsilon) \in (0, l_{vu})$ and $t - \epsilon \in T_{t_0, z_0} = T_{t-\delta t_{uv}, \tilde{z}(\epsilon)}$, we obtain

$$c_{uv}(t, 0) \overset{(1)}{=} c_{uv}\left(\lim_{\epsilon \to 0} t - \epsilon, \lim_{\epsilon \to 0} z(\epsilon) + \tilde{z}(\epsilon)\right)$$

$$\overset{(2)}{=} \lim_{\epsilon \to 0} c_{uv}(t - \epsilon, z(\epsilon) + \tilde{z}(\epsilon))$$

$$\overset{(3)}{=} \lim_{\epsilon \to 0} c_{uv}\left(t - \epsilon,\ \underbrace{\tilde{z}(\epsilon)}_{\overset{(1)}{\in}(0, l_{uv})} + \underbrace{\underbrace{\int_{t-\delta t_{uv}}^{t-\epsilon} \nu_{vu}(s)\, ds}_{=z(\epsilon)\overset{(1)}{\in}(0, l_{uv})}}_{=z(\epsilon)+\tilde{z}(\epsilon)\overset{(1)}{\in}(0, l_{uv})}\right)$$

$$\overset{(4)}{=} \lim_{\epsilon \to 0} c_{uv}(t - \delta t_{uv}, \tilde{z}(\epsilon))$$

$$\overset{(2)}{=} c_{uv}\left(t - \delta t_{uv}, \lim_{\epsilon \to 0} \tilde{z}(\epsilon)\right)$$

$$\overset{(1)}{=} c_{uv}(t - \delta t_{uv}, 0).$$

*Step 3:* If $\delta t_{uv} > 0$ holds, $c_{vu}(t, l_{uv}) = c_u(t - \delta t_{uv})$ holds. If $\delta t_{uv} < 0$ holds, $c_{vu}(t - \delta t_{uv}, l_{vu}) = c_u(t)$ holds.

If $\delta t_{uv} > 0$ holds, by (1) Definition C.1.1, (2) *Step 2* and (3) Definition C.1.2 together with the fact that by Lemma C.8.4, $q_{uv}(t - \delta t_{uv}) \geq 0$ holds, we obtain

$$c_{vu}(t, l_{uv}) \overset{(1)}{=} c_{uv}(t, 0) \overset{(2)}{=} c_{uv}(t - \delta t_{uv}, 0) \overset{(3)}{=} c_u(t - \delta t_{uv}).$$

If instead, $\delta t_{uv} < 0$ holds, by (1) Definition C.1.1, (2) *Step 2* and (3) Definition C.1.2 together with the fact that by Lemma C.8.4, $q_{uv}(t) \geq 0$ holds, we obtain

$$c_{vu}(t - \delta t_{uv}, l_{vu}) \overset{(1)}{=} c_{uv}(t - \delta t_{uv}, 0) \overset{(2)}{=} c_{uv}(t, 0) \overset{(3)}{=} c_u(t).$$

$\square$

### E.15 PROOF OF THEOREM C.13

**Theorem** (Theorem C.13). *Let $v \in V$, $u \in \mathcal{N}(v)$ and $t \in \mathbb{R}$ hold. If $\nu_{uv}(t) \neq 0$ holds and the set*

$$A_{uv}(t) := \left\{ \delta t \in (0, \infty) \mid \int_{t-\delta t}^{t} \nu_{uv}(s)\, ds \in \{-l_{uv}, 0, l_{uv}\} \right\}$$

$$= \left\{ \delta t \in (0, \infty) \mid z(\delta t) \in \{-l_{uv}, 0, l_{uv}\} \right\}$$

*is not empty, the following properties hold:*

  *1. $\delta t_{uv} := \min A_{uv}(t) = \{\delta t \in (0, \infty) \mid z(\delta t) \in \{-l_{uv}, 0, l_{uv}\}\} \in (0, \infty)$ exists.*

---

[18]Note that all the technical work of constructing $z$ and $\tilde{z}$ in *Step 1* is because we would actually like to choose $z_0 = 0$ and $t \in \mathbb{R}$ as given; however, this $z_0$ and this $t$ do not satisfy the requirements of Theorem C.5.

2. $\delta t_{uv}$ *is a positive pass-through time, inverse pass-through time, self-loop time or inverse self-loop time. More specifically, $\delta t_{uv}$ is (always w.r.t. $e_{uv}$ and $t$) ...*

*... an inverse pass-through time* $\qquad \Longleftrightarrow \; \mathbb{z}(\delta t_{uv}) = -l_{uv},$

*... a self-loop time or an inverse self-loop time* $\qquad \Longleftrightarrow \; \mathbb{z}(\delta t_{uv}) = 0,$

*... a pass-through time* $\qquad \Longleftrightarrow \; \mathbb{z}(\delta t_{uv}) = l_{uv}.$

3. *There exists no other positive pass-through time, inverse pass-through time, self-loop time or inverse self-loop time than $\delta t_{uv}$.*

*Proof.* By the fundamental theorem of calculus, the function $t^{'} \mapsto \int_{t}^{t^{'}} \nu_{vu}(s) \, ds$ is differentiable and therefore, continuous. Therefore, as a composition of continuous functions, the function $\mathbb{z}$ is continuous.

*1:* By assumption, $A_{uv}(t) \neq \emptyset$ holds. Moreover, by definition, the value $0 \in \mathbb{R}$ is a lower bound of $A_{uv}(t) \subset (0, \infty)$. Therefore, it suffices to show that $A_{uv}(t)$ is closed in $\mathbb{R}$; in this case, the minimum of $A_{uv}(t)$ exists.

To do so, let $(\delta t_n)_{n \in \mathbb{N}}$ be a sequence in $A_{uv}(t)$ such that $\delta t_n \to \delta t_0$ holds for $n \to \infty$. We need to show that $\delta t_0 \in A_{uv}(t)$ holds.

*Step 1.1:* $\mathbb{z}(\delta t_0) \in \{-l_{uv}, 0, l_{uv}\}$ *holds.*

On the one hand, by the continuity of $\mathbb{z}$, we obtain that $\mathbb{z}(\delta t_n) \to \mathbb{z}(\delta t_0)$ holds for $n \to \infty$. One the other hand, since $(\delta t_n)_{n \in \mathbb{N}}$ is a sequence in $A_{uv}(t)$, $(\mathbb{z}(\delta t_n))_{n \in \mathbb{N}}$ is a sequence in $\{-l_{uv}, 0, l_{uv}\}$. Consequently, since $\{-l_{uv}, 0, l_{uv}\}$ is closed in $\mathbb{R}$, the limit $\mathbb{z}(\delta t_0)$ is also an element of $\{-l_{uv}, 0, l_{uv}\}$.

*Step 1.2:* $\delta t_0 > 0$ *holds.*

Since $(\delta t_n)_{n \in \mathbb{N}}$ is a sequence in $A_{uv}(t)$, $\delta t_n > 0$ holds for a $n \in \mathbb{N}$. Consequently, the limit $\delta t_0 = \lim_{n \to \infty} \delta t_n$ satisfies $\delta_0 \geq 0$, and it suffices to show that $\delta t_0 \neq 0$ holds.

Let us assume that $\delta t_0 = 0$ holds. By the continuity of $\nu_{uv}$ and the assumption that $\nu_{uv}(t) \neq 0$ holds, there exists an $\epsilon > 0$ such that $\nu_{uv}(s) > 0$ or $\nu_{uv}(s) < 0$ holds for all $s \in (t - \epsilon, t + \epsilon)$. Consequently, (1) by basic results from analysis,

$$\mathbb{z}(\delta t) = \int_{t-\delta t}^{t} \underbrace{\nu_{uv}(s)}_{>0 \text{ or } <0} \, ds \overset{(1)}{\neq} 0$$

holds for all $\delta t \in (0, \epsilon)$. Specifically, since $\delta t_n \to \delta t_0 = 0$ holds for $n \to \infty$, for the $\epsilon > 0$ from above, there exists an $n_0 \in \mathbb{N}$ such that for all $n \geq n_0$, $\delta t_n \in (0, \epsilon)$ holds. Consequently, for all $n \geq n_0$, $\mathbb{z}(\delta t_n) > 0$ holds. However, by the continuity of $\mathbb{z}$, we obtain that $\mathbb{z}(\delta t_n) \to \mathbb{z}(\delta t_0) = \mathbb{z}(0) = 0$ holds for $n \to \infty$ – a contradiction.

*2:* For $\delta t_{uv} := \min A_{uv}(t)$, we show an even stronger claim (namely Lemma C.15.1) in several steps.

*Step 2.1:* $\text{im} \, \mathbb{z}_{|(0, \delta t_{uv})} \subset (-l_{uv}, 0) \sqcup (0, l_{uv})$ *holds.*

Let us assume that there exists a $\delta t \in (0, \delta t_{uv})$ such that $\mathbb{z}(\delta t) \in (-\infty, -l_{uv}] \sqcup \{0\} \sqcup [l_{uv}, \infty)$ holds.
*Case 1:* If $\mathbb{z}(\delta t) \in \{-l_{uv}, 0, l_{uv}\}$ holds, we obtain $\delta t \in A_{uv}(t)$ with $\delta t < \delta t_{uv}$ – a contradiction to the choice of $\delta t_{uv} = \min A_{uv}(t)$ as the minimum of $A_{uv}(t)$.
*Case 2:* If $\mathbb{z}(\delta t) \in (-\infty, -l_{uv})$ holds, by the fact that $\mathbb{z}(0) = 0$ holds and the intermediate value theorem, there exists a $\delta t^{'} \in (0, \delta t)$, such that $\mathbb{z}(\delta t^{'}) = -l_{uv}$ holds. Consequently, we obtain $\delta t^{'} \in A_{uv}(t)$ with $\delta t^{'} < \delta t_{uv}$ – a contradiction to the choice of $\delta t_{uv} = \min A_{uv}(t)$ as the minimum of $A_{uv}(t)$.
*Case 3:* If $\mathbb{z}(\delta t) \in (l_{uv}, \infty)$ holds, by the fact that $\mathbb{z}(0) = 0$ holds and the intermediate value

theorem, there exists a $\delta t^{'} \in (0, \delta t)$, such that $z(\delta t^{'}) = l_{uv}$ holds. Consequently, we obtain $\delta t^{'} \in A_{uv}(t)$ with $\delta t^{'} < \delta t_{uv}$ – a contradiction to the choice of $\delta t_{uv} = \min A_{uv}(t)$ as the minimum of $A_{uv}(t)$.

*Step 2.2: Either* $\operatorname{im} z_{|(0,\delta t_{uv})} \subset (-l_{uv}, 0)$ *or* $\operatorname{im} z_{|(0,\delta t_{uv})} \subset (0, l_{uv})$ *holds.*

By step *Step 2.1*, $\operatorname{im} z_{|(0,\delta t_{uv})} \subset (-l_{uv}, 0) \sqcup (0, l_{uv})$ holds. Let us assume that there exist $\delta t_1, \delta t_2 \in (0, \delta t_{uv})$ such that $z(\delta t_1) \in (-l_{uv}, 0)$ and $z(\delta t_2) \in (0, l_{uv})$ holds. Again by the mean value theorem, there exists a $\delta t \in (\delta t_1, \delta_2)$, such that $z(\delta t) = 0$ holds. Consequently, we obtain $\delta t \in A_{uv}(t)$ with $\delta t < \delta t_2 < \delta t_{uv}$ – a contradiction to the choice of $\delta t_{uv} = \min A_{uv}(t)$ as the minimum of $A_{uv}(t)$.

*Step 2.3: If additionally,* $z(\delta t_{uv}) = -l_{uv}$ *holds, then* $\operatorname{im} z_{|(0,\delta t_{uv})} \subset (-l_{uv}, 0)$ *holds. If in contrast, additionally,* $z(\delta t_{uv}) = l_{uv}$ *holds, then* $\operatorname{im} z_{|(0,\delta t_{uv})} \subset (0, l_{uv})$ *holds.*

By step *Step 2.2*, either $\operatorname{im} z_{|(0,\delta t_{uv})} \subset (-l_{uv}, 0)$ or $\operatorname{im} z_{|(0,\delta t_{uv})} \subset (0, l_{uv})$ holds. If additionally, $z(\delta t_{uv}) = -l_{uv}$ holds, let us assume that there exists a $\delta t \in (0, \delta t_{uv})$ such that $z(\delta t) \in (0, l_{uv})$ holds. Again by the intermediate value theorem, there exists a $\delta t^{'} \in (\delta t, \delta t_{uv})$, such that $z(\delta t^{'}) = 0$ holds. Consequently, we obtain $\delta t^{'} \in A_{uv}(t)$ with $\delta t^{'} < \delta t_{uv}$ – a contradiction to the choice of $\delta t_{uv} = \min A_{uv}(t)$ as the minimum of $A_{uv}(t)$.

If in contrast, $z(\delta t_{uv}) = l_{uv}$ holds, let us assume that there exists a $\delta t \in (0, \delta t_{uv})$ such that $z(\delta t) \in (-l_{uv}, 0)$ holds. Again by the intermediate value theorem, there exists a $\delta t^{'} \in (\delta t, \delta t_{uv})$, such that $z(\delta t^{'}) = 0$ holds. Consequently, we obtain $\delta t^{'} \in A_{uv}(t)$ with $\delta t^{'} < \delta t_{uv}$ – a contradiction to the choice of $\delta t_{uv} = \min A_{uv}(t)$ as the minimum of $A_{uv}(t)$.

*Step 2.4: For* $\delta t_{uv} := \min A_{uv}(t)$*, we obtain*

$$
\begin{cases}
\operatorname{im} z_{|(0,\delta t_{uv})} \subset (-l_{uv}, 0) & \text{if } z(\delta t_{uv}) = -l_{uv}, \\
\operatorname{im} z_{|(0,\delta t_{uv})} \subset (-l_{uv}, 0) \text{ or } \operatorname{im} z_{|(0,\delta t_{uv})} \subset (0, l_{uv}) & \text{if } z(\delta t_{uv}) = 0, \\
\operatorname{im} z_{|(0,\delta t_{uv})} \subset (0, l_{uv}) & \text{if } z(\delta t_{uv}) = l_{uv}.
\end{cases}
$$

The claim follows immediately from *Step 2.2* and *Step 2.3*.

*Step 2.5:* $\delta t_{uv}$ *is (always w.r.t.* $e_{uv}$ *and* $t$*) ...*

| ... an inverse pass-through time | $\iff z(\delta t_{uv}) = -l_{uv}$, |
| ... a self-loop time or an inverse self-loop time | $\iff z(\delta t_{uv}) = 0$, |
| ... a pass-through time | $\iff z(\delta t_{uv}) = l_{uv}$. |

"$\Rightarrow$" If $\delta t_{uv}$ is an inverse pass-through time, by (1) definition of $z$ and (2) condition (25), we obtain

$$
z(\delta t_{uv}) \overset{(1)}{=} \int_{t-\delta t_{uv}}^{t} \nu_{uv}(s) \, ds \overset{(2)}{=} -l_{uv}.
$$

Analogously, if $\delta t_{uv}$ is a self-loop time or an inverse self-loop time, we use condition (27) or (29) to conclude that $z(\delta t_{uv}) = 0$ holds. If $\delta t_{uv}$ is a pass-through time, we use condition (23) to conclude that $z(\delta t_{uv}) = l_{uv}$ holds.

"$\Leftarrow$" If $z(\delta t_{uv}) = -l_{uv}$ holds, by (1) definition of $z$ and (2) this assumption, we obtain

$$
\int_{t-\delta t_{uv}}^{t} \nu_{uv}(s) \, ds \overset{(1)}{=} z(\delta t_{uv}) \overset{(2)}{=} -l_{uv}.
$$

that is, condition (25) is satisfied.

Consequently, for any $t^{'} \in (\min\{t - \delta t_{uv}, t\}, \max\{t - \delta t_{uv}, t\}) = (t - \delta t_{uv}, t)$, or equivalently, for any $t^{'} = t - \delta t$ with $\delta t \in (0, \delta t_{uv})$, by (1) by basic results from analysis, (2) condition (25), (3) definition of $z$ and (4) *Step 2.4*, we obtain

$$\int_{t-\delta t_{uv}}^{t^{'}} \nu_{uv}(s)\, ds \overset{(1)}{=} \underbrace{\int_{t-\delta t_{uv}}^{t} \nu_{uv}(s)\, ds}_{\overset{(2)}{=}-l_{uv}} - \underbrace{\int_{t-\delta t}^{t} \nu_{uv}(s)\, ds}_{\overset{(3)}{=}z(\delta t)\overset{(4)}{\in}(-l_{uv},0)} \overset{(1)}{\in} (-l_{uv},0),$$

that is, condition (26) is satisfied, too, and thus, $\delta t_{uv}$ is an inverse pass-through time.

Analogously, if $z(\delta t_{uv}) = 0$ holds, we this assumption to show that condition (27), which is equal to condition (29), is satisfied. If $z(\delta t_{uv}) = l_{uv}$ holds, we use this assumption to show that condition (23) is satisfied. Consequently, we use these conditions and *Step 2.4* to show that condition (28), (30) and (24) are satisfied in alignment with the conditions in *Step 2.4*. We leave the details as an exercise to the reader, however, a summary of the to be considered cases are given as follows:

$$\underbrace{\int_{t-\delta t_{uv}}^{t} \nu_{uv}(s)\, ds}_{} - \underbrace{\int_{t-\delta t}^{t} \nu_{uv}(s)\, ds}_{} \overset{(1)}{=} \underbrace{\int_{t-\delta t_{uv}}^{t^{'}=t-\delta t} \nu_{uv}(s)\, ds}_{}$$

| | | | |
|---|---|---|---|
| $= -l_{uv}$ | $\in (-l_{uv},0)$ | $\in (-l_{uv},0)$ | if $z(\delta t_{uv}) = -l_{uv}$ |
| $= 0$ | $\in (-l_{uv},0)$ | $\in (0,l_{uv})$ | if $z(\delta t_{uv}) = 0$ and $\operatorname{im} z_{|(0,\delta t_{uv})} \subset (-l_{uv},0)$ |
| $= 0$ | $\in (0,l_{uv})$ | $\in (-l_{uv},0)$ | if $z(\delta t_{uv}) = 0$ and $\operatorname{im} z_{|(0,\delta t_{uv})} \subset (0,l_{uv})$ |
| $= l_{uv}$ | $\in (0,l_{uv})$ | $\in (0,l_{uv})$ | if $z(\delta t_{uv}) = l_{uv}$ |

*3.:* Similar to the proof of Lemma C.7.2 and Lemma C.8.1, let us assume that there exist two different and positive times $\delta t_{1uv}, \delta t_{2uv} \in \mathbb{R}_{\neq 0}$ which are each either a pass-through time, an inverse pass-through time, a self-loop time or an inverse self-loop time. W.l.o.g., let $0 < \delta t_{1uv} < \delta t_{2uv}$ hold. Consequently, for $t^{'} = t - \delta t_{1uv} \in (t - \delta t_{2uv}, t) = (\min\{t - \delta t_{2uv}, t\}, \max\{t - \delta t_{2uv}, t\})$, by (1) by basic results from analysis and (2) one or two of the conditions (23), (25), (27) or (29)[19], we obtain

$$\int_{t-\delta t_{2uv}}^{t^{'}} \nu_{uv}(s)\, ds \overset{(1)}{=} \int_{t-\delta t_{2uv}}^{t} \nu_{uv}(s)\, ds - \int_{t-\delta t_{1uv}}^{t} \nu_{uv}(s)\, ds \overset{(2)}{\in} \{-2l_{uv}, -l_{uv}, 0, l_{uv}, 2l_{uv}\}$$

– a contradiction to one of the conditions (24), (26), (28) or (30)[20]. $\qquad\square$

### E.16 PROOF OF LEMMA C.15

**Lemma** (Lemma C.15). *Let $v \in V$, $u \in \mathcal{N}(v)$ and $t \in \mathbb{R}$ hold. In the setting of Theorem C.13, the following property holds:*

1. *$\delta t_{uv}$ is (always w.r.t. $e_{uv}$ and $t$) ...*

   *... a pass-through time* $\qquad\qquad \iff z(\delta t_{uv}) \in \{-l_{uv}, l_{uv}\}$ *and* $q_{uv}(t) > 0$,
   *... an inverse pass-through time* $\qquad \iff z(\delta t_{uv}) \in \{-l_{uv}, l_{uv}\}$ *and* $q_{uv}(t) < 0$,
   *... a self-loop time* $\qquad\qquad\qquad \iff z(\delta t_{uv}) = 0$ *and* $q_{uv}(t) < 0$,
   *... an inverse self-loop time* $\qquad \iff z(\delta t_{uv}) = 0$ *and* $q_{uv}(t) > 0$.

*Proof.* "$\Rightarrow$" If $\delta t_{uv}$ is a pass-through time, similarly to *Step 2.5* in the proof of Theorem C.13.2, by (1) definition of $z$ and (2) condition (23), we obtain

$$z(\delta t_{uv}) \overset{(1)}{=} \int_{t-\delta t_{uv}}^{t} \nu_{uv}(s)\, ds \overset{(2)}{=} l_{uv} \in \{-l_{uv}, l_{uv}\}.$$

---

[19]Note that we have $3 \cdot 3 = 9$ possible combinations: If $\delta t_{2uv}$ is a pass-through time, we work with condition (23). If $\delta t_{2uv}$ is an inverse pass-through time, we work with condition (25). If $\delta t_{2uv}$ is a(n) (inverse) self-loop time, we work with condition (27), which is equal to condition (29). Each of these three options need to be combined with the three analogous options for the second time $\delta t_{1uv}$.

[20]Note that we have four possible combinations: If $\delta t_{2uv}$ is a pass-through time, we work with condition (24). If $\delta t_{2uv}$ is an inverse pass-through time, we work with condition (26). If $\delta t_{2uv}$ is a self-loop time, we work with condition (28). If $\delta t_{2uv}$ is an inverse self-loop time, we work with condition (30).

Moreover, by Lemma C.7 and the choice of $\delta t_{uv} > 0$ , $q_{uv}(t) \geq 0$ holds. Since $q_{uv}(t) \neq 0$ holds by assumption, $q_{uv}(t) > 0$ follows.

Similarly, if $\delta t_{uv}$ is an inverse pass-through time (w.r.t. $e_{uv}$ and $t$), by (1) definition of $z$ and (2) condition (25), we obtain

$$z(\delta t_{uv}) \overset{(1)}{=} \int_{t-\delta t_{uv}}^{t} \nu_{uv}(s) \, ds \overset{(2)}{=} -l_{uv} \in \{-l_{uv}, l_{uv}\}.$$

Additionally, we use Lemma C.9 to conclude that $\delta t_{uv}$ is a pass-through time w.r.t. $e_{vu}$ and $t$. Therefore, by (1) the conservation of flows (Eq. (4)), (2) Lemma C.7 and the choice of $\delta t_{uv} > 0$ considered as a pass-through time w.r.t. $e_{vu}$ and $t$, we obtain

$$q_{uv}(t) \overset{(1)}{=} - \underbrace{q_{vu}(t)}_{\substack{(2) \\ \geq 0}} \leq 0.$$

Since $q_{uv}(t) \neq 0$ holds by assumption, $q_{uv}(t) < 0$ follows.

Analogously, if $\delta t_{uv}$ is a self-loop time, we use condition (27) to conclude that $z(\delta t_{uv}) = 0$ holds and Lemma C.8 to conclude that $q_{uv}(t) \leq 0$ and thus, $q_{uv}(t) < 0$ holds. If $\delta t_{uv}$ is an inverse self-loop time, we use condition (29) to conclude that $z(\delta t_{uv}) = 0$ holds and Lemma C.10, the conservation of flows and Lemma C.8 to conclude that $q_{uv}(t) = -q_{vu}(t) \geq 0$ and thus, $q_{uv}(t) > 0$ holds.

"$\Leftarrow$" If $z(\delta t_{uv}) \in \{-l_{uv}, l_{uv}\}$ holds, we know by Theorem C.13.2 that $\delta t_{uv}$ is either a pass-through time or an inverse pass-through time.

If additionally, $q_{uv}(t) > 0$ holds, let us assume that $\delta t_{uv}$ is an inverse pass-through time. Consequently, by the same arguments as in the inclusion above, we can conclude that in this case, $q_{uv}(t) = -q_{vu}(t) \leq 0$ holds – a contradiction.

If instead, additionally, $q_{uv}(t) < 0$ holds, let us assume that $\delta t_{uv}$ is a pass-through time. Consequently, by the same arguments as in the inclusion above, we can conclude that in this case, $q_{uv}(t) \geq 0$ holds – a contradiction.

If $z(\delta t_{uv}) = 0$ holds, we know by Theorem C.13.2 that $\delta t_{uv}$ is either a self-loop time or an inverse self-loop time.

If additionally, $q_{uv}(t) < 0$ holds, let us assume that $\delta t_{uv}$ an inverse self-loop time. Consequently, by the same arguments as in the inclusion above, we can conclude that in this case, $q_{uv}(t) \geq 0$ holds – a contradiction.

If instead, additionally, $q_{uv}(t) > 0$ holds, let us assume that $\delta t_{uv}$ a self-loop time. Consequently, by the same arguments as in the inclusion above, we can conclude that in this case, $q_{uv}(t) \leq 0$ holds – a contradiction. $\qquad\square$

### E.17 PROOF OF THEOREM C.16

**Theorem** (Theorem C.16). *Let $v \in V$ and $t \in \mathbb{R}$ hold. We assume that for each $u \in \mathcal{N}(v)$, the set $A_{uv}(t)$ as defined in Theorem C.13 is not empty. If the function $c_v$ obeys Equation (22) and if for each $u \in \mathcal{N}(v)$, the function $c_{uv}$ obeys Equation (21), we obtain*

$$c_v(t) = \frac{1}{s} \left( \sum_{\substack{u \in \mathcal{N}(v) \\ \delta t_{uv} \in A_{uv}(\{-l_{uv}, l_{uv}\})}} \mathrm{ReLU}(q_{uv}(t)) \cdot c_u(t - \delta t_{uv}) \right.$$

$$\sum_{\substack{u \in \mathcal{N}(v) \\ \delta t_{uv} \in A_{uv}(0)}} \mathrm{ReLU}(q_{uv}(t)) \cdot c_v(t - \delta t_{uv})$$

$$\left. q_{v,ext.}(t) \cdot c_{v,ext.}(t) \right)$$

*with scaling factor*

$$s = \sum_{u \in \mathcal{N}(v)} \text{ReLU}(q_{uv}(t)) + q_{v,\text{ext.}}(t)$$

*and transport time $\delta t_{uv} = \min A_{uv}(t)$ as defined in Theorem C.13.*

*Proof.* **Well-definedness:** Since by assumption, for each $u \in \mathcal{N}(v)$, the set $A_{uv}(t) = A_{uv}(\{-l_{uv}, l_{uv}\}) \sqcup A_{uv}(0)$ as defined in Theorem C.13 is not empty, according to Theorem C.13.2, a transport time $\delta t_{uv} = \min A_{uv}(t) \in (0, \infty)$ exists for each $u \in \mathcal{N}(v)$. Even more, by Theorem C.13.3, is it uniquely determined by the condition that it is positive, allowing to define the injective map $\mathcal{N}(v) \to (0, \infty), u \mapsto \delta t_{uv}$ that associates a unique transport time with each neighbor $u \in \mathcal{N}(v)$. Especially to mention, for each $u \in \mathcal{N}(v)$, $\delta t_{uv} \in A_{uv}(t) = A_{uv}(\{-l_{uv}, l_{uv}\})$ is either an element of $A_{uv}(A_{uv}(\{-l_{uv}, l_{uv}\}))$ or $A_{uv}(0)$.

**Equality:** By (1) definition of the in-, out- and no-flow neighborhoods and (2) the conservation of flows (Eq. (4)), we obtain

$$\mathcal{N}_{\pm}(v, t) :\overset{(1)}{=} \{u \in \mathcal{N}(v) \mid \text{sgn}(q_{vu}(t)) = \pm 1\} \overset{(2)}{=} \{u \in \mathcal{N}(v) \mid \text{sgn}(q_{uv}(t)) = \mp 1\} \text{ and}$$

$$\mathcal{N}_0(v) :\overset{(1)}{=} \{u \in \mathcal{N}(v) \mid \text{sgn}(q_{vu}(t)) = 0\} \overset{(2)}{=} \{u \in \mathcal{N}(v) \mid \text{sgn}(q_{uv})(t) = 0\}.$$

Clearly, these three sub-neighborhoods define a partition of the neighborhood $\mathcal{N}(v)$, that is, they are pairwise disjoint and $\mathcal{N}(v) = \mathcal{N}_-(v, t) \sqcup \mathcal{N}_+(v, t) \sqcup \mathcal{N}_0(v, t)$ holds. Together with the definition of the ReLU function $\text{ReLU}(q) = \max\{0, q\}$ for all $q \in \mathbb{R}$, we observe that for each $u \in \mathcal{N}(v)$,

$$\text{ReLU}(q_{uv}(t)) = \begin{cases} q_{uv}(t) & \text{if } u \in \mathcal{N}_-(v, t), \\ 0 & \text{if } u \in \mathcal{N}_+(v, t), \\ 0 & \text{if } u \in \mathcal{N}_0(v, t) \end{cases}$$

and consequently,

$$\sum_{u \in \mathcal{N}(v)} \text{ReLU}(q_{uv}(t)) = \sum_{u \in \mathcal{N}_-(v, t)} q_{uv}(t)$$

and

$$\sum_{\substack{u \in \mathcal{N}(v) \\ \delta t_{uv} \in A_{uv}(\{-l_{uv}, l_{uv}\})}} \text{ReLU}(q_{uv}(t)) \cdot c_u(t - \delta t_{uv}) + \sum_{\substack{u \in \mathcal{N}(v) \\ \delta t_{uv} \in A_{uv}(0)}} \text{ReLU}(q_{uv}(t)) \cdot c_v(t - \delta t_{uv})$$

$$= \sum_{\substack{u \in \mathcal{N}_-(v, t) \\ \delta t_{uv} \in A_{uv}(\{-l_{uv}, l_{uv}\})}} q_{uv}(t) \cdot c_u(t - \delta t_{uv}) + \sum_{\substack{u \in \mathcal{N}_-(v, t) \\ \delta t_{uv} \in A_{uv}(0)}} q_{uv}(t) \cdot c_v(t - \delta t_{uv})$$

hold. Since all the appearing flows $q_{uv}(t) > 0$ are positive, by Theorem C.15.2, the transport times $\delta t_{uv}$ in the first sum are pass-through times ($\delta t_{uv} \in A_{uv}(\{-l_{uv}, l_{uv}\})$ and $q_{uv}(t) > 0$), while the transport times $\delta t_{uv}$ in the second sum are inverse self-loop times ($\delta t_{uv} \in A_{uv}(0)$ and $q_{uv}(t) > 0$). Since again by Theorem C.15.2, the other direction also holds, we can express the sums as

$$\sum_{\substack{u \in \mathcal{N}_-(v, t) \\ \delta t_{uv} \in A_{uv}(\{-l_{uv}, l_{uv}\})}} q_{uv}(t) \cdot c_u(t - \delta t_{uv}) + \sum_{\substack{u \in \mathcal{N}_-(v, t) \\ \delta t_{uv} \in A_{uv}(0)}} q_{uv}(t) \cdot c_v(t - \delta t_{uv})$$

$$= \sum_{\substack{u \in \mathcal{N}_-(v, t) \\ \delta t_{uv} \text{ is a pass-through time}}} q_{uv}(t) \cdot c_u(t - \delta t_{uv}) + \sum_{\substack{u \in \mathcal{N}_-(v, t) \\ \delta t_{uv} \text{ is a self-loop time}}} q_{uv}(t) \cdot c_v(t - \delta t_{uv}).$$

Since also the appearing transport times $\delta t_{uv} > 0$ are positive, by Theorem C.11, the concentrations $c_u(t - \delta t_{uv})$ in the first sum can be expressed as $c_u(t - \delta t_{uv}) = c_{uv}(t, l_{uv})$, while by Lemma C.9

and Theorem C.12, the concentrations $c_v(t - \delta t_{uv})$ in the second can be expressed as $c_v(t - \delta t_{uv}) = c_{uv}(t, l_{uv})$:

$$\sum_{\substack{u \in \mathcal{N}_-(v,t) \\ \delta t_{uv} \text{ is a pass-through time}}} q_{uv}(t) \cdot c_u(t - \delta t_{uv}) + \sum_{\substack{u \in \mathcal{N}_-(v,t) \\ \delta t_{uv} \text{ is a self-loop time}}} q_{uv}(t) \cdot c_v(t - \delta t_{uv})$$

$$= \sum_{\substack{u \in \mathcal{N}_-(v,t) \\ \delta t_{uv} \text{ is a pass-through time}}} q_{uv}(t) \cdot c_{vu}(t, l_{uv}) + \sum_{\substack{u \in \mathcal{N}_-(v,t) \\ \delta t_{uv} \text{ is a self-loop time}}} q_{uv}(t) \cdot c_{uv}(t, l_{uv})$$

$$= \sum_{\substack{u \in \mathcal{N}_-(v,t) \\ \delta t_{uv} \text{ is a pass-through time or} \\ \delta t_{uv} \text{ is a self-loop time}}} q_{uv}(t) \cdot c_{vu}(t, l_{uv}).$$

Nevertheless, by Theorem C.13.2, for each $u \in \mathcal{N}(v)$, $\delta t_{uv}$ is either a pass-through time, an inverse pass-through time, a self-loop time or an inverse self-loop time. However, again using all the appearing flows $q_{uv}(t) > 0$ are positive, by Theorem C.15.2, for each $u \in \mathcal{N}(v, t)$, $\delta t_{uv}$ is either a pass-through time or an inverse self-loop time. Therefore, given the definition of $\delta t_{uv}$, the additional information in the sum is redundant, yielding the overall result

$$\sum_{\substack{u \in \mathcal{N}(v) \\ \delta t_{uv} \in A_{uv}(\{-l_{uv}, l_{uv}\})}} \mathrm{ReLU}(q_{uv}(t)) \cdot c_u(t - \delta t_{uv}) + \sum_{\substack{u \in \mathcal{N}(v) \\ \delta t_{uv} \in A_{uv}(0)}} \mathrm{ReLU}(q_{uv}(t)) \cdot c_v(t - \delta t_{uv})$$

$$= \sum_{u \in \mathcal{N}_-(v,t)} q_{uv}(t) \cdot c_{uv}(t, l_{uv}).$$

Bringing it all together, we obtain

$$\frac{\sum_{\substack{u \in \mathcal{N}(v) \\ \delta t_{uv} \in A_{uv}(\{-l_{uv}, l_{uv}\})}} \mathrm{ReLU}(q_{uv}(t)) \cdot c_u(t - \delta t_{uv}) + \sum_{\substack{u \in \mathcal{N}(v) \\ \delta t_{uv} \in A_{uv}(0)}} \mathrm{ReLU}(q_{uv}(t)) \cdot c_v(t - \delta t_{uv}) + q_{v,\text{ext.}}(t) \cdot c_{v,\text{ext.}}(t)}{\sum_{u \in \mathcal{N}(v)} \mathrm{ReLU}(q_{uv}(t)) + q_{v,\text{ext.}}(t)}$$

$$= \frac{\sum_{u \in \mathcal{N}_-(v,t)} q_{uv}(t) \cdot c_{uv}(t, l_{uv}) + q_{v,\text{ext.}}(t) \cdot c_{v,\text{ext.}}(t)}{\sum_{u \in \mathcal{N}_-(v,t)} q_{uv}(t) + q_{v,\text{ext.}}(t)},$$

which equal to $c_v(t)$ according to the mixing at junctions (Eq. (22)).  $\square$

