# OpenReview forum: "MeGA-MP: Metric Graph Advection Message Passing - Solving Dynamical Processes on Metric Graphs with Graph Neural Networks"
_ICLR.cc/2026/Conference — Submitted to ICLR 2026_

### Official Review · Reviewer_HvHc · 2025-10-27

**Soundness:** 4
**Presentation:** 3
**Contribution:** 3
**Rating:** 4
**Confidence:** 2

**Summary:**

The paper introduces MeGA-MP, a novel message-passing learning-free operator designed to model dynamical processes governed by linear advection on metric graphs. MeGA-MP leverages semi-Lagrangian backtracing, a classic method for solving PDEs, to formulate message passing updates. The authors also provide theoretical analyses, including convergence guarantees and interpolation error bounds. Experimental results demonstrate MeGA-MP’s effectiveness both as a standalone numerical solver for advection-type systems and as a component within learnable models.

**Strengths:**

1. The paper is well-written and easy to follow.
2. The problem in concern is novel and interesting.
3. Using the method of characteristics to define message functions is an elegant way to incorporate advection priors into message passing.
4. Experimental results are generally convincing and include both synthetic PDE settings and real-world water distribution datasets.

**Weaknesses:**

1. The authors position their work at the “intersection of physics-informed ML, numerical PDE solvers, and neural PDE solvers.” However, this framing feels somewhat misleading, as it implies a general applicability that the proposed method does not have — not all PDEs can be reformulated into the specific graph advection form considered here. To improve clarity, the authors should qualify this statement and explicitly state that the method targets linear advection-type equations on metric graphs. Moreover, the paper’s main focus isn't about PDE solving — which fundamentally relies on the method of characteristics — and is more about its novel adaptation of this framework to metric graph structures.
2. For Table 1, runtime and computational cost comparisons with neural baselines are missing. Such results are important to evaluate the practical efficiency of MeGA-MP relative to standard MPNNs. Furthermore, the authors use EPANET-MSX to generate all the ground truth data for this experiment. However, they do not compare their method to EPANET-MSX as a baseline.
3. For Table 2, MeGA-MP’s performance appears comparable or weaker than classical numerical solvers (e.g., semi-Lagrangian, RK4). The paper should clarify under what conditions MeGA-MP offers benefits. Furthermore, computation time should be brought forward to fully evaluate the effectiveness of MeGA-MP. I would be more than happy to raise my score if computation-related statistics are provided.
4. The paper does not explore the performance of MeGA-MP under noisy, incomplete, or perturbed flow fields, which are common in real systems such as water networks. An ablation or discussion on robustness to noise, measurement error, and missing temporal samples would strengthen the empirical evaluation.

**Questions:**

1. How MeGA-MP works in noisy settings, which are common in real systems such as water networks?
2. The paper argues that general-purpose PDE solvers cannot be directly applied to advection on graphs. However, could one instead apply a neural PDE solver such as FNO to each edge, or integrate a traditional ODE integrator like RK4 for temporal updates? How would such hybrid baselines compare?
3. I am not an expert with metric graphs. I notice that only NNConv and PDE-GNN are used as baselines in the WDS task, but PDE-GNN performs poorly. Are these two methods considered state-of-the-art for dynamical learning on metric graphs?

---

> ### Author Response · Authors · 2025-11-24
> **Official Comment by the Authors, Part 1**
>
> We thank the reviewer for their time and their detailed and constructive feedback.
> We also appreciate their acknowledgement for the clarity in writing, the importance of the problem that we are addressing,  and on the theoretical and practical aspects highlighted.
> We have revised the manuscript according to their remarks and questions, and address each point below.
>
> > 1. The authors position their work at the “intersection of physics-informed ML, numerical PDE solvers, and neural PDE solvers.” However, this framing feels somewhat misleading, as it implies a general applicability that the proposed method does not have — not all PDEs can be reformulated into the specific graph advection form considered here. To improve clarity, the authors should qualify this statement and explicitly state that the method targets linear advection-type equations on metric graphs. Moreover, the paper’s main focus isn't about PDE solving — which fundamentally relies on the method of characteristics — and is more about its novel adaptation of this framework to metric graph structures.
>
> The referenced section that positions our work at the "intersection of physics-informed ML, numerical PDE solvers, and neural PDE solvers" describes how our work integrates into the corpus of ML research. We clearly state that we present a method tailored to the specific PDE of linear advection throughout the title, first paragraphs including the abstract as well as the lines immediately before the referenced section. We have made a few additions to the Introduction highlighted in blue and hope that by that improve the clarity to this end.
>
> > 2. For Table 1, runtime and computational cost comparisons with neural baselines are missing. Such results are important to evaluate the practical efficiency of MeGA-MP relative to standard MPNNs.
>
> > 3. For Table 2, [...]. Furthermore, computation time should be brought forward to fully evaluate the effectiveness of MeGA-MP. I would be more than happy to raise my score if computation-related statistics are provided.
>
> We agree that this is important and have added both a theoretical and an empirical runtime analysis, including computational complexity and wall-clock comparisons across baselines (s. [General Rebuttal Comment](https://openreview.net/forum?id=fxlrnYBOZ4&noteId=RXshR7SfSi)).
>
> > 2. [...] Furthermore, the authors use EPANET-MSX to generate all the ground truth data for this experiment. However, they do not compare their method to EPANET-MSX as a baseline.
>
> We use EPANET-MSX as the simulator to generate ground truth, the error that we report in Table 1 is computed between the EPANET-MSX result and the prediction of each method, hence we cannot include the simulator in Table 1. Instead we provide a qualitative result plotting the result of EPANET-MSX, MeGA-MP and other baselines in Figure 3 in Section 5.1 and Figures 10 to 17 in Appendix D.1.
>
> > 3. For Table 2, MeGA-MP’s performance appears comparable or weaker than classical numerical solvers (e.g., semi-Lagrangian, RK4). The paper should clarify under what conditions MeGA-MP offers benefits. Furthermore, computation time should be brought forward to fully evaluate the effectiveness of MeGA-MP.
>
> We are not trying to beat numerical baselines on Euclidean domains, experiment 2 is more about showing that we can compete with classical numerical solvers and thus have implemented a valid solver. The main benefits over other numerical (and also neural) solvers is its applicability to metric graphs and the formalization as a message-passing scheme, which allows the direct combination with other message functions including learnable components. The latter, we show with experiment 1 that includes reactions. To simply compute linear advection dynamics on Euclidean spaces, we recommend the classical methods and the highly optimized solvers that already exist for this problem.

---

> ### Author Response · Authors · 2025-11-24
> **Official Comment by the Authors, Part 2**
>
> > 4. The paper does not explore the performance of MeGA-MP under noisy, incomplete, or perturbed flow fields, which are common in real systems such as water networks. An ablation or discussion on robustness to noise, measurement error, and missing temporal samples would strengthen the empirical evaluation.
>
> > 1. How MeGA-MP works in noisy settings, which are common in real systems such as water networks?
>
> This indeed an interesting avenue for future research, which we are planning to address in the future.
>
> > 2. The paper argues that general-purpose PDE solvers cannot be directly applied to advection on graphs. However, could one instead apply a neural PDE solver such as FNO to each edge, or integrate a traditional ODE integrator like RK4 for temporal updates? How would such hybrid baselines compare?
>
> This is indeed a great idea and there already exist some work addressing metric graphs with such a domain decomposition approach: Blechschmidt et al. [1,2] use this concept to solve drift-diffusion dynamics on metric graphs; Laczkó et al. [3] do this for the time-independent Schrödinger equation. However, as we also discuss in Section 1, these approaches do no extend to advection-dominated problems with long-range spatial interactions that span multiple hops on the metric graph. Therefore, we try to solve the advective part of the dynamic system without learning. This allows MeGA-MP to focus the learning on very local dependencies (e.g. reactions), circumventing this issue.
>
> > 3. I am not an expert with metric graphs. I notice that only NNConv and PDE-GNN are used as baselines in the WDS task, but PDE-GNN performs poorly. Are these two methods considered state-of-the-art for dynamical learning on metric graphs?
>
> We improved the manuscript by highlighting the difference between advection on metric graphs and on non-metric graphs. For that, we also refer to our  [General Rebuttal Comment](https://openreview.net/forum?id=fxlrnYBOZ4&noteId=RXshR7SfSi). With that being said, approaches that implement *advection* are only applicable to *non-metric* graphs, while the approaches that address *metric* graphs cannot solve *advection* (as elaborated above). Therefore, we are the first ones that approach *advection* on *metric* graphs and classical baselines do not exist. We chose our baselines to cover a variety of architectures styles (NNConv as a "vanilla" GNN, GPSConv for graph transformer and PDE-GNN for dynamic-system based architectures).
> ***
>
> **[1]** Jan Blechschmidt, Jan-Frederik Pietschman, Tom-Christian Riemer, Martin Stoll, and Max Winkler. A comparison of PINN approaches for drift-diffusion equations on metric graphs. arXiv preprint arXiv:2205.07195, 2022.
>
> **[2]** Jan Blechschmidt, Tom-Christian Riemer, Max Winkler, Martin Stoll, and Jan-F Pietschmann. Physics-Informed DeepONets for drift-diffusion on metric graphs: simulation and parameter identification. In Proceedings of the 42th International Conference on Machine Learning (ICML), volume 267 of Proceedings of Machine Learning Research, pp. 4482–4506. PMLR, 2025.
>
> **[3]** Csongor L Laczkó, Mihály A Vághy, and Mihály Kovács. A transferable PINN-based method for quantum graphs with unseen structure. IFAC-PapersOnLine, 59(1):67–72, 2025.

---

> > ### Comment · Reviewer_HvHc · 2025-11-25
> >
> > Thank you for addressing some of my concerns. However, from the provided run-time statistics, I find MeGA-MP’s competitiveness appears limited to coarse discretizations. This raises concerns about scalability issues: does MeGA-MP fundamentally struggle on higher-resolution settings due to algorithmic overhead? Given these remaining uncertainties, I will maintain my original score.

---

> > > ### Author Response · Authors · 2025-11-26
> > > **Official Comment by the Authors**
> > >
> > > Thank you for your quick response.
> > >
> > > We don't agree, as to our knowledge, MeGA-MP is the first method that allows zero-shot generalization on metric graphs. It also scales linearly with the size of the graph, as shown in Section 3.3. Therefore, it is clearly competitive.
> > >
> > > The premise of experiment 2 is only to show that MeGA-MP behaves similar to popular numerical methods for the standard Euclidean domain, where we can compare to classical solvers, demonstrating competitive behavior. In addition to competitiveness, it is stable in scenarios where, e.g., RK4 is not. We apologize for the mistake of reporting runtimes in Table 6 even for configurations where RK4 diverges. We have updated Table 6 accordingly.

---

### Official Review · Reviewer_ziVw · 2025-10-27

**Soundness:** 3
**Presentation:** 3
**Contribution:** 2
**Rating:** 4
**Confidence:** 3

**Summary:**

This paper deals with solving differential equations on metric graphs, which is a very interesting an timely topic

**Strengths:**

Interesting problem with a clear goal. Numerically challenging examples are chosen.

**Weaknesses:**

The comparison to classical numerical methods is somewhat missing and the authors only introduce a machine learning model in the final steps without much discussion. I feel this to be more of a numerical analysis contribution than an ML paper.

**Questions:**

I remain a bit confused about this paper whether it really is a learning paper. The authors refer to previous work, e.g. current ICML, but there main contribution seems to be using the method of characteristics and a nodal condition that is labeled as a message passing equation.

- Why do the authors claim in the introduction that numerical approaches are having a hard time as they require large amounts of compute and memory. The paper cited by the authors of Blechschmidt et al. in ICML provides a finite volume implementation that is difficult to beat in the simulation case. In my opinion learning approaches shine for inverse or parameter identification approaches.
- What are the weights in equation (3)? The usual formulation on networks requires nodal conditions such as Kirchoff-Neumann conditions (only mentioned in the appendix). I guess the information aggregation plays this role for their formulation?
- What is the learning component of this paper? The message passing formulation seems to be a numerical approach for implementing nodal conditions. How would this compare to a Kirchoff-Neumann formulation? The MLP is only added to the mix in Section 5 without much discussion of the details.
- How does this scale for large graphs?

---

> ### Author Response · Authors · 2025-11-24
> **Official Comment by the Authors (Part 1)**
>
> We thank the reviewer for their time and their detailed and constructive feedback.
> We also appreciate their acknowledgement for the importance of the problem that we are addressing.
> We have revised the manuscript according to their remarks and questions, and address each point below.
>
> > The comparison to classical numerical methods is somewhat missing
>
> In both our experiments we compare MeGA-MP not only to neural, but also to classical numerical methods:
> * **Experiment 1** includes a comparison to the EPANET simulator that internally implements a Lagrangian solver for advection. We use this as the ground truth for the advection and advection-reaction dynamics.
> * **Experiment 2** Here we compare against a Runge-Kutta 4 and a semi-Lagrangian solver. However, there are no established implementations to apply these methods on metric graphs, that is why this experiment is done for a 1-dimensional Euclidean domain.
>
> > I remain a bit confused about this paper whether it really is a learning paper [...]
>
> We introduce a theoretically sound framework to solve a physical task using message passing and show that such a formulation can straight-forwardly be extended with learned components, such as MLPs.  We showcase a model with zero-shot generalization between graphs of drastically different sizes: The Hanoi network with 34 edges, and the and L-Town network with 909 edges. We believe that this constitutes a contribution to the ML research corpus.
>
> > Why do the authors claim in the introduction that numerical approaches are having a hard time as they require large amounts of compute and memory.
>
> The claim is meant to clarify the overhead of discretizing the edge spaces for, e.g. FVM or FDM, to be applicable and also stable.  For large graphs, this would quickly become prohibitive. E.g. for the L-Town example (~900 edges) discretizing 10m pipes into 0.1m segments scales the number of grid points to 900.000.  We avoid this with our method.
>
> > The paper cited by the authors of Blechschmidt et al. in ICML provides a finite volume implementation that is difficult to beat in the simulation case.
>
> We tried to utilize the FVM solver in Blechschmidt et al. [1] as an alternative solver, however, this would require modifications to the implementation of the node coupling conditions. The implementation as it is cannot model the directional characteristics of advection in-between two edges, i.e. flux balance (cf. below) breaks at this point.
>
> > What are the weights in equation (3)?
>
> Indeed, Equation (3) (information aggregation) plays the role of nodal conditions which can be applied for general dynamics on metric graphs.
> We specify the weights from the general Equation (3) for advection dynamics in Section 3 as $w_{uv}(t) = \big(\textstyle \sum_{u \in \mathcal{N}-(v,t)} q_{uv}(t) \big)^{-1} q_{uv}(t)$ (Equation (10)). They ensure the conservation of mass by normalizing inflow concentration by the total inflow volume (inflow rates). The inflow rate $q_{uv}(t) = \nu_{uv}(t) \alpha_{uv}$ per edge $e_{uv}$ is defined by the product of flow velocity $\nu_{uv}(t)$ and edge capacity $\alpha_{uv}$ (e.g. a cross sectional area). We elaborate on the physical background in Section 4, and more detailed in the beginning of Appendix C (Definition C.1).
> We noticed that introducing both, dynamic systems in metric graphs and the specific case of advection, at the same time can lead to confusion, and we modified the background paragraph in Section 2, *Advection on Metric Graphs* accordingly.
>
> ***
>
> **[1]** Jan Blechschmidt, Tom-Christian Riemer, Max Winkler, Martin Stoll, and Jan-F Pietschmann. Physics-Informed DeepONets for drift-diffusion on metric graphs: simulation and parameter identification. In Proceedings of the 42th International Conference on Machine Learning (ICML), volume 267 of Proceedings of Machine Learning Research, pp. 4482–4506. PMLR, 2025.

---

> ### Author Response · Authors · 2025-11-24
> **Official Comment by the Authors (Part 2)**
>
> > The usual formulation on networks requires nodal conditions such as Kirchoff-Neumann conditions (only mentioned in the appendix). I guess the information aggregation plays this role for their formulation?
>
> Indeed, our nodal conditions implement the Kirchoff-Neumann conditions in advection problems:
> Kirchhoff-Neumann conditions are usually formulated as a balance in flux at nodes.
> In our notation, this translates to
> $$\sum_{u \in \mathcal{N}-(v,t)} q_{uv}(t) ~ c_{uv}(t,l_{uv})=\sum_{u \in \mathcal{N}+(v,t)} q_{vu}(t) ~ c_{vu}(t,0)$$
> (ingoing mass flux is equal to outgoing mass flux).
> This equation is satisfied as a consequence of the outflow boundary condition (Definition C.1.2), the incompressibility of the flow field, basic transformations and the inflow boundary condition (Definition C.1.1):
> $$
> \sum_{u \in \mathcal{N}-(v,t)} q_{uv}(t) ~ c_{uv}(t,l_{uv})=c_v(t)\sum_{u \in \mathcal{N}-(v,t)} q_{uv}(t)=c_v(t)\sum_{u \in \mathcal{N}+(v,t)} q_{vu}(t)=\sum_{u \in \mathcal{N}+(v,t)} q_{vu}(t) ~ c_v(t)=\sum_{u \in \mathcal{N}+(v,t)} q_{vu}(t) ~ c_{vu}(t,0).
> $$
> Note that we have not formulated the property
> $$
> \sum_{u \in \mathcal{N}-(v,t)} q_{uv}(t)=\sum_{u \in \mathcal{N}+(v,t)} q_{vu}(t)
> $$
> explicitly in our paper before, but it is a property that any incompressible flow field naturally satisfies. We have added it to Appendix A, highlighted in blue.
>
> > What is the learning component of this paper? The message passing formulation seems to be a numerical approach for implementing nodal conditions.
>
> The message-passing scheme not only implements a coupling condition (as discussed above) as the *aggregation function*, but also a *message function* $\phi$ that solves the 1-dimensional advection PDE along the edge space via the method of characteristics. As is, this solves advection on metric graphs without learning, but also allows extending the *message function* with learnable components - an MLP in our case -  which we utilize to solve an advection-reaction system in experiment 1. This arrangement has the advantage of zero-shot generalizing to other, potentially larger graphs over purely data-driven models.
>
> > The MLP is only added to the mix in Section 5 without much discussion of the details.
>
> The MLP that we add is very simple and all the details are included in Appendix D.1, *Model*.
>
> > How does this scale for large graphs?
>
> **Time Complexity Scaling:** \
> MeGA-MP scales linearly with the number of edges. We have included a theoretical and practical runtime complexity (s. [General Rebuttal Comment](https://openreview.net/forum?id=fxlrnYBOZ4&noteId=RXshR7SfSi)). \
> **Accuracy Scaling**: \
> We show that the error of MeGA-MP does not depend on the size of the graph, but on the dependency structure between nodes that arises from the physical parametrization of the system (s. Theorem B.8).

---

### Official Review · Reviewer_NJwg · 2025-10-30

**Soundness:** 2
**Presentation:** 3
**Contribution:** 1
**Rating:** 2
**Confidence:** 4

**Summary:**

The authors claim that they introduce a new advection operator and that this have not been addressed so far.
They bring convincing results to show that this is indeed a needed operation.

**Strengths:**

Advection is indeed needed for graphs. The numerical experiments demonstrate that.

**Weaknesses:**

Advection was already introduced for graphs. There are a number of recent references. The author mention a few but fail to say why what they do it different. Furthermore, new theoretical results on graph advection were analyzed in recent work Mathematical Models and Methods in Applied Sciences Vol. 35, No. 5 (2025) 1237–1265, Modeling advection on distance-weighted directed networks



I find the novelty of the paper very minimal

**Questions:**

See above

---

> ### Author Response · Authors · 2025-11-24
> **Official Comment by the Authors**
>
> We thank the reviewer for their time and their constructive feedback.
> We also appreciate their acknowledgement for the importance of the problem that we are addressing.
> We address their remarks and questions below.
>
> > Advection was already introduced for graphs. There are a number of recent references. The author mention a few but fail to say why what they do it different. Furthermore, new theoretical results on graph advection were analyzed in recent work Mathematical Models and Methods in Applied Sciences Vol. 35, No. 5 (2025) 1237–1265, Modeling advection on distance-weighted directed networks
>
> There are indeed several works that address advection on (non-metric) graphs, e.g., [1, 2, 3], but there is an important distinction to our contribution addressing advection on *metric* graphs: \
> Existing works treat nodes of a graph as discrete grid points of an underlying continuous spatial domain. This representation constitutes a non-metric graph, since the graph edges only define proximity relationships across infinitesimal distances, but are not themselves treated as a domain on their own. In contrast, in a metric graph domain edges are associated with continuous 1D domains that are coupled at nodes. The system dynamics occurs along edges, and not across node-adjacent grid cells. This fundamental discrepancy makes existing operators incompatible with advection on metric graphs. \
> We have further highlighted this difference in the manuscript and refer to our [General Rebuttal Comment](https://openreview.net/forum?id=fxlrnYBOZ4&noteId=RXshR7SfSi) for details.
>
> **Does the paper *Modeling advection on distance-weighted directed networks* apply?** \
> The referenced work [1] does associate lengths to edges, but does not treat the edges as a own Euclidean domain. Instead, the work [1] defines the advection operator on page 2 as
>
> $$
> \[A_G f_t\](u) = \sum_{w \in N^+(u)} \omega_{uw}f_t(u) - \sum_{v \in N^-(u)}\omega_{uv}f_t(v),
> $$
>
> similar to related works [2,3].
> This ensures conservation of mass that only holds if the system does not evolve along 1-dimensional edge spaces, in which case this formula has to be changed to
> $$
> \[A_G f_t\](u) =
>     \sum_{w \in N^+(u)}
>     \omega_{wu}(t)f_t(w, u, 0+\epsilon)
> 	-
> 	\sum_{v \in N^-(u)}
>     \omega_{uv}(t)f_t(u, v, l_{uv} - \epsilon),
> $$
> to compute the conservation of mass not between neighboring nodes, but for an $\epsilon$-ball around the node $u$, accessing points along adjacent edges. We base our work on the latter formulation.
> ***
>
> **[1]** Michele Benzi, Fabio Durastante, and Francesco Zigliotto. Modeling Advection on Distance-weighted Directed Networks. Mathematical Models and Methods in Applied Sciences Vol. 35, No. 5, p. 1237–1265, 2025.
>
> **[2]** Airlie Chapman. Advection on Graphs. Semi-Autonomous Networks: Effective Control of Networked Systems through Protocols, Design, and Modeling, pp. 3–16, 2015.
>
> **[3]** Moshe Eliasof, Eldad Haber, and Eran Treister. Feature Transportation Improves Graph Neural Networks. In Proceedings of the 38th AAAI Conference on Artificial Intelligence, number 11, pp. 11874–11882, 2024.

---

> > ### Comment · Reviewer_NJwg · 2025-11-27
> > **Did not explain real difference**
> >
> > The difference between your interpretation and other work is minor. It is a simple scaling with respect to the edges. Even in other work the edges can have a weight of 1 or not at all. Therefore the contribution of this paper is minor.
> > I maintain my score

---

> > > ### Author Response · Authors · 2025-11-27
> > > **Official Comment by the Authors**
> > >
> > > > The difference between your interpretation and other work is minor. It is a simple scaling with respect to the edges.
> > >
> > > The reviewers statement merges two fundamentally different concepts.
> > > The physical definition of advection used in the work the reviewer references does not generalize to metric graphs, it only holds in an $\epsilon$-ball around each node. Beyond that - for metric graphs - the advection equation must be solved along the 1-dimensional edge domains themselves. This is what our message function implements via the method of characteristics, which in turn cannot be properly modeled by any "simple scaling".
> > >
> > > We again want to refer the reviewer to the [General Rebuttal Comment](https://openreview.net/forum?id=fxlrnYBOZ4&noteId=RXshR7SfSi](https://openreview.net/forum?id=fxlrnYBOZ4&noteId=RXshR7SfSi)), and especially to Section A and Figure 5 that conceptualize the difference. Our experiments support this theoretically founded difference and shows that MeGA-MP is the first model to achieve zero-shot generalization on metric graphs. We therefore think that we provide a big contribution to the newly evolving topic of *metric* graphs.
> > >
> > > > It is a simple scaling with respect to the edges. Even in other work the edges can have a weight of 1 or not at all.
> > >
> > > It is not clear what the reviewer means by "simple scaling with respect to the edges" and ask for clarification. If the reviewer knows of any work that solves advection on metric graphs, we would be grateful to get a corresponding reference so we can check the work.

---

### Official Review · Reviewer_u85F · 2025-10-30

**Soundness:** 2
**Presentation:** 1
**Contribution:** 1
**Rating:** 2
**Confidence:** 4

**Summary:**

The paper proposes MeGA-MP, a message-passing framework that models linear advection on metric graphs, iterating an MPNN-style update to propagate information between nodes. The authors provide theoretical results and two experimental evaluations: (i) advection-reaction forecasting on simulated water distribution systerms, where MeGA-MP outperforms baselines; and (ii) solving advection on a 1D domain comparing MeGA-MP to standard numerical solvers, where it achieves comparable perfomance.

**Strengths:**

- On 1D advection problems, the method achieves accuracy comparable to semi-Lagrangian and RK4 schemes across different discretizations.
- The appendices include detailed proofs and descriptions of the experimental setup, contributing to the theoretical rigor and partial reproducibility of the work.

**Weaknesses:**

Limited novelty and scope:
- Advection-based dynamics have already been explored in GNNs (i.e., Advection–Diffusion–Reaction GNN [1]). A comparison with ADR-GNN is therefore necessary to clarify MeGA-MP’s novelty and relative performance.
- The paper restricts its scope to linear advection, which limits applicability to more realistic systems that may also involve diffusion or nonlinearities.

Insufficient empirical comparison and limited experimental validation:
- The experimental evaluation is limited only to 2 GNN baselines and 3 tasks, weakening the empirical benefit of MeGA-MP. Authors should consider including [1], transformer based GNNs (e.g. [2,3]), dynamical system based GNNs (e.g., [4,5]), or other spatiotemporal GNNs (e.g., [6,7]).
- A complexity and runtime analysis (both theoretical and empirical) with respect to other baselines is missing; this would strengthen the claims of efficiency and scalability, and could further highlight the potential runtime advantages of MeGA-MP compared to baseline methods.
- The authors are encouraged to test MeGA-MP on spatiotemporal benchmarks such as Metr-LA and Pems-Bay [5] to demonstrate broader applicability to real-world problems, where coupling advection and reaction proved beneficial in [1].
- The paper lacks an ablation study comparing the learned and unlearned versions of MeGA-MP, which would help isolate and quantify the contribution of the learnable components to the overall performance.

Technical aspects lack clarity or are insufficiently detailed:
- The presentation is mathematically dense, with some overly complex notation (e.g., for defining forecasting problems). Including more intuitive explanations or illustrative examples in the main text would improve accessibility.
- To improve readability and ease of understanding, the paper should include pseudocode or a concise algorithmic summary of the proposed MeGA-MP method.
- It is unclear whether model selection (e.g., hyperparameter tuning, early stopping criteria) was performed properly and consistently across baselines.
- It is not clearly stated whether MeGA-MP evolves dynamics solely from initial conditions or whether it can incorporate external or time-varying inputs during simulation. Clarifying this point would help assess its flexibility for real-world forecasting tasks.

-----

[1] Eliasof et al. Advection Diffusion Reaction Graph Neural Networks for Spatio-Temporal Data. In LoG 2023

[2] Rampášek et al. Recipe for a General, Powerful, Scalable Graph Transformer. In NeurIPS 2022

[3] Shi et al. Masked Label Prediction: Unified Message Passing Model for Semi-Supervised Classification. In IJCAI 2021

[4] Gravina et al. Anti-Symmetric DGN: a stable architecture for Deep Graph Networks. In ICLR 2023

[5] Heilig et al. Port-Hamiltonian Architectural Bias for Long-Range Propagation in Deep Graph Networks. In ICLR 2025

[6] Li et al. Diffusion Convolutional Recurrent Neural Network: Data-Driven Traffic Forecasting. In ICLR 2018

[7] Wu et al. Graph wavenet for deep spatial-temporal graph modeling. In IJCAI 2019

**Questions:**

see weaknesses

---

> ### Author Response · Authors · 2025-11-24
> **Official Comment by the Authors, Part 1**
>
> We thank the reviewer for their time and their detailed and constructive feedback.
> We also appreciate their acknowledgement for theoretical rigor and reproducibility of our work.
> We have revised the manuscript according to their remarks and questions, and address each point below.
>
> ## Limited novelty and scope
>
> > Advection-based dynamics have already been explored in GNNs (i.e., Advection–Diffusion–Reaction GNN [1]). A comparison with ADR-GNN is therefore necessary to clarify MeGA-MP’s novelty and relative performance.
>
> We are familiar with ADR-GNN [1], but it is not directly applicable for the following reasons:
>  * **Different Problem Domain:**  ADR-GNN defines advection on non-metric graphs by treating nodes as discrete grid points of an underlying continuous spatial domain. This formulation does not apply to metric graphs, where edges represent continuous 1D domains and advection occurs along edges, not across node-adjacent grid cells. We clarified this distinction more explicitly in the revised manuscript [(s. General Rebuttal Comment)](https://openreview.net/forum?id=fxlrnYBOZ4&noteId=RXshR7SfSi).
> * **Incompatible Flow Field Assumptions:** ADR-GNN relies on a learned flow field with softmax-normalized unit outflow at each node. In contrast, flows in our tasks are physical, dynamic, and not constrained to unit outflow.
> * **Missing Edge Features:** Despite the concerns addressed above, one could train an ADR-GNN similar to NNConv and GPSConv. However, ADR-GNN does not make use of any edge features, which is why we needed to omit the knowledge of the flow field. As a result, it could not adapt to different flow conditions.
>
> For these reasons, we have decided against ADR-GNN as a baseline.
>
> > The paper restricts its scope to linear advection, which limits applicability to more realistic systems that may also involve diffusion or nonlinearities.
>
> We acknowledge this limitation and explicitly discuss it in the paper. Our goal is to establish a principled formulation of linear advection on metric graphs and demonstrate that message passing can reproduce the method of characteristics. We also show that the framework naturally accommodates learnable nonlinear reaction terms. Pursuing full nonlinear advection on metric graphs requires numerical solvers that, to our knowledge, do not yet exist.
> In this work we want to keep this focus and want to highlight that linear advection and reaction dynamics have utility, e.g., in hydraulics and water quality modeling, which we use as a benchmark for this reason.
>
> ## Insufficient Empirical Comparison and Limited Experimental Validation
>
> > The experimental evaluation is limited only to 2 GNN baselines and 3 tasks, weakening the empirical benefit of MeGA-MP. Authors should consider including [1], transformer based GNNs (e.g. [2,3]), dynamical system based GNNs (e.g., [4,5]), or other spatiotemporal GNNs (e.g., [6,7]).
>
> We have included the graph transformer [2] as a baseline to represent models able to capture global dependencies. As a representative for dynamical-systems-based GNNs we have included PDE-GNN [3]. We also checked the other proposed methods and decided against them for similar reasons as elaborated in the ADR-GNN case.
> To the best of our knowledge, no existing GNN architecture directly targets hyperbolic PDEs on metric graphs.
>
> > A complexity and runtime analysis (both theoretical and empirical) with respect to other baselines is missing; this would strengthen the claims of efficiency and scalability, and could further highlight the potential runtime advantages of MeGA-MP compared to baseline methods.
>
> We agree that this is important and have added both a theoretical and an empirical runtime analysis, including computational complexity and wall-clock comparisons across baselines [(s. General Rebuttal Comment)](https://openreview.net/forum?id=fxlrnYBOZ4&noteId=RXshR7SfSi).
>
> > The authors are encouraged to test MeGA-MP on spatiotemporal benchmarks such as Metr-LA and Pems-Bay [5] to demonstrate broader applicability to real-world problems, where coupling advection and reaction proved beneficial in [1].
>
> These datasets are indeed widely used, but they are agent-based traffic datasets and do not provide the physical quantities required by MeGA-MP: A metric graph with physically meaningful edge lengths and flow field. As such, applying MeGA-MP to these datasets would require artificially constructing physical inputs, making the evaluation and comparison to other methods unreliable. We emphasize that our focus is on physical modeling, and we have selected datasets aligned with that goal.
>
> > The paper lacks an ablation study comparing the learned and unlearned versions of MeGA-MP, which would help isolate and quantify the contribution of the learnable components to the overall performance.
>
> We agree that this is valuable and have added the requested ablation study [(s. General Rebuttal Comment)](https://openreview.net/forum?id=fxlrnYBOZ4&noteId=RXshR7SfSi).

---

> ### Author Response · Authors · 2025-11-24
> **Official Comment by the Authors, Part 2**
>
> ## Technical Aspects Lack Clarity or Are Insufficiently Detailed
>
> > The presentation is mathematically dense, with some overly complex notation (e.g., for defining forecasting problems). Including more intuitive explanations or illustrative examples in the main text would improve accessibility.
>
> We simplified the notation and added more intuitive explanations and examples in the manuscript. We also specifically simplified the definition of the forecasting problem [(s. General Rebuttal Comment)](https://openreview.net/forum?id=fxlrnYBOZ4&noteId=RXshR7SfSi).
>
> > To improve readability and ease of understanding, the paper should include pseudocode or a concise algorithmic summary of the proposed MeGA-MP method.
>
> We have added pseudo-code that describes the MeGA-MP update procedure [(s. General Rebuttal Comment)](https://openreview.net/forum?id=fxlrnYBOZ4&noteId=RXshR7SfSi).
>
> > It is unclear whether model selection (e.g., hyperparameter tuning, early stopping criteria) was performed properly and consistently across baselines.
>
> We have conducted hyperparameter tuning for all baselines using Optuna [4] and added an extended description, including search spaces for all baselines, to Appendix D.1, *Baselines*, highlighted in blue.
>
> > It is not clearly stated whether MeGA-MP evolves dynamics solely from initial conditions or whether it can incorporate external or time-varying inputs during simulation. Clarifying this point would help assess its flexibility for real-world forecasting tasks.
>
> MeGA-MP evolves dynamics from initial and boundary conditions and supports time-varying inputs at any node (e.g., control signals).
> However, since MeGA-MP solves the temporal dynamics in a distribution manner and not in a time-step-by-time-step style, the prediction horizon should be shortened in such situations. We implemented a [JavaScript demo](https://anonymous.4open.science/w/tmp-preprint-BB4F/) where injections can be controlled by the user. We will also make this demo publicly available as part of the supplementary material [(s. General Rebuttal Comment)](https://openreview.net/forum?id=fxlrnYBOZ4&noteId=RXshR7SfSi).
> ***
>
> **[1]** Moshe Eliasof, Eldad Haber, and Eran Treister. Feature Transportation Improves Graph Neural Networks. In Proceedings of the 38th AAAI Conference on Artificial Intelligence, number 11, pp. 11874–11882, 2024.
>
> **[2]** Ladislav Rampášek, Mikhail Galkin, Vijay Dwivedi, Anh Luu, Guy Wolf, and Dominique Beaini. Recipe for a General, Powerful, Scalable Graph Transformer. May 2022. doi: 10.48550/arXiv.2205.12454.
>
> **[3]** Luca Hermes, André Artelt, Stelios G Vrachimis, Marios M Polycarpou, and Barbara Hammer. A Benchmark for Physics-informed Machine Learning of Chlorine Concentration States in Water Distribution Networks. SN Computer Science, 6(5):522, 2025.
>
> **[4]** Takuya Akiba, Shotaro Sano, Toshihiko Yanase, Takeru Ohta, and Masanori Koyama. Optuna:
> A next-generation hyperparameter optimization framework. In Proceedings of the 25th ACM
> SIGKDD International Conference on Knowledge Discovery and Data Mining, 2019.

---

> ### Comment · Reviewer_u85F · 2025-11-26
>
> I thank the authors for their detailed response. However, several of my concerns regarding applicability to real-world scenarios, inclusion of baselines, and the use of real datasets remain. I have increased my score to acknowledge the effort and thoroughness of the rebuttal.
>
> **Regarding the insufficient Empirical Comparison.**
>
> To further strengthen the claim that MeGA-MP is particularly well suited to this problem, and to better better appreciate and position the advantage its advantages over classical GNNs, I continue to suggest comparing with:
> (i) at least [4] (which is also implemented in standard libraries such as [PyG](https://pytorch-geometric.readthedocs.io/en/latest/generated/torch_geometric.nn.conv.AntiSymmetricConv.html)), as it simulates a dynamical system, supports edge attributes and appears capable of effectively propagating information beyond local neighborhoods, an important property for the tasks addressed in the paper;
> (ii) a temporal GNN baseline (e.g., [[ROLAND](https://arxiv.org/pdf/2208.07239)], which can also support edge attributes), given that the problem is a classical spatiotemporal forecasting task and none of the current baselines explicitly model temporal information.
>
> Finally, could the authors clarify what backbone GPS uses? Does it incorporate positional encodings?
>
> ---
>
> [4] Gravina et al. Anti-Symmetric DGN: a stable architecture for Deep Graph Networks. In ICLR 2023
>
> [ROLAND] You et al. ROLAND: Graph Learning Framework for Dynamic Graphs. In SIGKDD 2022

---

> > ### Author Response · Authors · 2025-11-28
> > **Official Comment by the Authors**
> >
> > We thank the reviewer for the quick response and the constructive feedback.
> > We also appreciate the acknowledgement of effort and thoroughness and thank the reviewer for increasing the score.
> >
> > >  I continue to suggest comparing with: (i) at least [4] (which is also implemented in standard libraries such as [PyG](https://pytorch-geometric.readthedocs.io/en/latest/generated/torch_geometric.nn.conv.AntiSymmetricConv.html)), as it simulates a dynamical system, supports edge attributes and appears capable of effectively propagating information beyond local neighborhoods, an important property for the tasks addressed in the paper
> >
> > We agree with the reviewer that a comparison of our method to the A-DGN model [4] is interesting, especially since our baselines do not yet contain a model specifically made for long-range interactions. We have therefore included A-DGN in two variations:
> >
> > **A-DGN:** Similar to other baselines, we run our hyperparameter search (including number of iterations) for A-DGN and report the test error for the best configuration.
> >
> > **A-DGN$_{Dia}$** : Since our test graph is much larger, we replace the optimized number of iterations by the graph diameter, such that all spatial dependencies can theoretically be covered. Experimentally we found that we also have to change the step size $\epsilon$ to account for the increased number of iterations. We added the details to Appendix D, *Baselines*. For completeness, we want to disclose the results for A-DGN$\_{Dia}$ (A) without the adaptation of $\epsilon$, and for A-DGN$_{Dia}$ (B) with the adaptation of $\epsilon$. In the paper we only add the better performing version (B) with both parameters changed.
> >
> > |                   | Hanoi                         | L-Town                        |
> > | ----------------- | ----------------------------- | ----------------------------- |
> > | A-DGN             | 0.0283$\scriptsize\pm 0.0560$ | 0.0865$\scriptsize\pm 0.0915$ |
> > | A-DGN$_{Dia}$ (A) | 0.1386$\scriptsize\pm 0.1093$ | 2.7849$\scriptsize\pm 2.1797$ |
> > | A-DGN$_{Dia}$ (B) | 0.0597$\scriptsize\pm 0.0661$ | 0.1572$\scriptsize\pm 0.1278$ |
> >
> > > I continue to suggest comparing with: [...] (ii) a temporal GNN baseline (e.g., [[ROLAND](https://arxiv.org/pdf/2208.07239)], which can also support edge attributes), given that the problem is a classical spatiotemporal forecasting task and none of the current baselines explicitly model temporal information.
> >
> > We think that the [ROLAND] model would not be a good fit for our setting, since this method is tailored to dynamic graphs that are subjected to structural changes that happen irregularly over time. This is beyond our application. Moreover, the availability of GNN methods that perform node forecasting and incorporate edge feature vectors is quite limited. However, we are generally open to test other GNNs designed for spatiotemporal forecasting that also incorporate edges features.
> >
> > > However, several of my concerns regarding applicability to real-world scenarios, [...] and the use of real datasets remain.
> >
> > Our argument remains the same to this end:
> > The present work is situated at physics-informed machine learning that requires full knowledge of the relevant physical quantities (e.g. the flow and edge lengths).
> > While the extension of physical models to real-world data is very interesting, it is our goal to first explore the fundamental modelling strategies based on clean data from simulators. Since physical modeling on metric graphs is a newly emerging field and there is no other work for advection/hyperbolic dynamics on metric graphs, we think we are paving the way for a transfer to real-world data.
> >
> > Furthermore, metric graphs usually model complex systems from our daily life such as electrical grids, compressed air, or water distribution systems. Those systems belong to high risk infrastructure, about which data is not shared due to safety or competitive reasons. One usually relies on simulators such as EPANET as surrogates for the real world.
> >
> > > Finally, could the authors clarify what backbone GPS uses? Does it incorporate positional encodings?
> >
> > Please find the details in Appending D.1, *Baselines*. We use the GatedGCN as the message-passing layer, and the positional encodings are truncated Laplacian eigenvectors, because the random-walk-based encodings generally don't work if the node features can be zero.
> > We are currently running hyperparameter tuning to find whether NNConv or GINE work better as a backbone, since these are the models that we use for A-DGN as well. We are also checking the effect of relative positional encodings and will update the results should they improve.
> >
> > ***
> >
> > **[4]** Gravina et al. Anti-Symmetric DGN: a stable architecture for Deep Graph Networks. In ICLR 2023
> >
> > **[ROLAND]** You et al. ROLAND: Graph Learning Framework for Dynamic Graphs. In SIGKDD 2022

---

### Author Response · Authors · 2025-11-24
**General Rebuttal Comment**

We thank the reviewers for their time and constructive feedback on our work.
According to the feedback we received, next to smaller changes, we improved our manuscript as follows and highlighted all changes in blue:

**Advection on Metric Graphs vs. on Non-Metric Graphs**\
We highlighted the challenges of modeling advection on metric graphs (Appendix A, *The Challenges of Advection-dominated Dynamics on Metric Graphs*) and extended our elaborations from Section 2, *Advection on MetricGraphs*, on the two paradigms of modeling advection on metric graphs vs. on non-metric graphs by Figure 5.
This explains why conventional GNNs, even if they conceptually discretize the advection equation on an underlying Euclidean domain, are not able to address these challenges. Related modifications can additionally be found in Section 1, *Related Work*, and in Section 2, *Advection on Metric Graphs*.

**JavaScript Demo**\
We implemented a [JavaScript demo](https://anonymous.4open.science/w/tmp-preprint-BB4F/) that shows our method on the two benchmark graphs Hanoi and L-Town from experiment 1. This demo runs our model live, helps understanding the concept of advection on a metric graph and showcases its real-time applicability. We plan to make it publicly available as part of the supplementary material.

**Improved Mathematical Representation**\
We improved the mathematical notation whenever possible (e.g., in Section 2, *Problem Definition*, and in Section 3.3) and provided a less notation-heavy and more intuitive proof of Theorem 3.3 and 3.4. In this realm, we also moved the strategies behind our proofs to an own section (Appendix B.2) that - together with a guiding visualization (Figure 6) - can help to get a better intuition on the message passing iterations (Equation (12)). We reference the new section accordingly in Section 3.3, *Model Iterations*.

**Runtime Analysis including Pseudocode**\
We added a theoretical (Section 3, *Runtime Complexity*, and Appendix B.3) and practical (Section 5.1, *Results*, and Section 5.2, *Results*) runtime analysis of MeGA-MP. MeGA-MP scales similarly to conventional GNNs linearly with the graph size. At the benefit of generalizability, we sacrifice on runtime, but are still two orders of magnitude faster than the simulator EPANET (Table 1).

**More Baselines**\
We think that most baselines proposed in the reviews do not fit to our task. We address these issues in the reviewer-individual responses. However, we added a graph transformer [1] as a baseline to cover a global graph model and a dynamical-system based GNN [2] (Section 5.1, *Baselines* and *Results*, and Appendix D.1, *Baselines* and *Results*). As expected (and as discussed in the before-mentioned challenges of modeling advection on metric graphs with conventional GNNs), the new baselines do not compete with MeGA-MP and even more, the graph transformer performs worse than any other baseline (Table 1).

**Ablation Studies**\
We added ablation studies that investigate on the role of both the non-learnable advection-message-generation function $\phi$ and the learnable MLP (Section 5.1, *Baselines* and *Results*, and Appendix D.1, *Baselines* and *Results*). The experiments show that indeed, MeGA-MP does not require any learning components to model advection alone accurately while the learning components are necessary to model more complex, advection-dominated dynamics.

***
[1] Ladislav Rampášek, Mikhail Galkin, Vijay Dwivedi, Anh Luu, Guy Wolf, and Dominique Beaini. Recipe for a General, Powerful, Scalable Graph Transformer. In *Proceedings of the 36th International Conference on Neural Information Processing Systems (NeurIPS)*, 2022.

[2] Alessio Gravina, Davide Bacciu, and Claudio Gallicchio. Anti-Symmetric DGN: A Stable Architecture for Deep Graph Networks. In *Proceedings of the 11th International Conference on Learning Representations (ICLR)*, 2023.

---

### Author Response · Authors · 2025-11-28
**Response to the Data Leak**

Dear Reviewers and ACs,
in response to the data leak from OpenReview [1], we want to assure you that we did not, and will not try to gain access to this data to find out about identities of the persons involved in this reviewing process. We want to keep up trust and a fair reviewing process.
Best Regards, Authors

[1] [https://x.com/iclr_conf/status/1994104147373903893](https://x.com/iclr_conf/status/1994104147373903893)

---

### Meta-Review · Area_Chair_p8ob · 2025-12-23

**Summary:**

A well recongnised strength of the work is the conceptual clarity of modeling advection via the method of characteristics within a message-passing framework, which elegantly encodes physical priors.  The theoretical analysis, including error bounds and convergence guarantees, was appreciated for its rigor by a number of reviews.
Another point of appreciation (in particular by Reviewer HvHc) concerned the benchmarking on water network simulations, and the fact that the experimental analysis demonstrates that MeGA-MP is competitive with classical numerical solvers and outperforms some neural baselines in the target metric-graph setting.

On the other hand, the paper seems to have raised more concerns than positive responses.
The most significant point of contention is novelty: nearly all reviewers argued that advection on graphs has been studied before and viewed the distinction between metric and non-metric graphs as incremental, while the Authors considered this distinction fundamental. The distinction may actually be there, but it is the paper's duty to make this evident: in this sense, the limited analysis and referencing of relevant related literature on advection-based GNNs in the paper may have triggered a negative response in the reviewers. Connected to such novelty aspects, some reviewers questioned whether the contribution is primarily a numerical method reformulated as message passing, and asked for clearer justification of its contribution as an ML method.

Another shared concern was about the narrow scope of the contribution which may limit applicability to more complex real-world dynamics.
On the one end, the method is restricted to linear advection (Rev u85F). On the other, robustness to noise, incomplete flow information, and real-world measurement uncertainty was not explored and remains an open issue for the method. Some concerns were raised about runtime and scalability, with mixed views on whether MeGA-MP remains competitive at higher resolutions or large graphs.

Finally, there were repeated requests for broader and stronger baselines, including additional dynamical-system GNNs, temporal GNNs, and clearer positioning relative to numerical solvers.

**Reviewer Concerns:**

The authors partially clarified the distinction between advection on metric graphs and on non-metric graphs, which was positively acknowledged by u85F (at least to a partial extent), but not by NJwg, who still argues that the differences are not convincingly laid out by the Authors. I must admit that the rebuttal, in this sense, is not entirely satisfactory as the Authors cross reference other responses and paper sections rather than engaging with the reviewers to clarify on a point-by-point basis the differences.

The authors added both theoretical and empirical runtime analyses, improving transparency around computational complexity and scaling behavior. Also concerns about lack of ablation studies were resolved by including experiments that disentangle the contributions of the non-learnable advection operator and the learnable MLP components.

The limited scope to linear advection continues to constrain perceived impact, with extensions to nonlinear or fully realistic dynamics left for future work. Similarly, questions regarding robustness to noise and incomplete flow information were acknowledged but not empirically addressed.

Finally, requests for additional baselines were only partially addressed by adding graph transformers, PDE-GNN, and Anti-Symmetric DGN- Justifications were provided for excluding other requested comparisons, but these do not appeared entirely satisfactory in some cases (e.g. when considering the decision of ruling out a comparison with ROLAND or any other spatio-temporal network).

**Reviewer Scores:**

Reviewer u85F made explicit their willingess to raise the score, while maintaining criticism towards the paper. This would have most likely raised the score to 4.

Reviewer NJwg is evidently not satisfied by the positioning of the work with respect to relevant literature and its claim for novelty, so no change in score expected here.

Reviewer ziVw was not convinced by the ML component of the work and its correct positioning in ICLR scope, which remained an unresolved issues.

Similarly, Reviewer HvHc did not receive satisfactory responses and in their response to the rebuttal stated that they would not change the score.

Overall the likely final average score for the work could have ended up in between 3.5 and 4, which appears below the acceptance threshold.

---

### Decision · Program_Chairs · 2026-01-26

Reject